# Rapid and accurate polarimetric radar measurements of ice crystal fabric orientation at the Western Antarctic Ice Sheet (WAIS) Divide ice core site

Tun Jan Young[1], Carlos Martín[2], Poul Christoffersen[1], Dustin M. Schroeder[3,4], Slawek M. Tulaczyk[5], and Eliza J. Dawson[3]

[1]Scott Polar Research Institute, University of Cambridge, Cambridge CB2 1ER, United Kingdom
[2]British Antarctic Survey, Natural Environment Research Council, Cambridge CB3 0ET, United Kingdom
[3]Department of Geophysics, Stanford University, Stanford, CA 94305, USA
[4]Department of Electrical Engineering, Stanford University, Stanford, CA 94305, USA
[5]Department of Earth and Planetary Sciences, University of California, Santa Cruz, CA 95064, USA

**Correspondence:** T. J. Young (tjy22@cam.ac.uk)

**Abstract.** The crystal orientation fabric (COF) of ice sheets records the past history of ice sheet deformation and influences present-day ice flow dynamics. Though not widely implemented, coherent ice-penetrating radar is able to detect bulk anisotropic fabric patterns by exploiting the birefringence of ice crystals at radar frequencies, with the assumption that one of the crystallographic axes is aligned in the vertical direction. In this study, we conduct a suite of quad-polarimetric measurements consisting of four orthogonal antenna orientation combinations near the Western Antarctic Ice Sheet (WAIS) Divide Ice Core site. From these measurements, we are able to quantify the azimuthal fabric asymmetry at this site to a depth of $1400$ m at a bulk-averaged resolution of up to $15$ m. Our estimates of fabric asymmetry closely match corresponding fabric estimates directly measured from the WAIS Divide Ice Core. While ice core studies are often unable to determine the absolute fabric orientation due to core rotation during extraction, we are able to identify and conclude that the fabric orientation is depth-invariant to at least $1400$ m, equivalent to $6700$ years BP (years before 1950), and aligns closely with the modern surface strain direction at WAIS Divide. Our results support the claim that the deformation regime at WAIS Divide has not changed substantially through the majority of the Holocene. Rapid polarimetric determination of bulk fabric asymmetry and orientation compares well with much more laborious sample-based COF measurements from thin ice sections. Because it is the bulk-averaged fabric that ultimately influences ice flow, polarimetric radar methods provide an opportunity for its accurate and widespread mapping and its incorporation into ice flow models.

## 1 Introduction

There is a growing need to understand the dynamics of ice sheets and how they will respond to future climate change (IPCC, 2013). The flow of ice sheets is governed by the balance between the gravitational driving stress, basal resistance to sliding, and the internal deformation of ice (Cuffey and Paterson, 2010). Past flow history influences the ice crystal orientation fabric (COF), which, in turn, influences the present-day anisotropic ice viscosity and flow field. Because ice crystals effectively re-

orient themselves to minimise resistance when subjected to stress, the COF of ice is reflective of long-term strain at time scales proportional to the depth-age relationship (e.g., Alley, 1988), and has consistently been observed to align with ice deformation (Matsuoka et al., 2012). The COF of ice is also known to be influenced by perturbations in climate on a yearly timescale (Kennedy et al., 2013). Additionally, abrupt vertical changes in COF are often indicative of paleoclimatic transitions (e.g., Durand et al., 2007; Montagnat et al., 2014; Paterson, 1991). Therefore, an examination of the present-day COF can reveal past changes in the stress-strain configurations associated with historical ice flow (Brisbourne et al., 2019).

Ice core analyses represent the traditional method to quantify COF within ice sheets, and remain the only direct means of ground-truth observation. However, sites suitable for ice coring are often restricted to slow-moving ($< 50 \, \mathrm{m \, a^{-1}}$) sections of ice sheets, and therefore only reveal a subset of the dynamics between ice flow and COF. These slow-flowing areas most likely do not encapsulate the dynamics and physical processes responsible for ice sheet stability and sea level rise. Furthermore, ice core analyses are often unable to resolve the absolute direction of fabric orientation due to the rotation of the core in the barrel during or following extraction (Fitzpatrick et al., 2014). In contrast, ice-penetrating radar offers an alternative method to calculate anisotropic COF patterns at a bulk resolution (as opposed to the individual orientations of ice crystals) through exploiting the birefringence of polar ice without the practical limitations of drilling. Although polarimetric radar sounding data analysis has been implemented to detect horizontally-asymmetric COF for almost half a century (e.g., Hargreaves, 1977), it has not yet been widely implemented, with the majority of radar studies that measure COF variations in ice being conducted within the last fifteen years at coincident ice-coring sites for comparative analysis (Dall, 2010, 2021; Drews et al., 2012; Eisen et al., 2007; Ershadi et al., 2021; Fujita et al., 2006; Jordan et al., 2019, 2020a; Li et al., 2018; Matsuoka et al., 2003, 2009, 2012). Moreover, the majority of previous polarimetric radar studies infer COF at a coarse azimuthal resolution that is limited by the number of observations made along an acquisition plane that rotates around an azimuth centre (Brisbourne et al., 2019; Doake et al., 2002, 2003; Jordan et al., 2020c; Matsuoka et al., 2003, 2012). Because the maximum azimuthal resolution that can be achieved is subject to human error in measuring the angles between each acquisition plane, there is a coarse limit to the precision of the orientation of fabric asymmetry that can be achieved through this acquisition method.

Traditionally, radar studies estimated COF through power-based analyses by investigating the periodicity of birefringent patterns in power anomaly (Fujita et al., 2006; Matsuoka et al., 2012; Young et al., 2021) and phase difference measurements (Brisbourne et al., 2019). Recently, Jordan et al. (2019) developed a polarimetric coherence framework that extends existing methods (Dall, 2010; Fujita et al., 2006) to quantify COF eigenvalues through estimating the relative phase between orthogonal co-polarised measurements. A follow-up study by Jordan et al. (2020c) demonstrated the validity of this framework, albeit with a coarse azimuthal resolution via the azimuthal rotational setup, from measurements obtained using an autonomous phase-sensitive radio-echo sounder (ApRES). The ApRES is a phase-sensitive frequency-modulated continuous-wave (FMCW) ground-based radar system (Brennan et al., 2014) that has been gaining traction over the last five years, not only to investigate englacial polarimetry (Brisbourne et al., 2019; Jordan et al., 2020c), but also in wider radioglaciological investigations involving englacial deformation (Gillet-Chaulet et al., 2011; Kingslake et al., 2014, 2016; Nicholls et al., 2015; Young et al., 2019), englacial meltwater content (Kendrick et al., 2018; Vaňková et al., 2018), basal melting (Corr et al., 2002;

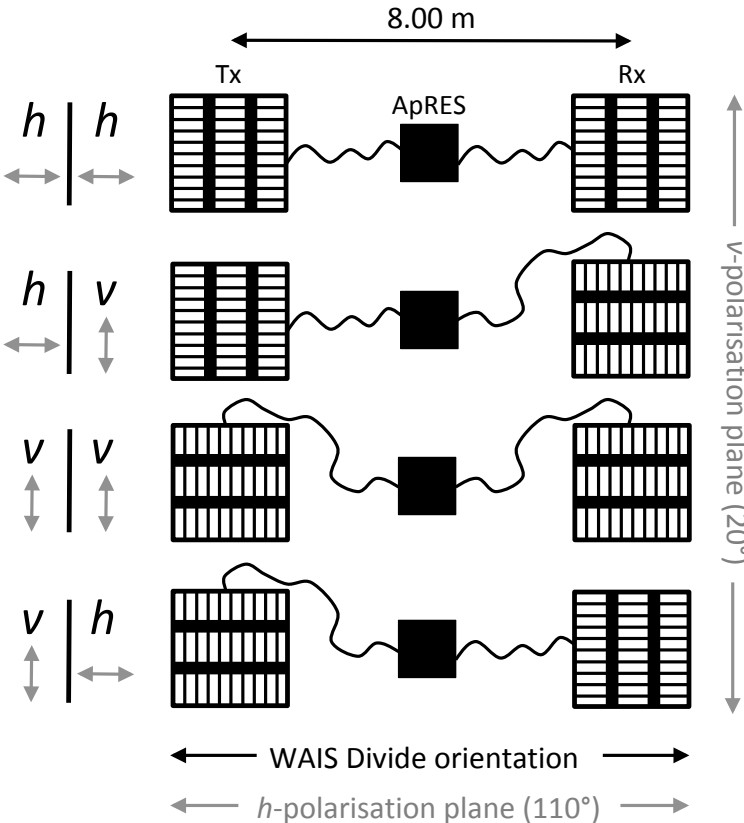

**Figure 1.** Four orthogonal combinations of antenna orientations as used for the polarimetry experiments in this study. Antennas (Tx = transmitting, Rx = receiving) were positioned 8.00 m apart and rotated 90° in a vertical (*v*) or horizontal (*h*) orientation. The orientation nomenclature (Tx | Rx) is with respect to the electric field of the antenna (grey arrows). The *h*-polarisation plane (110°) was oriented approximately parallel with the WAIS Divide orientation.

Davis et al., 2018; Jenkins et al., 2006; Lindbäck et al., 2019; Marsh et al., 2016; Nicholls et al., 2015; Stewart et al., 2019; Sun et al., 2019; Vaňková et al., 2020; Washam et al., 2019), and subsurface imaging (Young et al., 2018).

In addition to using an azimuthal rotational setup, acquisitions can also be obtained through a combination of four orthogonal antenna orientations, from which the received signal can then be reconstructed at any azimuthal orientation (Fig. 1) (Fujita et al., 2006; Jordan et al., 2019). This quadrature- (quad-) polarised setup significantly reduces the field time required to obtain each set of acquisitions.

In our study, we use an ApRES and two antennas to acquire quad-polarised measurements at the Western Antarctic Ice Sheet (WAIS) Divide. We apply the polarimetric coherence method to these measurements to present estimates of COF values that closely align with previous ice core COF measurements at WAIS Divide to a depth of 1400 m at a nominal bulk-averaged resolution of 15 m. We show that, using this setup and method, our estimates of fabric asymmetry are comparable to that from

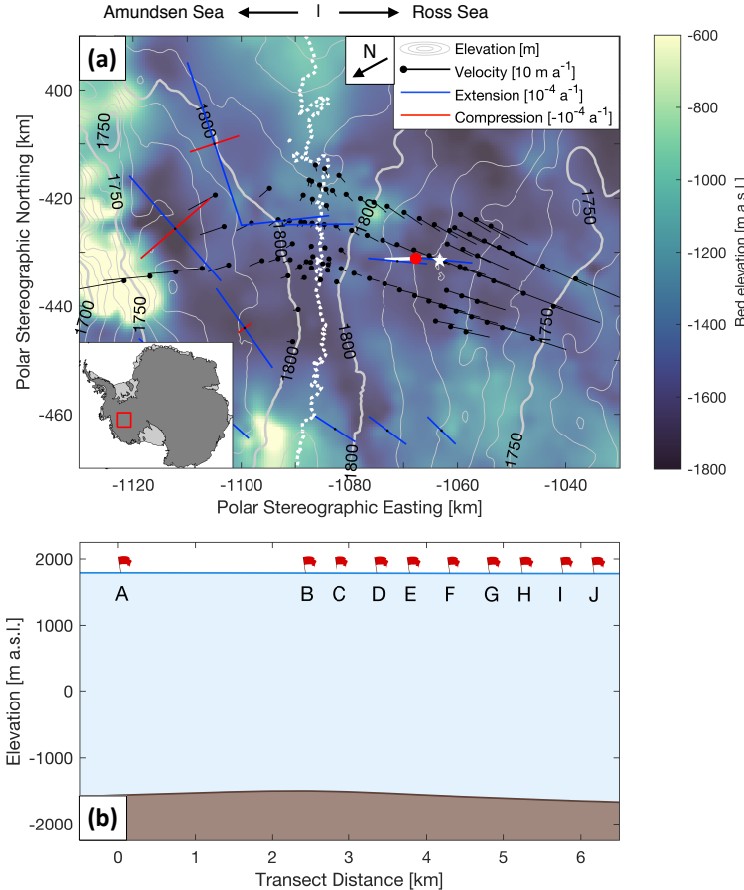

**Figure 2.** (a) Map of local surface (m a.s.l. with white contours at 10 m intervals, REMA, Howat et al., 2019) and bed topography (background colour, MEaSUREs BedMachine Antarctica, Morlighem et al., 2020) in the WAIS Divide area, as well as GPS-measured surface velocities (black lines, Conway and Rasmussen, 2009) and strain configurations (blue and red arrows, Matsuoka et al., 2012). Strain rates below $2.5 \times 10^{-5}$ a$^{-1}$ are not shown. The locations of the ApRES polarimetry transect (solid white line) and the results presented from Site I (red dot) in this study are ~5 km NE of the WAIS Divide Ice Core site (white star). The WAIS Divide is delineated as a thick dotted white line, with ice flowing northwards towards the Amundsen Sea and southwards towards the Ross Sea. Location of (a) is shown as a red box in the map inset. (b) Surface and basal topography along the ApRES polarimetry transect showing the relative locations of the 10 ApRES measurement sites. Site A is closest to the WAIS Divide, and Site J is closest to the core site. The results shown in Figures 4-6 depict the fabric profile of Site I, which has an estimated ice thickness of 3426 m from BedMachine.

ice core thin sections taken at similar depth intervals. From our results, we explicitly determine the principal axis of present and past flow.

## 2 Study area

The West Antarctic Ice Sheet (WAIS) divide delineates the surface topographic boundary separating ice flow towards the Ross and Amundsen Sea Embayments (Fig. 2a). A subglacial topographic saddle runs approximately orthogonal to the ice divide in the study region. Ice flow is oriented approximately SW ($230°$) and NNE ($10°$) in the Ross and Amundsen Sea catchments respectively (Conway and Rasmussen, 2009), which is offset from predicted strain configurations, especially in the Amundsen Sea catchment (Matsuoka et al., 2012). The WAIS divide flow boundary is observed to be migrating in the direction towards the Ross Sea at $10\,\mathrm{m\,a^{-1}}$, faster than the surface velocity of $3\,\mathrm{m\,a^{-1}}$, the migration being attributed to differential flow dynamics in the Ross Sea catchment (Conway and Rasmussen, 2009). Despite this imbalance, the ice-divide position has likely remained on average within $5\,\mathrm{km}$ of its present position throughout the Holocene epoch (Koutnik et al., 2016).

10 sets of polarimetric ApRES measurements were obtained along a $6\,\mathrm{km}$ transect approximately $15-20\,\mathrm{km}$ southeast of the WAIS divide in the Ross Sea embayment (Fig. 2b). A portion of our ApRES transect (Sites E to J) is spatially coincident with one of the active-source seismic profiles reported in Horgan et al. (2011). We present and analyse results from Site I in the main text body below, with equivalent results from all 10 sites available in the Supplementary Information (Fig. S1-S10). Site I ($79°\ 25'\ 53''$ S, $111°\ 59'\ 15''$ W, $1781\,\mathrm{m}$ a.s.l.) is approximately $5\,\mathrm{km}$ northeast of the location of the WAIS Divide deep ice core site ($79°\ 28'\ 3''$ S, $112°\ 5'\ 11''$ W, $1766\,\mathrm{m}$ a.s.l.), which was completed to a total depth of $3405\,\mathrm{m}$ (~$50\,\mathrm{m}$ above the ice-bed interface) (Fitzpatrick et al., 2014; Voigt et al., 2015) (Fig. 2a).

## 3 Theory and methods

We primarily follow a combination of three matrix-based methods—Fujita et al. (2006), Dall (2010), and Jordan et al. (2019), in which each study builds on the previous—to process the polarimetric measurements, thereby obtaining estimates of ice fabric anisotropy and orientation. Our approach is physically justified through an effective medium model that expresses the bulk dielectric properties of anisotropic polar ice in terms of the birefringence of individual ice crystals (Fujita et al., 2006). First, we follow the framework of Fujita et al. (2006) and Brisbourne et al. (2019) in modelling the expected power anomaly and phase difference (Section 3.2) to explicate the ApRES' response to the underlying COF parameters from data collected in proximity at WAIS Divide. We then calculate the azimuthal fabric asymmetry using the polarimetric coherence methods outlined in Jordan et al. (2019, 2020c).

### 3.1 Electromagnetic propagation and COF representation in anisotropic ice

In an anisotropic medium such as polar ice, birefringence and anisotropic scattering are two related, but separate mechanisms that affect the polarisation and azimuthal variation in power of radar returns (Brisbourne et al., 2019). For downward-looking ice-penetrating radar, birefringence occurs as a result of a phase shift between two orthogonally-oriented waves travelling between the surface and the interior of an ice mass, the phase shift manifested in the radar return as characteristic variations with azimuth and depth in power and phase. As a result, birefringence reflects the bulk COF and is azimuthally asymmetric

in the direction of radio wave propagation. On the other hand, anisotropic scattering arises as a consequence of rapid but microscopic continuous depth variations in the orientation of the bulk COF. Therefore, the polarimetric response of radio waves is determined by the bulk (macroscopic) birefringence of the COF (Hargreaves, 1978), of which the area illuminated by the waves is a function of the radar antenna footprint through depth. The birefringence of an individual ice crystal and their COF are related to the bulk dielectric properties of anisotropic polar ice (Appendix of Fujita et al., 2006):

$$\varepsilon(z) = \begin{bmatrix} \varepsilon'_\perp + \Delta\varepsilon' E_1 & 0 & 0 \\ 0 & \varepsilon'_\perp + \Delta\varepsilon' E_2 & 0 \\ 0 & 0 & \varepsilon'_\perp + \Delta\varepsilon' E_3 \end{bmatrix} \tag{1}$$

where $\varepsilon(z)$ is the bulk birefringence tensor with $z$ positive with increasing depth (and $x$ and $y$ in the horizontal directions), $\Delta\varepsilon' = \varepsilon'_\parallel - \varepsilon'_\perp$ is the crystal (microscopic) birefringence with $\varepsilon'_\parallel$ and $\varepsilon'_\perp$ the dielectric permittivities for polarisation planes parallel and perpendicular to each crystallographic ($c$)-axis. In this study, $z$ increases with depth with $x$ and $y$ oriented orthogonal to $z$. Across the spectrum of ice-penetrating radar frequencies and ice temperatures, $\varepsilon'_\parallel$ and $\varepsilon'_\perp$ vary within a narrow band of 3.16–3.18 and 3.12–3.14 respectively (Fujita et al., 2000). In this study, following Jordan et al. (2020c), we assign $\varepsilon'_\parallel = 3.169$ and $\varepsilon'_\perp = 3.134$, with $\Delta\varepsilon' = 0.035$.

The tensor eigenvalue $E$ describes the relative concentration of $c$-axes aligned with each principal coordinate eigenvector, with $E_1 + E_2 + E_3 = 1$ and $E_3 > E_2 > E_1$ following conventional radar notation, which is opposite to conventions normally in ice core studies ($E_1 > E_2 > E_3$). The relative proportions of $E$ can be used to describe different fabric patterns, including (i) random (isotropic) fabrics ($E_1 \approx E_2 \approx E_3 \approx 1/3$); (ii) cluster fabrics ($E_1 \approx E_2 \ll E_3$); and (iii) vertical girdle fabrics ($E_1 \ll E_2 \approx E_3$). When ice deforms solely by vertical uniaxial compression, such as at the centre of an ice dome, the $c$-axes rotates towards the vertical and forms a cluster fabric; where lateral tension exists from flow extension, such as at an ice divide, the $c$-axes orient in a vertical girdle distribution orthogonal to the direction of strain extension (Alley, 1988). In addition, fabric strength (its asymmetry or anisotropy) and orientation are also influenced to some extent by perturbations in climate (Kennedy et al., 2013). Following previous studies (Fujita et al., 2006; Drews et al., 2012; Brisbourne et al., 2019; Jordan et al., 2019, 2020c), we assume that the $E_3$ eigenvector is aligned in the vertical direction, and the $E_1$ and $E_2$ eigenvectors are parallel to the horizontal plane. The direction of the greatest horizontal $c$-axis concentration through depth corresponds with the $E_2$ eigenvector in our notation. The $E_1$ eigenvector is orthogonally oriented to both the $E_2$ and $E_3$ eigenvectors, and is sometimes referred to as the "symmetry axis" (e.g., Brisbourne et al., 2019). In the case of a vertical girdle fabric, the $c$-axes are oriented in a girdle that is planar to the $E_2$ and $E_3$ eigenvectors, with the $E_1$ eigenvector indicative of the orientation of lateral flow extension at its corresponding age-depth (Brisbourne et al., 2019; Matsuoka et al., 2012).

In the horizontal plane, Equation 1 simplifies to $\Delta\varepsilon(z) = \Delta\varepsilon'(E_2 - E_1)$, where the horizontal eigenvalue difference $E_2 - E_1$ quantifies the horizontal asymmetry of the crystal orientation fabric (i.e. strength of the vertical girdle). This equation directly relates the bulk-averaged ($\Delta\varepsilon$) and crystal ($\Delta\varepsilon'$) birefringence anisotropy to dielectric anisotropy, which serves as the basis for the radar processing methods that follow.

## 3.2 Modelling radio-wave signal propagation

The matrix-based formulation calculates the backscatter that is transmitted, reflected, and received at the antennas for each discrete scattering layer and azimuthal orientation:

$$S\left(\theta\right) = \left[\frac{\exp\left(jk_0z\right)}{4\pi z}\right]^2 \cdot \left[\prod_{i=1}^{N}\left(RTR'\right)_{\theta,\varepsilon,N+1-i}\right] \cdot \left[R\Gamma R'\right]_{\theta,\beta,N} \cdot \left[\prod_{i=1}^{N}\left(RTR'\right)_{\theta,\varepsilon,i}\right] \quad (2)$$

Equation 2 represents the polarimetric backscatter model described in Eqs. 9–12 of Fujita et al. (2006), which calculates the polarimetric backscatter for each antenna orientation combination as a function of angle in the horizontal ($\theta$), anisotropic scattering ratio ($\beta$), and birefringence ($\varepsilon$), through all depth layers $z = i$ to the $N$th layer. The first term on the right hand side represents the (i) free space propagation (squared to reflect two-way wave travel). Here, $j = \sqrt{-1}$ is the imaginary number, and $k_0 = 2\pi/\lambda_0$ $(\mathrm{rad\,m^{-1}})$ the wavenumber in a vacuum with $\lambda_0$ the wavelength in a vacuum. Besides the first expression, the second to fourth terms respectively represent three physical processes: (ii) received (upward) propagation; (iii) boundary scattering; and (iv) transmitted (downward) propagation to each boundary depth $i$. The rotation matrix $R$, with $R' = R^T$ its inverse, is used in Equation 2 to reconstruct the theoretical signal components with respect to $\theta$, for which the components are either $T$ (transmission between the antennas and the scattering layer) or $\Gamma$ (reflection at the scattering layer). $T$, $\Gamma$, and $R$ are all $2 \times 2$ matrices and are each detailed respectively in Eqs. 5 and 6, 8, and 10 of Fujita et al. (2006), with the propagation constants required to calculate $T$ defined specifically for the ApRES unit in Brennan et al. (2014).

In this study, anisotropic scattering is prescribed as a relative term. Following Fujita et al. (2006), $\beta$ is defined as the log-scaled ($20\log_{10}$) intensity anisotropic scattering ratio between the (electric field) Fresnel reflection coefficient along the $y$-polarisation plane relative to its equivalent in the $x$-polarisation plane (respectively the $(2,2)$ and $(1,1)$ elements in $\Gamma$). Therefore, a $\beta$ value of 0, 5, and 10 dB (for example) translates to the amount of anisotropic scattering in the $y$-polarisation plane being $10^0$ (i.e. equal to), $10^{\frac{1}{4}}$, and $10^{\frac{1}{2}}$ times stronger than in the $x$-polarisation plane.

## 3.3 Radar data acquisition

On 25 and 26 December 2019, we conducted radar experiments at 10 sites along a 6 km transect near the WAIS Divide ice core site (Fig. 2b). At each site, we acquired a suite of four quad-polarimetric measurements using a single-input single-output autonomous phase-sensitive radio echo sounder (ApRES; Brennan et al., 2014; Nicholls et al., 2015). The ApRES was operated with a linear up-chirp from 200 to 400 MHz over the course of 1 second, corresponding to a centre frequency of 300 MHz over a bandwidth of 200 MHz. An ensemble (burst) of 100 chirps were recorded for each polarimetric measurement. Two open-structure antennas (1 transmitting and 1 receiving) identical to those described in Nicholls et al. (2015) were used to transmit and receive each burst. Although not implemented in this study, we note that the four quad-polarimetric measurements can potentially also be obtained simultaneously in one single burst using a multiple-input multiple-output configuration (e.g., Young et al., 2018) with two transmitting and two receiving antennas.

The quad-polarimetric measurements represent the combination of orthogonal antenna orientations, where antennas were positioned in either a horizontal ($h$) or vertical ($v$) alignment with respect to the acquisition geometry. The four orientations

that correspond to the four polarimetric measurements are therefore (i) $hh$, (ii) $hv$, (iii) $vv$, and (iv) $vh$, the nomenclature reflective of the transmitting and receiving antenna in respective order (Fig. 1). Measurements were conducted sequentially, and each antenna pair orientation was established simply by rotating one or both antennas by $90°$ with respect to their previous orientations while keeping the position of each antenna centre constant. At all sites, the $h$-polarisation plane (ESE; $110°$) was aligned perpendicular to the transect line (NNE; $20°$). We follow Brisbourne et al. (2019) and assign a nominal $\pm 8°$ to their orientations. The nomenclature attached to the $h$ and $v$ alignments are indicative of the electric field (Fig. 1) and are consistent with those used in previous polarimetric ApRES studies of ice fabric (Jordan et al., 2019, 2020c, b; Ershadi et al., 2021) with the exception of Brisbourne et al. (2019), which reverses the two assignments (i.e. $h$ in our study corresponds to $v$ in their study, and $v$ in our study corresponds to $h$ in their study).

## 3.4 Radar data processing

Data were pre-processed and range-processed following procedures detailed in Stewart et al. (2019). Specifically, for each of the four bursts, the 20 noisiest chirps were culled and the remaining chirps averaged. Each resulting burst-mean was then weighted with a Blackman window, zero-padded, time-shifted to align the phase centre with the start of the signal, and Fourier-transformed. The resulting complex-valued spectra (referred to as their "complex amplitudes") store the amplitude and the phase of the signal as the magnitude and the angle of the spectra respectively.

Using the four processed quad-polarised complex amplitudes, the $2 \times 2$ integrated scattering matrix $S$ (Eq. 2) in the frame of reference can be constructed as (Doake et al., 2003):

$$S = \begin{bmatrix} s_{hh} & s_{hv} \\ s_{vh} & s_{vv} \end{bmatrix} \tag{3}$$

Equation 3 is often referred to as the Sinclair matrix. From here, we can reconstruct the ApRES received signal $S$ from any transmission angle through the application of an azimuthal (rotational) shift of principal axes at the transmitting and receiving antennas (e.g., Mott, 2006):

$$
\begin{aligned}
S(\theta) &= \begin{bmatrix} \cos\theta & -\sin\theta \\ \sin\theta & \cos\theta \end{bmatrix} \begin{bmatrix} s_{hh} & s_{hv} \\ s_{vh} & s_{vv} \end{bmatrix} \begin{bmatrix} \cos\theta & \sin\theta \\ -\sin\theta & \cos\theta \end{bmatrix} \\
&= \begin{bmatrix} s_{hh}\cos^2\theta - (s_{vh}+s_{hv})\sin\theta\cos\theta + s_{vv}\sin^2\theta & s_{hv}\cos^2\theta + (s_{hh}-s_{vv})\sin\theta\cos\theta - s_{vh}\sin^2\theta \\ s_{vh}\cos^2\theta + (s_{hh}-s_{vv})\sin\theta\cos\theta - s_{hv}\sin^2\theta & s_{vv}\cos^2\theta + (s_{vh}+s_{hv})\sin\theta\cos\theta + s_{hh}\sin^2\theta \end{bmatrix}
\end{aligned} \tag{4}
$$

In Eq. 4, the cross-polarised measurements obtained from the ApRES are in theory geometrically congruent by the Lorentz Reciprocity Theorem (i.e. $s_{vh} = s_{hv}$) and therefore the two measurements should be identical. In practice, there will be small differences including (but not limited to): (i) manufactured differences in the beam pattern between the transmitting and receiving antenna aerials; (ii) random clutter within the transmitted media; and (iii) human error in antenna positioning (Stumpf, 2018). In our datasets, we observe minimal difference between $s_{vh}$ and $s_{hv}$ throughout most of the measured ice column, although there is some additional variability seen at the near-surface (Fig. 3). Additionally, in our analyses, as well as those reported in Brisbourne et al. (2019), we find that $s_{vh} = -s_{hv}$, for reasons we have yet to identify.

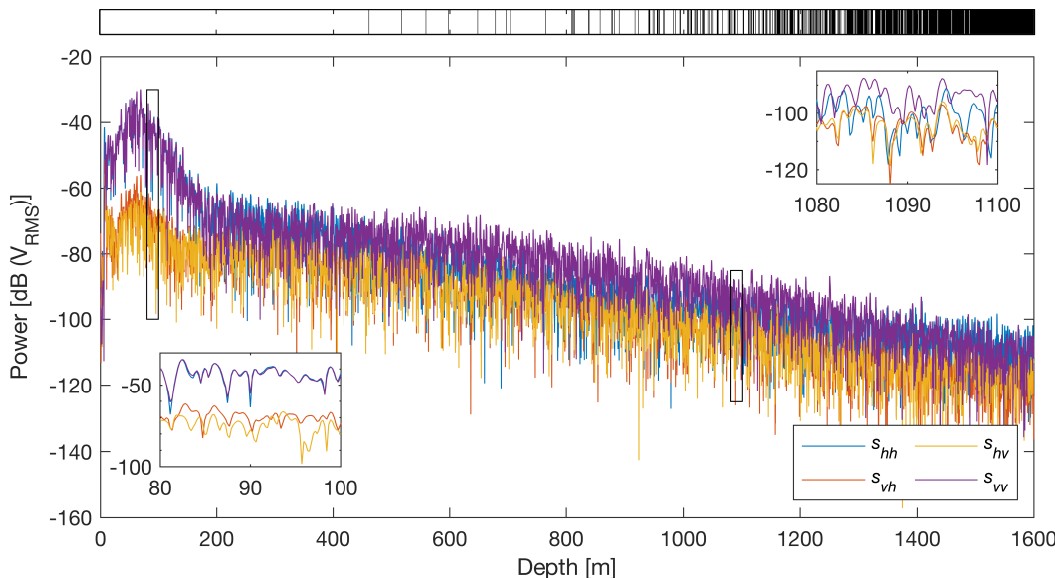

**Figure 3.** Mean (polarisation-averaged) power return for each antenna orientation combination acquired at Site I. Insets show magnification of power returns at two different 20 m intervals. Top bar shows range bins (black) that fall below the noise floor and were excluded from further analysis.

Because the complex amplitudes retrieved from the ApRES (Eq. 4) are phasors representing the radar return signal, the phase of the signal at a given depth is simply the argument of the complex number. To avoid the typical problems of working with phase—that is, employing phase unwrapping methods for sampled data within $[0, 2\pi]$—we calculate the phase difference with respect to azimuth:

$$\Delta\phi_{\theta,z} = \arg\left(s_{\theta+\Delta\theta/2,z} \cdot s_{\theta-\Delta\theta/2,z}^*\right) \tag{5}$$

where the asterisk represents the complex conjugate of its respective phasor.

The shift in phase also results in the modulation of received power as a function of azimuth. This can be visualised by calculating the power anomaly from the resulting multipolarisation data (e.g. Eq. 7 in Matsuoka et al., 2003).

The polarimetric coherence and its corresponding phase is computed over a local window via the discrete approximation

(Eq. 1 in Dall, 2010):

$$c_{hhvv}^{\star} = \frac{\sum_{i=1}^{N} s_{hh,i} \cdot s_{vv,i}^*}{\sqrt{\sum_{i=1}^{N} |s_{hh,i}|^2}\sqrt{\sum_{i=1}^{N} |s_{vv,i}|^2}} \tag{6a}$$

$$\phi_{hhvv}^{\star} = \arg\left(c_{hhvv}\right) \tag{6b}$$

where the superscript stars in Eq. 6 account for the use of the deramped phase stored by the ApRES rather than the original received signal phase, and we do not notate this explicitly hereafter (Eq. 7 in Jordan et al., 2020c). From Eq. 6b, we can then

estimate the horizontal, or azimuthal fabric asymmetry of the underlying ice column $E_2 - E_1$ by (Eqs. 22 and 23 in Jordan et al., 2019):

$$E_2 - E_1 = \frac{c}{4\pi f_c} \frac{2\sqrt{\varepsilon}}{f(\nu)\Delta\varepsilon'} \left| \frac{d\phi_{hhvv}(\alpha = 0°, 90°)}{dz} \right| \tag{7a}$$

$$\frac{d\phi_{hhvv}}{dz} = \frac{R\frac{dI}{dz} - I\frac{dR}{dz}}{R^2 + I^2} \tag{7b}$$

with the associated phase error (standard deviation) estimated through the Cramér-Rao Bound, following the methods of Jordan et al. (2019). In Eq. 7b, $R$ and $I$ respectively represent the real and imaginary components of $c_{hhvv}$. $f(\nu)$ represents a reduction parameter for the birefringence of firn with respect to solid ice, with $\nu$ the firn density, as detailed in the Appendix of Jordan et al. (2020c). Firn densities from Fig. 9a of Gregory et al. (2014) were used as values of $\nu$. The presence of $f(\nu)$ into Eq. 7a amplifies the estimated $E_2 - E_1$ values for the top ~100 m of the ice column. Firn correction was implemented only for the permittivity anisotropy in the horizontal plane and not for the mean propagation speed (i.e. no depth correction was made to account for the effect firn density on wave propagation speeds).

In this study, we use a pad factor of 2 (equating to a depth resolution of $0.27$ m) and an azimuthal resolution of $1°$ in the phase processing steps to produce co- and cross-polarised profiles of power anomaly and phase difference (Fig. 4b,c,d). Here, the pad factor represents the total length of the signal after zero-padding relative to the total length of the original signal. We restrict our observations of the co- and cross-polarised measurements only to measurements with sufficiently high signal-to-noise ratios (SNR). For each of the four acquisitions, the SNR was found by calculating the 95th percentile of the noise floor. Observations were excluded from the output if the magnitude of the complex amplitude of any one acquisition falls below the calculated SNR for any one acquisition at a given depth (Fig. 3). A depth window of $15$ m was used in the $hhvv$ coherence and phase estimates (Fig. 4e). We evaluated $d\phi_{hhvv}/dz$ using the real and imaginary components of $c_{hhvv}$ (Eq. 7b) and estimated its respective error following the suggested procedures in Jordan et al. (2019), with the exception that, in substitution of the finite impulse response filter, we used a 2-D median filter consisting of a $1° \times 5$ m matrix moved over the profile, and then a 2-D peak convolution using a Gaussian low-pass filter with the same moving matrix dimensions (Young et al., 2018). From here, estimates of $d\phi_{hhvv}/dz$ and their respective errors for each depth bin were both scaled using Equation 7a to then produce estimates and uncertainties for $E_2 - E_1$.

## 4 Results

### 4.1 Experimental results from WAIS Divide

Figure 4 shows the processed results from Site I, with equivalent results from the other 9 sites reported in the Supplementary Information. The results shown in Figure 4 are visually representative of all 10 sites. We compare the measured results in Fig. 4 with modelled results in Fig. 5 to parse the relative influence of birefringent propagation and anisotropic scattering on the ice column. From here, we estimate the strength of the azimuthal fabric asymmetry via Eq. 7a to solve for $E_2 - E_1$ (Fig. 6). We do not make any inferences in the uppermost $20$ m due to potential antenna radiation pattern effects.

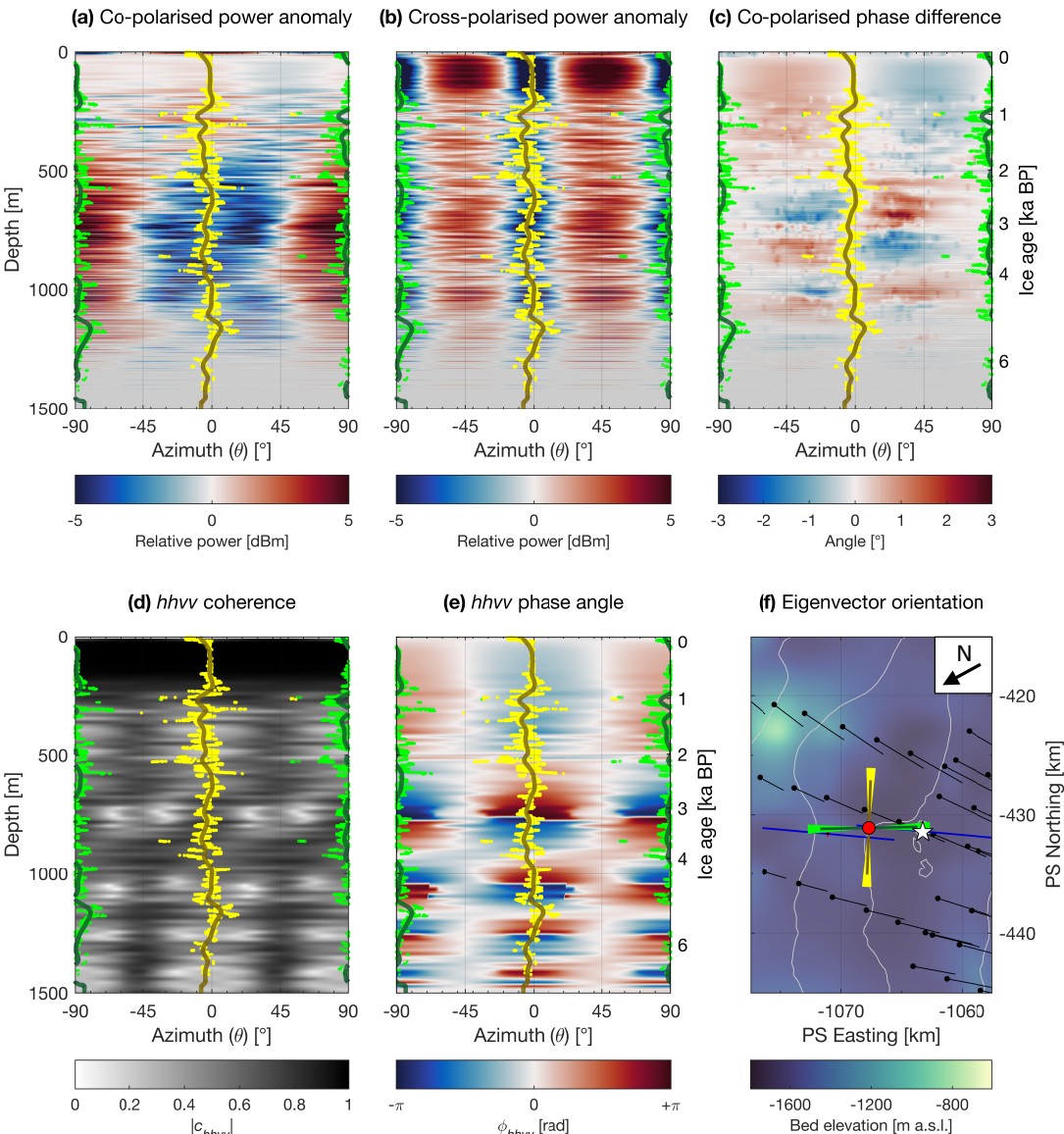

**Figure 4.** Polarimetric power and coherence measured using ApRES at WAIS Divide identifying the orientations of the $E_1$ (green) and $E_2$ (yellow) eigenvectors. (a) Co-polarised and (b) cross-polarised power anomaly. (c) Co-polarised phase difference. Quad-polarised ($hhvv$) (d) coherence and (e) phase angle. (f) Orientation of the $E_1$ and $E_2$ eigenvectors (dark lines) and their respective uncertainties (bright arc patches). The two eigenvector orientations (dark lines) were calculated using a Gaussian-weighted moving average of the azimuthal minima (bright dots). The depth-dependent gradient along the orientation of the $E_1$ eigenvector represents $E_2 - E_1$, the fabric asymmetry of the measured vertical ice column. Map shown in (f) is an inset of Fig. 2b, with ice flow oriented approximately SW ($230°$).

The depth-azimuth variation of the radar return power anomaly in the co-polarised and cross-polarised measurements are respectively shown in Fig. 4a and Fig. 4b, variations in the co-polarised return signal phase difference in Fig. 4c, the $hhvv$ signal coherence in in Fig. 4d, and variations in the $hhvv$ phase angle in Fig. 4e. Here, we observe azimuthal variations larger than 10 dB in the observed backscatter power both in the co-polarised and cross-polarised measurements, and variations larger than $6°$ in the co-polarised phase measurements. The SNR and the $hhvv$ coherence both remain relatively high for the uppermost 1000 m, with markedly lower $hhvv$ coherence values past ~1200 m (Fig. 4d) and the SNR failing to reach the prescribed threshold past ~1350 m (greyed-out sections in Fig. 4a-c).The backscatter variations in the co-polarised power anomaly (Fig. 4a) and $hhvv$ phase measurements (Fig. 4e) show near-reflectional symmetry at $90°$ at all measured depths in our experiment reference frame, with minor deviations occurring around ~1000 m and ~1400 m. The co-polarised phase difference measurements (Fig. 4d) show characteristic 'four-quadrant patterns' formed by azimuthal rotation of the phase-difference sign reversals (Brisbourne et al., 2019) along the same reflection axis. The cross-polarised power anomaly measurements, in contrast, show a $90°$ periodicity in the return power difference with a depth-constant azimuthal minima at $0°$ and $90°$ (Fig. 4c).

By tracing the azimuthal minima in the cross-polarised power anomaly profiles through depth (Fig. 4c), we can identify the orientations of the $E_1$ and $E_2$ eigenvectors (Li et al., 2018). However, because there exists a $90°$ ambiguity in the cross-polarised power anomaly profiles, we rely on the sign of the gradient of the $hhvv$ phase angle (Fig. 4e) to distinguish between the two. Because the $E_1$ and $E_2$ eigenvectors align with the orientations of the smallest and largest dielectric permittivities respectively, the location of the azimuthal minima resulting in a negative $\phi_{hhvv}$ gradient through depth indicates the direction of the $E_1$ eigenvector, and the azimuthal minima resulting in a positive $\phi_{hhvv}$ gradient indicates the direction of the $E_2$ eigenvector (Jordan et al., 2019). As a reminder, we are primarily interested in the orientation of the $E_1$ eigenvector, which is thought to be indicative of the direction of flow extension. Rounding to the nearest degree, we identify the orientations (with their associated standard deviation) of the $E_1$ and $E_2$ eigenvectors at $91°$ ($\pm6°$) and $-3°$ ($\pm6°$) respectively. Because a positive angular shift in the polarimetric reconstruction results in counterclockwise rotation (Jordan et al., 2019), these orientations correspond to cardinal directions of $19°$ (NNE) and $113°$ (ESE) respectively, with the same associated errors. The standard deviations attached to the eigenvector orientations are independent of the nominal $\pm8°$ arising from human error in antenna alignment during data collection. The identified horizontal eigenvector orientations both do not significantly change azimuthal orientation (within $\pm6°$) through the observed depth range to at least 1400 m, equivalent to a depth age of 6700 years (Sigl et al., 2016). Though we similarly observe no change in azimuth beyond 1400 m, we do not extend our findings further due to the limited depth samples with sufficient SNR available within this range.

Using the same methods, the orientation of the $E_1$ eigenvector identified for the other nine sites ranged from $82°$ (Site J) to $102°$ (Site D) and follows a normal distribution centred at $93°$ with a standard deviation of $\pm7°$ (to the nearest degree) (Table S1). For Site I, our estimate of the principal axis is $-7°$ to the nearest strain configuration (~5 km southwest) as estimated by Matsuoka et al. (2012), and is ~$+31°$ from the direction of flow as estimated by Conway and Rasmussen (2009) (Fig 4f).

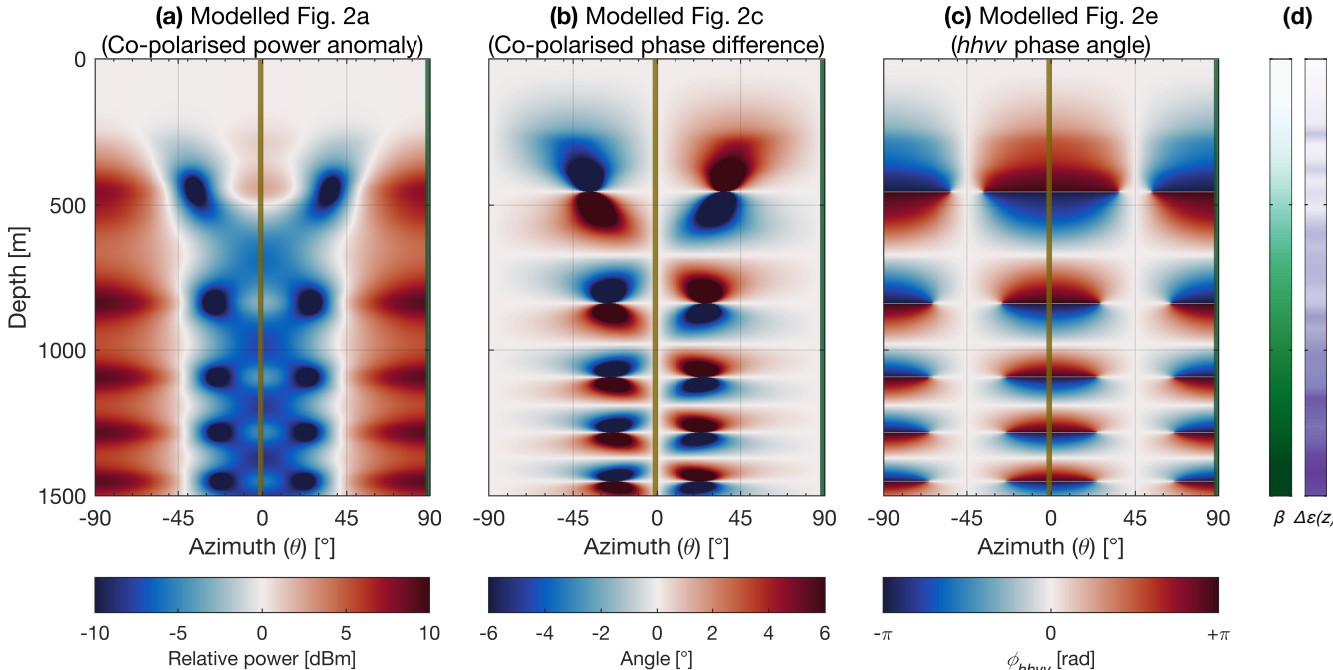

**Figure 5.** Modelled (a) co-polarised power anomaly, (b) co-polarised phase difference, and (c) quad-polarised ($hhvv$) phase angle, corresponding to Fig. 4a, c, and e respectively. (d) Model input parameters for the ratio of anisotropic scattering ($\beta$), and birefringence anisotropy ($\Delta\varepsilon(z)$) through depth, the former estimated through 2-D optimisation from Fig. 4b, and the latter using eigenvalues from the WAIS Divide ice core that specify the bulk COF (Fitzpatrick et al., 2014; Voigt et al., 2015). Value ranges for $\beta$ (white to dark green) and $\Delta\varepsilon(z)$ (white to dark purple) are $[0\ 15.6]$ dB and $[0\ 1.5\times10^{-2}]$ respectively. The orientation of the $E_1$ (dark green) and $E_2$ (dark yellow) eigenvectors were prescribed depth-invariant at $90°$ and $0°$ respectively following measured observations.

## 4.2 Comparison between observed and modelled polarimetric signals

The backscatter and phase patterns in the co-polarised measurements, as well as in the $hhvv$ coherence phase ($\phi_{hhvv}$), all show variation with depth, which indicate changes in either or a combination of birefringence and anisotropy. To better understand what drives these changes, we modelled the azimuth and phase dependence of these three measurements (Fig. 5) through the matrix-based backscatter model (Eq. 2), which predicts the combined polarimetric effect of birefringent propagation and anisotropic scattering at each depth and azimuth step (1 m and $1°$ in the model). The birefringence in the models ($\Delta\varepsilon(z)$) was estimated through Eq. 1 by directly using eigenvalue estimates from the WAIS Divide Ice Core (Fitzpatrick et al., 2014; Voigt et al., 2015) to calculate the horizontal asymmetry $E_2 - E_1$, and linearly interpolating between each defined fabric measurement depth. The eigenvalues from the ice core analysis suggest a gradual linear transition from an isotropic fabric ($E_2 - E_1 \approx 0.04$) near the ice surface to a moderately-strong girdle fabric ($E_2 - E_1 \approx 0.3$) at 1400 m depth (Fig. 6). For the model, we fixed the $E_1$ and $E_2$ eigenvector orientations at $89°$ and $-1°$ respectively. These values are $-2°$ and $+2°$

from their measured orientations of $91°$ and $-3°$ respectively, and this adjustment was made to satisfy the orthogonality of eigenvectors of a symmetric matrix. For simplicity, we prescribed both eigenvectors as depth-invariant, given that both their measured standard deviations were only $±6°$ across the measured depth range of $1500$ m (Fig. 4). The remaining parameter, the anisotropic scattering ratio $\beta$, was simply estimated as an optimisation problem through identifying the azimuthal distance of nodes (minima) in power anomaly from the principal axis orientation at $100$ m depth intervals. At each $100$ m depth interval (i.e. $100$ m, $200$ m, $300$ m, etc.), $\beta$ was estimated to the nearest integer before converting to the dB scale and linearly interpolated to match the model depth step of $1$ m. Beyond ~$1300$ m, where the SNR is deemed insufficient, we rely on the azimuthal range (width) of the alternating phase signatures in the $hhvv$ phase angle in Fig. 4e to make these estimates.

The model outputs for the co-polarised power anomaly and phase difference, as well as the $hhvv$ phase angle cases are shown in Fig. 5, and reinforces the observations of Brisbourne et al. (2019), where characteristic "node pairs" in the minima of the power anomaly (Fig. 5a) are coincident with the four-quadrant vertical pair centres in the co-polarised phase difference (Fig. 5b). These centres are also coincident with the width of the asymptotic zone of similarly alternating positive and negative phase gradients in the $hhvv$ coherence phase plot (Fig. 5c) (Jordan et al., 2019). Additionally, in Fig. 5a, cruciate-shaped zones of power minima link each node pair diagonally. In all three panels, the nodes and inflexion points are either reflectionally (Fig. 5a,c) or rotationally symmetric (Fig. 5b) around the $90°$ principal axis. The azimuthal distance of these nodes, centres, and the width of the asymptotic zones from the principal axis are a function of the ratio of anisotropic scattering (Fig. 5d). The depth-periodicity of pattern repetition is a function of birefringence, interpreted in past observations (e.g., Brisbourne et al., 2019; Drews et al., 2012) as a radar-measured phenomenon that arises as a consequence of bulk COF (Fig. 5d). In other words, the more closely spaced the nodes and quadrants are in depth, the stronger the azimuthal fabric asymmetry.

Several minor differences observed between the measured and modelled results include: (i) a scalar reduction in the overall ranges of power anomaly and phase difference values; (ii) the absence of the upper half of the shallowest phase reversal in the measured co-polarised phase difference and phase angle profiles (Fig. 4c,e); (iii) an offset in pattern repetition (e.g., nodes, quadrants, phase reversals) that increases in depth; and (iv) an absence of noticeable patterns in the shallowest $200$ m of the vertical ice column. Notwithstanding these differences, model results in Fig. 5 overall match their counterparts in Fig. 4a, c, and e to a high degree. The discrepancy between the received power returns of $s_{hv}$ and $s_{vh}$ (Fig. 3) as a result of a combination of human and instrumental error in the upper $200$ m (Stumpf, 2018) may potentially explain some of these caveats. Otherwise, the corresponding locations and sizes of nodes in the power anomaly, quadrants in the phase difference, and asymptotic zones predicted by the model can all be observed in the measured results at $200–1400$ m. Observations below $1400$ m depth remain inconclusive, due to insufficient SNR present at this depth range (Fig. 3).

### 4.3 Estimation of azimuthal fabric asymmetry

Estimates of $E_2 - E_1$, a measure of azimuthal fabric asymmetry, were made at Site I by calculating the depth-dependent gradient of the $hhvv$ phase difference along the $E_1$ eigenvector (Fig. 4e) at each $15$-m depth window to obtain $d\phi_{hhvv}/dz$ (Eq. 7b), and are shown alongside equivalent $E_2 - E_1$ measurements from the WAIS Divide ice core (Fig. 6). We do not calculate $E_2 - E_1$ in the uppermost $20$ m due to the antennas being subjected to near-surface non-axisymmetric antenna radiation pattern effects.

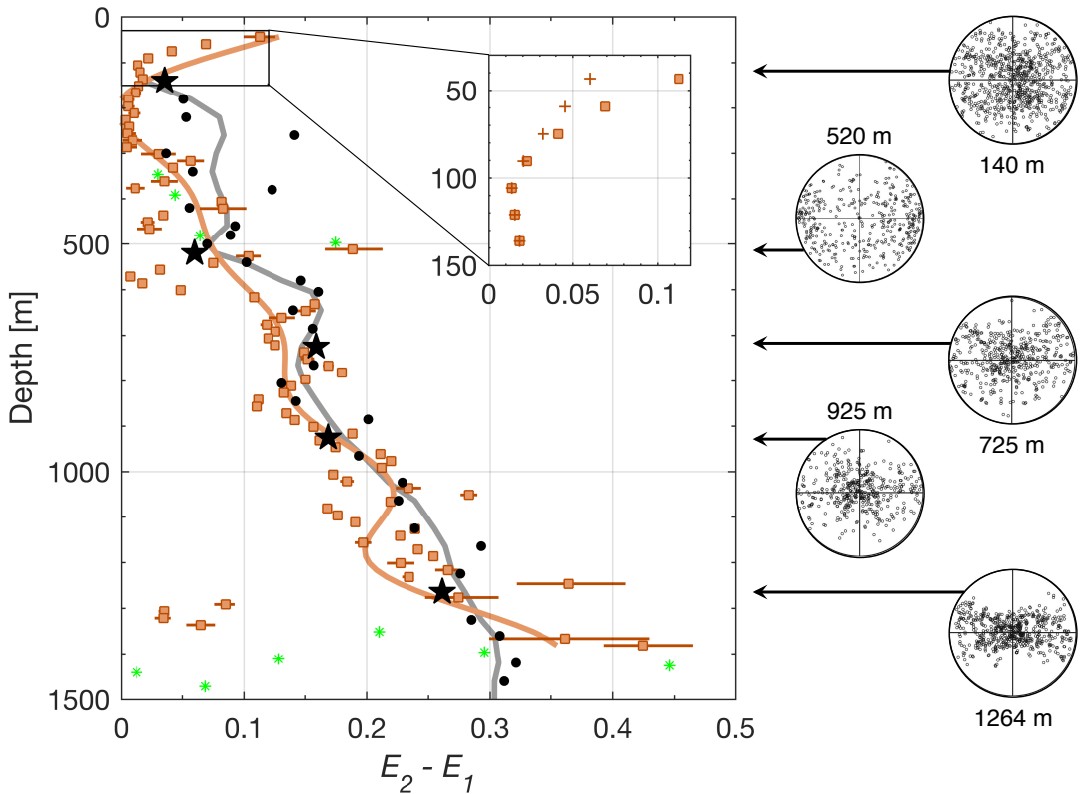

**Figure 6.** $E_2 - E_1$ values derived from the ApRES experiment (orange squares, with associated standard deviations) and from the WAIS Divide ice core (black dots, Fitzpatrick et al., 2014; Voigt et al., 2015). Smoothing curves were generated through a low pass filter on each dataset are shown in their respective colour scheme. Green asterisks display $E_2 - E_1$ values below a $|c_{hhvv}|$ threshold of 0.3. Inset displays $E_2 - E_1$ values with (orange squares) and without (orange crosses) firn correction applied to Eq. 7a. Firn correction was implemented only for calculating $E_2 - E_1$ and not for depth correction. A representative selection (black stars) of horizontal Schmidt plots from which the fabric eigenvalues for the WAIS Divide ice core were derived from are displayed to the right of the graph (Fitzpatrick et al., 2014). In all Schmidt plots, the $E_1$ eigenvalue points upwards.

Overall, fabric asymmetry estimates from ApRES match well to ice core estimates, especially between 600–1200 m, where the mismatch between the two depth series averaged less than 0.02. When comparing the two independently-calculated fabric asymmetry datasets at the discrete depths at which the ice core thin sections were extracted from, the resulting correlation was moderately high between 600–1200 m ($r^2 = 0.76$), and slightly lower over the entirety of the depth series overlap to 1400 m ($r^2 = 0.60$). Both datasets show a general increase in $E_2 - E_1$ with increasing depth, with ApRES fabric measurements overall showing more variability than estimates from the core site along their respective trends. The one exception to this positive correlation occurs between 100 and 400 m, where a small decrease in fabric asymmetry with depth can be observed in the uppermost 100 m of the ice column, where upon reaching minimum values at ~200 m, rebound and linearly increase beyond

this depth. This trend is exaggerated with the inclusion of firn correction (inset in Fig. 6). Even after filtering out $E_2 - E_1$ below a $|c_{hhvv}|$ threshold of 0.3 (green asterisks in Fig. 6), ApRES measurements beyond 1200 m show a marked increase in variability that, although centred around corresponding depth values in the WAIS Divide ice core, varied between 0.04 to 0.42. We similarly observe a sevenfold jump increase in the associated standard deviation, ranging from values averaging 0.006 at depths of 200–1200 m to 0.04 within the depth range of 1200–1400 m. There exists a small cluster of four outliers with low values ($E_2 - E_1 < 0.1$) at depths between 1250 and 1350 m with anomalously low error bars, even after initial $|c_{hhvv}|$ filtering. Setting increasingly higher $|c_{hhvv}|$ thresholds to 0.4 and 0.5 removes these outliers as well as all calculated $E_2 - E_1$ values beyond 1250 and 1100 m respectively. We do not calculate $E_2 - E_1$ from the ApRES record beyond 1400 m due to exceedingly low SNR and $|c_{hhvv}|$.

Estimates of azimuthal fabric asymmetry at the other 9 sites reveal similar trends, with those situated closer to the WAIS Divide ice core site in general showing a higher correlation with the core-derived fabric estimates (Fig. S11). The match between the ApRES fabric asymmetry and ice core COF estimates were generally higher at depths below 1000 m. Larger errors were observed where ApRES estimates deviated from the depth-coincident ice core measurement.

## 5  Discussion

### 5.1  Competing influences between anisotropic scattering and birefringence

The azimuthal dependency of backscattered power in ground-penetrating radar is a function of both anisotropic scattering and birefringence (Hargreaves, 1977). Although these two terms are related, they manifest from different electromagnetic phenomena. The birefringent propagation of radio waves arises from differences in dielectric permittivity along two axes perpendicular to the propagation direction, the two axes often referred to as the fast and slow axes or the ordinary and extraordinary axes. On the other hand, anisotropic scattering is a consequence of changes with depth in the anisotropic permittivity that may not necessarily be related to changes in crystal orientation fabric (COF) (Drews et al., 2012). Both phenomena often occur simultaneously, but our technique focuses on the analysis of the birefringent signals, which provides information on the bulk COF (Brisbourne et al., 2019).

The power anomaly model that we use to emulate measured results (Fig. 5a) incorporate a variable anisotropic scattering ratio $\beta$, which predicted an isotropic scattering medium at depths above 200 m that gradually increased in anisotropic scattering until at least 1400 m, where $\beta$ was estimated to be 15.6 dB. Drews et al. (2012) also observed azimuthal variations in backscatter power anomaly that varied similarly through depth, the variations which they attributed to microscopic (sub-metre) depth transitions in COF. The study also observed the superimposition of elongated bubbles that varied similarly with depth. While the induced polarimetric dependence through these bubbles was calculated to be minimal, where microscopic vertically-varying COF dominates the observed anisotropy, this effect was observed to be amplified at shallower depths. Although the observed anisotropic scattering can be exploited to infer the strength of the third eigenvalue ($E_3$) under assumptions of fabric isotropy at the ice surface (Ershadi et al., 2021), we do not attempt this method given our observations of significant fabric anisotropy in the firn layer (inset of Fig. 6). Although we calculate COF in our analysis, we cannot confirm the mechanisms for the observed

anisotropic scattering in our results, as the COF values represent not only a bulk-depth average within the calculated depth bin but also a manifestation of the horizontal rather than vertical asymmetry.

Our method applies density-dependent firn correction as suggested by Jordan et al. (2020c), which effectively reduces the value of $\Delta\varepsilon$ by taking into account the birefringence of firn with respect to solid ice. This in turn increases the azimuthal fabric asymmetry $E_2 - E_1$ (inset of Fig. 6). We observe slight fabric asymmetry that is most apparent in the uppermost $100$ m of the ice column that is inversely proportional to depth (Fig. 6). While there are observed deviations between the $s_{vh}$ and $s_{hv}$ received signals at the near-surface despite theoretical reciprocity between the cross-polarised terms (Fig. 3), this lack of reciprocity is independent of the fabric assymetry observed in the firn layer, which rely only on the co-polarised terms (Eq. 6a). As the applied firn correction is based upon established ice-air volume fractions in the mixing relations of Looyenga (1965), we believe these corrections to be physically representative of any fabric anisotropy within the firn layer. Separately, we are able to discount the possibility of a tilt angle between the $E_3$ eigenvector and the direction of radio wave propagation being the source of this observed firn asymmetry, given that the quad-polarisation measurements were conducted using a ground-based monostatic antenna setup (Jordan et al., 2019; Matsuoka et al., 2009). Crystal anisotropy in snow (Calonne et al., 2017) and firn (DiPrinzio et al., 2005; Fujita et al., 2009) have previously been observed on multiple occasions, and is thought to be induced by perturbations in climate such as temperature, solar radiation, winds, and deposition, where the combined effects of these variables influence the initial orientation and size of ice crystals (Kennedy et al., 2013), potentially amplifying the resulting ice flow dynamics (Wang et al., 2018). Although comparative studies addressing the physical origins of fabric anisotropy do not exist for the firn layer, it is likely that the effects of prolonged firn densification on crystal rotation will induce some amount of azimuthal anisotropy within this layer (Burr et al., 2017). While we cannot at this point conclusively link the observed fabric asymmetry in the uppermost $200$ m of the ice column to climate perturbations, it is certainly a plausible explanation especially given increasingly volatile climatic conditions over the larger Western Antarctic Ice Sheet over the past century that will likely intensify in the near future (Nicolas and Bromwich, 2011; Scott et al., 2019).

## 5.2   Flow history at WAIS Divide

With the addition of our ApRES-derived dataset, there are now three calculations of ice fabric at WAIS Divide. Each dataset was obtained using independent methods, with the other two arising from ice core observations (Fitzpatrick et al., 2014; Voigt et al., 2015) and sonic logging (Kluskiewicz et al., 2017). A fourth method, observed from shear-wave splitting in seismic surveys, was conducted in the 2018 summer at WAIS Divide and results gleaned from this seismic experiment (Nakata et al., *in prep*) would likely complement observations made from the corresponding datasets as an additional independent experiment.

We also note that an areal radar polarimetric study was conducted at WAIS Divide by Matsuoka et al. (2012), which quantified the relative orientation of the fabric asymmetry across a $60 \text{ km} \times 150 \text{ km}$ study area, but stopped short of calculating the COF structure within their polarimetric measurements. Across the study area, they observed depth-variable azimuthal shifts that varied according to the strain regime at the corresponding age-depth period. At the core site (their S-W24), however, their results were inconclusive due to the multiple unevenly-spaced azimuthal power maxima at depth in both radar frequency returns. Our results contrast with those of Matsuoka et al. (2012) in that we observe no azimuthal ambiguity in our determination

of the $E_1$ eigenvector orientation down to a depth of $1400$ m, which translates to a depth-age of ~6700 years BP (before 1950), which encompasses the majority of the Holocene epoch. The depth-age of the record is short because the rates of accumulation over this time period are high in comparison to both historical rates over the same area (Fudge et al., 2013) and present-day rates over other areas across the Antarctic Ice Sheet (Koutnik et al., 2016).

Although the time taken to overprint a pre-existing fabric is poorly constrained, excluding those from laboratory results (Brisbourne et al., 2019), the removal of previous fabric evidence is thought to take significant time and may require anomalously strong deformation regimes (Alley, 1988). At all sites, the alignment of our identified $E_1$ eigenvector orientation with the observed present-day strain regime is consistent with theory relating ice flow and crystal anisotropy (Azuma, 1994). We are confident that the observed surface strain orientation likely reflects the current deformation regime, given this alignment, the temporal permanence of fabric signatures, and the comparatively short depth-age of our record. Consequently, given that our ApRES measurement location is situated within $20$ km from the present-day location of WAIS Divide, our observations of depth-invariant eigenvector orientations with the above deductions support the proposition that the ice divide has likely remained on average within $5$ km of its present position throughout the Holocene (Koutnik et al., 2016).

## 5.3   Radar polarimetric methods to determine fabric strength and orientation

While older radar studies infer azimuthal fabric asymmetry at broad (100s to 1000s of metres) depth resolution by investigating the depth between sequential "co-polarised nodes" (e.g., Fujita et al., 2006; Matsuoka et al., 2003, 2012), more recent studies are able to quantitatively calculate the azimuthal fabric asymmetry via the polarimetric coherence method at comparatively higher resolutions (10s of metres) (Dall, 2010; Jordan et al., 2019; Ershadi et al., 2021). The ability to directly validate our results with measurements from the WAIS Divide ice core to a satisfactory degree gives confidence in our choice of processing parameters. We provide results using both methods (power anomaly/phase difference and polarimetric coherence), and use all outputs to arrive at our estimates of fabric orientation, thereby reconciling the aims of the two methods.

In this study, we were able to achieve coherent estimates of fabric strength at depth intervals (the bulk-depth resolution) down to $15$ m. In combination with a convolutional derivative, the use of depth averaging improves fabric estimates by reducing noise, removing anomalous "phase excursions", and isolates the effects of propagation-related phase behaviour with that from scattering Dall (2010); Jordan et al. (2019). A limitation of this method is that, due to the depth-averaging when calculating the $hhvv$ phase, it is not suited to detect and calculate fabric strength at and crossing fabric boundaries Jordan et al. (2019). Additionally, the size of the window and filter used is important, especially when applied over sections with high fabric asymmetry. As the depth periodicity of asymptotes present in the $hhvv$ phase angle (which manifests in phase wrapping) are proportional with azimuthal fabric strength (Fig. 5c), a large window has a greater risk of smoothing over these areas. This caveat may possibly be the reason behind the cluster of anomalously low measured $E_2 - E_1$ values at depth (Fig. 6). Conversely, a smaller window may naturally produce results with higher variability as a result of lower number of samples used to calculate the bulk-average. Therefore, with respect to the methods used in this study, there is likely a delicate balance between the bulk-average resolution and precision.

Jordan et al. (2019) puts forth the advantage of using polarimetric radar methods to determine the orientation of fabric, especially with regards to the fact that ice core studies can only be conducted in a relative azimuthal reference frame due to the rotational spin of the core during the drilling process. Our study confirms this proposition by establishing the orientation of the $E_1$ and $E_2$ eigenvectors through the identification of depth-local minima in the cross-polarised power anomaly measure-

ments (Fig. 4c). In this study, we distinguished between the two eigenvectors using the polarity of the $hhvv$ phase gradient $(d\phi_{hhvv}/dz)$ following Jordan et al. (2019). However, if anisotropic scattering is present, the azimuthal location of the four-quadrant patterns in the co-polarised phase difference is also an effective way to discriminate between the two eigenvectors (Brisbourne et al., 2019). Here, the four-quadrant patterns are centred around the $E_2$ eigenvector (Fig. 4c, Fig. 5c). Although the results of Brisbourne et al. (2019) observe the patterns to instead be centred around the $E_1$ eigenvector, we can reconcile

this discrepancy due to opposite assignments of $h$ and $v$ antenna alignments used between the two studies.

The cross-polarised power anomaly is generally a robust method of identifying the fabric orientation in slow-moving ice (Li et al., 2018). Here, we show that this method is reasonably accurate for depth-invariant eigenvectors (Fig. 4). In the case of a gradual rotation of the fabric orientation through depth, the cross-polarised power anomaly should undergo a similarly gradual rotation (Ershadi et al., 2021). This is also true in the case of an abrupt switch in COF, as evidenced at Korff Ice Rise

(Brisbourne et al., 2019), where the cross-polarised power anomaly undergoes a similarly abrupt shift in azimuth. In elementary cases, the $90°$ ambiguity that exists in the cross-polarised power anomaly (Li et al., 2018) can potentially be resolved from the methods given in the previous paragraph. However, if the fabric orientation were to change rapidly with depth, using only the cross-polarised power anomaly to determine and distinguish between the two eigenvector orientations may produce erroneous results as demonstrated by Ershadi et al. (2021). In all cases, if the radar-derived fabric orientation is offset in azimuth from its

true orientation, this mismatch will result in corresponding over- or under-estimation of azimuthal fabric asymmetry (Jordan et al., 2020b).

Because we did not conduct azimuthal rotational measurements at our study sites, we are unable to make a full and direct comparison between quad-polarimetric and rotational measurements in terms of their output results, and therefore are unable to advocate for one method over the other. However, a visual comparison between our results and those obtained at Site S-W24

of Matsuoka et al. (2012) show similar polarimetric power anomalies in the upper $1400$ m of ice, which give us confidence in our results. Separately, comparative analyses of results obtained using both types of measurements at Korff Ice Rise (C. Martín, *unpublished data*) as well as at EPICA Dronning Maud Land (Ershadi et al., 2021) reveal no structural differences between datasets. This comparative similarity may not hold in areas with more dynamic and/or complex flow, where the $E_3$ eigenvector is not vertically-aligned, and requires further investigation. While our estimation of the $E_1$ and $E_2$ eigenvector

orientations in our measurements is to the nearest $1°$, this precision reflects the angular bin size used to azimuthally reconstruct the received signal from quad-polarised data in this study, and is not synonymous with angular resolution, which instead is largely dependent on human errors in positioning the antennas for each acquisition (here assumed to be $\pm 8°$). However, under the assumption that the two acquisition methods do indeed produce physically equivalent datasets, then a quad-polarimetric reconstruction allows for a comparatively higher precision in identifying of the two eigenvector orientations.

The use of quad-polarised measurements is depth-limited by the signal-to-noise ratio of the cross-polarised terms ($s_{hv}$ and $s_{vh}$). Our COF measurements obtained from polarimetric radar, where chirps were coherently summed during pre-processing for each measurement, correlate well with equivalent ice-core measurements made to a depth of $1400$ m, after which estimated values become increasingly unconstrained and the phase is dominated by noise. Given that the relative power of the cross-polarised terms are almost a magnitude lower than that of the co-polarised terms (Fig. 3), there is a limit to the depth at which

ApRES is able to make accurate COF calculations. High SNR does not always equate to high polarisation coherence ($c_{hhvv}$), and vice versa. It is, however, plausible that larger datasets that employ higher amounts of chirp-averaging may increase the SNR needed to extend beyond the current depth limitation of $1400$ m.

### 5.4    Broader comparisons of geophysical methods to infer ice fabric properties

Of the methods available to quantify depth changes in ice COF, only ice core analyses are currently able to produce a fully

three-dimensional set of fabric estimates, and remain the only empirical measurement of COF. Thin (~10 cm) section analyses from ice cores, while providing direct orientation estimates for the majority of grains within each section, are often conducted at depth intervals of tens of metres, with each analysis capturing the local decimetre-wide fabric regime, as is the case for the WAIS Divide ice core at an average depth interval of $40$ metres (Fitzpatrick et al., 2014). In contrast, waveform-based methods average out fabric properties in bulk where, for radar systems, the planar footprint of which the COF is averaged

from is dependent on the radius of the first Fresnel zone (Haynes et al., 2018). This footprint would be approximately $6$ m in radius at a depth of $100$ m, and expands to approximately $23$ m in radius at $1400$ m. Therefore, the bulk COF estimates obtained from ApRES is averaged from a much larger area at depth than near the surface. Along with the SNR, this observed scale-dependence would therefore heavily influence the accuracy and error of results, especially in areas of complex flow and deformation.

The COF resolution vary significantly between waveform-based methods. For example, seismic surveys generate high azimuthal resolution at the cost of depth resolution (e.g., Horgan et al., 2011; Brisbourne et al., 2019), while equivalent results from sonic logging techniques reveal the converse (e.g., Gusmeroli et al., 2012; Kluskiewicz et al., 2017). While bulk-averaging over a large amount of crystals induces higher amounts of statistical noise compared to thin-section analysis, they are better able to resolve smaller scale features and discontinuities in the observed fabric (Wilen et al., 2003). Our results using a phase-

sensitive radio echo sounder, while reconstructed from orthogonal measurements, offer a compromise between azimuthal and depth precision, providing COF estimates comparable to results from the WAIS Divide ice core at resolutions comparable to that of seismics in azimuth and sonic logging in depth.

      As this study and those by Jordan et al. (2019, 2020c) show, the polarimetric coherence method measures the horizontal asymmetry of the vertical ice column through quantifying the birefringence effects from orthogonally-oriented measurements

and relating this to the difference in magnitude between the $E_2$ and $E_1$ eigenvectors. As such, an obvious limitation of this method is its inability to discern azimuthally-invariant fabric such as single-cluster crystal distributions or more complex fabrics such as horizontal girdles from each other. However, given the high resolution of the results, ApRES-derived COF are directly complementary to results from sonic logging in that the former calculates the horizontal asymmetry of the fabric through

quantification of $E_2 - E_1$, whilst the latter quantifies the vertical fabric asymmetry (the strength of the $E_3$ eigenvector) through P-wave interpretation (Kluskiewicz et al., 2017).

An obvious advantage of seismic and radar surveys, in comparison to ice coring and sonic logging, is that they can be implemented as a much smaller operation in terms of team size, cost, and field time, with ground-based radar surveys an order of magnitude even lower than that of seismics and airborne radar in all three aspects. As a reference, the quad-polarised measurements collected in this study took approximately thirty minutes inclusive of radar and antenna setup, and involved only two persons (co-authors Young and Dawson). While radar surveys can be conducted efficiently in comparison to that of seismics surveys, given the normalised eigenvalue framework in Eq. 1, radar measurements of ice fabric can at most provide information about the second order orientation tensor, which may be insufficient to describe the elastic anisotropy of ice (Sayers, 2018). In contrast, seismics wave propagation in anisotropic materials is based on a fourth-order elasticity tensor (Diez and Eisen, 2015; Sayers, 2018) that enables estimation of COF in three dimensions (albeit with a vertically-integrated value). The additional degrees of freedom enable seismic methods to distinguish azimuthally-symmetric forms of fabric anisotropy that equivalent radar methods at present cannot. A combination of these two methods can therefore reduce ambiguity in fabric estimation (Brisbourne et al., 2019), especially in areas of complex flow which have the potential to produce complex fabric. Preliminary results show that ApRES quad-polarised measurements are capable of reconstructing $E_3$ through inverse methods, with recovered values comparable to equivalent ice core measurements (Ershadi et al., 2021). Radar characterisation of the full fabric orientation tensor would represent a significant step forward in fabric measurements. Radar, in addition to seismics, have the ability to measure COF in a wide variety of flow regimes that may not be suitable as an ice core drilling site. In this respect, the use of one or a combination of geophysical methods has the potential to investigate more dynamic areas that reveal the complexities of ice flow, such as shear margins or grounding zones. In conclusion, as shown in our study, the use of an ApRES in conjunction with the polarimetric coherence method can produce estimates of fabric asymmetry with accuracies and intervals comparable to that of thin ice core section analyses.

## 6  Conclusions

Using a phase-sensitive radar and two open-structure antennas, we conducted a suite of quadrature-polarimetric measurements within the proximity of the WAIS Divide Ice Core. Using a combination of the matrix-based backscatter model (Brisbourne et al., 2019; Fujita et al., 2006) and the polarimetric coherence method (Jordan et al., 2019, 2020c), we were able to (i) quantify the horizontal asymmetry ($E_2 - E_1$) of the crystal orientation fabric (COF) to a depth of $1400$ m; and (ii) unambiguously identify the fabric asymmetry to be depth invariant with the $E_1$ eigenvector oriented at $19°\pm8°$ (relative to true North) to the same depth of $1400$ m. Our findings in (i) were conducted at an angular and depth intervals of $1°$ and $15$ m respectively, exceeding that of ice core-derived measurements made at the core site (Fitzpatrick et al., 2014; Voigt et al., 2015). The correlation between these two independent measurements of fabric asymmetry is moderately high over the depth range of $600$–$1200$ m ($r^2 = 0.76$), and is slightly lower over the entire depth range to $1400$ m ($r^2 = 0.60$). Our findings in (ii) are consistent with the direction of principal strain independently measured by Matsuoka et al. (2012). Our determination of depth-invariant fabric orientation

to at least 1400 m, equivalent to 6700 years BP (years before 1950), covers ~59% of the Holocene epoch, which suggests that the deformation regime at WAIS Divide has not changed substantially during this period. These observations of fabric orientation and strength were consistent for all 10 measured sites along a 6 km-long transect extending away from the core
site. While ice core-based measurements still represent the only method that empirically measures COF in three dimensions, the logistics required to conduct polarimetric radar measurements to quantify fabric asymmetry are minimal and non-invasive. In this regard, polarimetric radar methods provide an opportunity for accurate and widespread profiling and mapping of bulk COF across a diverse range of flow regimes, with the potential to illuminate the role of fabric asymmetry on ice rheology.

*Code and data availability.*  The full set of ApRES polarimetric measurements and associated processing code is available and accessible
through the UK Polar Data Centre as Young and Dawson (2021).

*Author contributions.*  TJY designed the polarimetric experiment with assistance from CM, who wrote the polarimetric backscatter model. TJY and EJD collected the ApRES data as part of the TIME project within the International Thwaites Glacier Collaboration. TJY processed the ApRES data, analysed the results, and wrote the manuscript with guidance from CM. All authors contributed to manuscript preparation.

*Competing interests.*  CM is an Editor of The Cryosphere.

*Acknowledgements.*  This work is ITGC Contribution No. ITGC-035, and is an output from the Thwaites Interdisciplinary Margin Evolution (TIME) project as part of the International Thwaites Glacier Collaboration (ITGC), supported by Natural Environment Research Council (NERC) research grant #NE/S006788/1 supporting TJY and PC, and National Science Foundation (NSF) research grant #1739027 supporting SMT and DMS. Logistics for this project were provided by the NSF-U.S. Antarctic Program and NERC-British Antarctic Survey. EJD is supported by an NSF Graduate Research Fellowship. We appreciate logistical support from the 2019/20 station leader (James King) and
staff at WAIS Divide Field Camp, as well as aviation support from the United States Air National Guard and Kenn Borek Air during our field season. We thank Jake Walter, Stephen Veitch, Forrest McCarthy, and Julie Baum as fellow team members in our field season. We thank all members of the ITGC TIME project as well as Thomas M. Jordan for insightful discussions on waveform analysis, ice fabric, and anisotropy. We similarly thank Keith Nicholls for insightful discussions as well as loaning the ApRES unit at short notice. We are grateful to Howard Conway for providing GPS-measured velocity data to contextualise our results. The authors would like to thank the Editor Kenichi
Matsuoka (Norsk Polarinstitutt), Martin Rongen (Johannes Gutenberg-Universität Mainz), and Reinhard Drews (University of Tübingen) for their constructive comments that improved this paper. The Schmidt plots in Figure 6 were reprinted from Figure 19 in Fitzpatrick et al. (2014) with permission from the International Glaciological Society as well as from the lead author.

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
