# Peer review of "Rapid and accurate polarimetric radar measurements of ice crystal fabric orientation at the Western Antarctic Ice Sheet (WAIS) Divide ice core site"

_The Cryosphere, 2020_

## Referee Comment (RC1) · Martin Rongen (Referee) · 30 Oct 2020

**General comments**

The authors present a quad-polarimetric radar measurement at WAIS Divide. The dataset is interesting in its innovative nature as well as in the quality of the derived results. The method follows in direct succession to earlier works developed by Shuji Fujita and Tom Jordan among others. While previous measurements required

a manual rotation of the antenna system in small steps in order to measure the azimuthal variation associated with the birefringence signature, the quad-polarimetric measurement allows for signal at arbitrary azimuths to be deduced from just four antenna orientations. The results are validated by comparison to ice core data.

The overall presentation is detailed and rigorous. Some improvements may be made by giving a clearer structure to the results and discussion sections (see later specific comments). As a non-expert on glaciological radar measurements, the theory and methods section was challenging but the provided references proofed to be very helpful. For the paper to stand on its own some more context/details may be added (see specific comments). Some more discussion may also be added to section 5.3 (Methods comparisons and limitations). It currently gives a fair comparison between radar and ice core / sonic measurements but is short on the specific limitations and assumptions involved in generating data for arbitrary azimuth angles using the quad-polarimetric data.

**Specific comments**

- The readability of the results section would be greatly improved by structuring it in sub-sections, as is also done for the other sections. A possible structure could be:

  - 4.1 Experimental results from WAIS (up to line 224)
  - 4.2 Modelling the observed data (lines 224-255)
  - 4.3 Fabric asymmetry estimation (lines 255 ff)

- The meaning of the pad factor mentioned in line 193 is unclear.

- The details and reliability of the firn correction as introduced in line 198 are unclear. It is mentioned that this correction amplifies the estimated $E_2 - E_1$ values in the shallow ice and surprisingly large fabric asymmetries are then measured in that depth range. Thus the firn correction merits more attention (and maybe test without the correction) during the discussion (line 265 and 305) may be warranted.

- In line 196 it may be worth mentioning that the Jordan et al. (2019) prescription to evaluate $d\phi_{hhvv}/dz$ is not actually based on the phase plot itself but on the real and imaginary components of the coherence as given in equation 7b.

- The reasons for and consequences of switching from an FIR filter to the method described in l.197ff remain unclear.

- The anisotropy parameter $\beta$ seems to be missing a unit (dB?) in the caption of Figure 3.

- For depth greater than 1200 m (and to a lesser extent around 600 m) the derived $E_2 - E_1$ values become rather unstable. While this is commented on and partially reflected in larger error bars, a population of outliers with small fabric asymmetries as well as small error bars is a bit worrying. It may be beneficial to show a plot of the coherence magnitude. Given vanishing magnitudes, the phase becomes unconstrained leading to erratic $d\phi_{hhvv}/dz$ values. In the deepest region the phase in Figure 2.e is more unstable as a function of depth than expected from the model calculation.

- The sentence "The birefringence of an individual crystal and its COF are related to the bulk ..." in line 97 reads a bit odd as a COF only applies to an ensemble of crystals. Maybe change to something like "the birefringence of individual crystals and their COF".

- The term "depth step" in line 124 is a bit technical. Something like "depth where a reflection occurs" would be clearer to the reader.

- The meaning and relevance of the rotation matrix $R$ as introduced in line 133 is unclear. To my understanding, it represents the rotation of the COF principle axis *of each traversed ice layer* with respect to a reference system defined by the antennas.

- It is mentioned that $s_{vh}$ and $s_{vh}$ should be identical given an ideal measurement. Has the difference between these two orientations been studied and the potential impact quantified?

- The COF orientation of each depth layer is resolved "by tracing the azimuthal minima in the cross-polarized power anomaly". While this may be a good approximation for this measurement, I wonder if this technique is generally applicable in the presence of strongly varying COF orientations. Assuming, for example, a constant angle of $20°$ in the top 500 m and a constant angle of $60°$ below, my understanding is that the minimum in the cross-polarized power anomaly would only slowly migrate towards $60°$ below 500 m as the bulk propagation is initially still dominated by the conditions above. One essentially measures the average COF orientation up to the scattering depth. But there may be a misunderstanding on my part here. A comment would be appreciated.

- As noted in the general comments section 5.3 would benefit from a discussion of the specific limitations and assumptions involved in generating data for arbitrary azimuth angles using the quad-polarimetric data.

- Section 5.1 paragraph 2 (lines 281-288) seems better suited in section 4.1 (results, modeling), here a Figure similar to Figure 5 in the Fujita 2006 paper may also be illustrative, showing that birefringence results in nodes in the power

anomalies while anisotropic scattering results in a band structure, the spacing of which is a function of the scattering strength.

**Technical comments**

- The link for the Mott, H. (2006) reference appears to be dead.

- In line 156, spectra so should be plural as are the amplitudes.

- In the title of subsection 5.1 it would be more consistent to refer to "anisotropic scattering" instead of the more ambiguous "anisotropy"

---

## Referee Comment (RC2) · Reinhard Drews (Referee) · 19 Nov 2020

Remark 1: I apologise for the delayed submission of this review. I had warned the editor that this would be the case, but also realise that this is obviously of no great help to the authors.
Remark 2: I had at first declined the review due to a perceived conflict of interest as we are currently working on a quite similar topic (Ph. D. thesis M. Ershadi who contributed to this review). However, after discussing with the editor, we decided to move on anyway. I hope that this review is perceived as constructive & helpful.

**Summary**

In their paper "Rapid and accurate polarimetric radar measurements of ice crystalfabric orientation at the Western Antarctic Ice Sheet (WAIS) Divide deep ice core site", Young and co-authors present an ApRES radar dataset, which they use to infer the ice-fabric characteristics continuously to a depth of 1500 m. Main results include quantification of the horizontal ice anisotropy with a depth invariant ice-fabric orientation that is aligned with the directions of the principal strain rates. The inferences are validated with data from the WAIS ice core, and some conclusions are drawn about the ice-divide stability throughout the Holocene.

Overall, this paper is nicely written and the authors do a commendable job in guiding the reader through the methods and results. However, in places I find the paper unnecessarily superficial and I don't see novel aspects clearly. I also suspect (but I am not certain) that parts of the azimuthal reconstruction may be erroneous leading to wrong inferences in terms of the ice-fabric orientation. Below, I mention a number of major comments/questions how this can be improved. Applications of radar polarimetry are still rare, and I hope that the points raised below will help to improve the next version of this paper.

Reinhard Drews, Tübingen University, Germany

**Clarify methodological advance**

It is stated that this study "...extends previous qualitative analyses [...] to obtain quantitative measurements.." (I. 285). Can you highlight more clearly what those extensions were compared to previous studies? From what I can see so far, this study nicely applies previous developments to a single new site, but I struggle to see the extensions. The link between the polarimetric phase gradient and icefabric parameters is based on the cited papers Fujita et al., 2006 and Jordan et al., 2019. Arguably matching the angular distance of co-polarization nodes with a 2D optimisation is new (I.233), but at least the dependency of this distance as a function of anisotropic scattering is already approximated in Fujita 2006. Also advantages or pitfalls (e.g., in terms of uniqueness and uncertainties involved) of this approach are not discussed.

I suppose that this paper is the first to explicitly focus on synthesising quad-polarimetric measurements for ApRES, although the related methodology is known from radar polarimetry textbooks (e.g., the cited Mott, 2006). The inferences drawn from this method about the "high angular" resolution are not credible as currently presented (see comment below). Also the lack of rotational dataset at this site makes it hard to discuss advantages/disadvantages of both approaches. I suggest a dedicated section were improvements and distinct differences compared to previous studies are highlighted more explicitly.

**Coincidental symmetry at $\theta = 90^{\circ}$ ?**

In Figs. 2b-e one principal axis of the ice-fabric appears at the local azimuthal angle  $\theta = 90^{\circ}$  (i.e., all panels have a reflectional or rotational symmetry around the  $\theta = 90^{\circ}$  axis). This means that during measurements antennas were coincidentally placed parallel (hh) and perpendicular (vv) to the (at the time) unknown ice-fabric orientation. It is possible that the operators in the field made

a conscious decision here because  $\theta = 90^{\circ}$  aligns with the strain rate (not the ice-flow) direction. However, given uncertainties involved in determining the direction of maximum strain rate and the antenna orientation, the  $\theta = 90^{\circ}$  symmetry almost seems too much of a coincidence. Based on our own experience with analysing quad-polarimetric data, we suggest that the authors double-check that indeed  $s_{hv} = s_{vh}$ . We found occasionally that  $s_{hv} = -s_{vh}$  without satisfying explanation as to why this can be the case (e.g., inconsistencies in labelling and naming of antenna orientations in the field?). However, if it is the case, then reconstruction of the ApRES signal using eq. (4) forces a symmetry axis at  $\theta = 90^{\circ}$  exemplified below for the  $s_{hh}$  component:

$$S_{11} = s_{hh}(\theta) = \underbrace{s_{hh}\cos^2\theta + s_{vv}\sin^2\theta}_{\text{symmetric at } \theta = 90^{\circ}} + \underbrace{(s_{vh} + s_{hv})\sin\theta\cos\theta}_{\text{anti-symmetric at } \theta = 90^{\circ}}_{\text{in general no symmetry axis at } \theta = 90^{\circ}} \underbrace{unless}_{s_{hv} = -s_{vh}}$$

The graphic below illustrates how this would be reflected in a full azimuthal reconstruction where the principal axis around  $\theta = 35,125^{\circ}$  in the top plot are erroneously mapped to  $\theta = 90,180^{\circ}$ . Without a co-polarized, rotational dataset this will occur unnoticed.

Maybe it will be helpful to investigate this further. Alternatively, state explicitly how the hh and vv directions were defined in the field, and why it makes sense that those axis align almost perfectly with the principal directions of the ice-fabric.

**Terminology linked to azimuthal resolution**

In numerous instances (e.g., I.7, I39, I49..) the authors advertise that synthesizing the azimuthal response from quad-polarimetric data (eq. 4) results in improved angular *resolution* compared to rotational setups. I disagree with that. The chosen azimuthal spacing of 1° (I. 397) is completely arbitrary and any value works with eq. 4. I agree that advantages and disadvantages of quad-polarimetric vs. rotational measurements should be discussed, but choosing an arbitrary gridding for  $\theta$  is not enough in this regard. Also, no rotational dataset is presented so that the claims about

the superiority of quad-polarimetric measurements are not rigorously substantiated (apart from the obvious fact that they are much quicker to obtain).

**Minor remarks**

- Abstract should state limitation that the methodology only works if one of the c-axis is pointing upwards.
- I 57: Eq. 4 *reconstructs* the azimuthal response, but this is something different than *resolving* it. See comments above.
- I 63: Specify what resolution you refer to. ApRES surely has lower potential for vertical resolution than ice-core data.
- I 65: Specify what the angular resolution is. It cannot be the 1°. I would also prefer more modest wording for "unambiguously". Defining the direction of the ApRES antennas alone is already error-inflicted and there is no rigorous statement in this paper on how this was done.
- Overall nice structure of the introduction. This works for me.
- Fig. 1a include orientation of the E-field vector. Statement that antennas are oriented parallel to the divide is conflicting with inference that principal axis is at  $\theta = 90^{\circ}$  (which is parallel to flow, which is oblique to the divide according to Fig 1b). See major comment 2.
- I 114 not only ice-dynamics but also ice properties induced through climate variations imprint on the ice-fabric evolution. I am not sure the principal ice-fabric axis always line up with today's strain rate regime as suggested here.
- I 127 the terminology "anisotropy" for  $\beta$  has confused me. You also need "anisotropy" for birefringence. Why not call it the "anisotropic reflection ratio" or something like that? "boundary reflection" is more appropriate than "boundary scattering" (the latter suggests some diffuse scattering which is not accounted for in this context. However, this may be a matter of taste.)
- I 137 remove abbreviation SISO. It is not used later on.
- I 150 include uncertainties for this angle here and elsewhere.
- I 169 double-check that  $s_{vh} = s_{hv}$  (see comment above).
- I understand how this azimuthal phase difference is calculated, but I don't understand what the additional value is. What inferences are drawn from the co-polarised phase difference that cannot be drawn from the hhvv phase angle?
- I 193 This first paragraph is more methods to me than results.
- As stated above the 90° are suspicious. Also, what are the  $\pm 7^{\circ}$  based on? I would think that errors in antenna positioning are larger.
- I 213 Typo? This ambiguity cannot be resolved in the hh power anomaly shown in Fig 2b. You need to use the polarity of the phase gradient.
- I 221 at least estimate these "human" errors
- why is the "anisotropy" an integer value?
- I 237 I think I missed something here: Aren't those node pairs simply depths where the phase shift between ordinary and extraordinary wave is odd integer multiple of  $\pi$ ? Clearly they will have a correspondence in the azimuthal phase difference (which is directly related to the phase angle). I don't understand the deeper physical implication of this 'four quadrant pattern yet.
- Fig .4 I appreciate the error bars on the ApRES derived  $E_2 E_1$ . Please state more clearly how those were derived.
- I 289 This is not the "best model" that matches observed results. It is a model that explains some of the features in the observations
- I 330 How fast does the ice-fabric structure adapt to a new strain regime? I think some sort of

statement in this regard is required to better justify statements of ice-divide stability.

---

## Author Comment (AC1) · 21 Dec 2020

**Author response to the review of Young et al. "Rapid and accurate polarimetric radar measurements of ice crystal fabric orientation at the Western Antarctic Ice Sheet (WAIS) Divide deep ice core site" [Manuscript # tc-2020-264]**

We would like to thank the two reviewers, Martin Rongen and Reinhard Drews, for their thorough, thoughtful and constructive reviews. Please find below the two reviewers' comments (RC) in **bold**, each followed by the authors' response (AC). Line, figure, and page numbers mentioned by the reviewers (within RC) refer to the original manuscript. The proposed changes to a future version of the manuscript are underlined. We note that we have not yet implemented these proposed changes and therefore the actual implementation of these Author Comments may vary somewhat from what is stated below.

**Review by Martin Rongen (30 October 2020)**

**General comments**

RC1.1. The authors present a quad-polarimetric radar measurement at WAIS Divide. The dataset is interesting in its innovative nature as well as in the quality of the derived results. The method follows in direct succession to earlier works developed by Shuji Fujita and Tom Jordan among others. While previous measurements required a manual rotation of the antenna system in small steps in order to measure the azimuthal variation associated with the birefringence signature, the quad-polarimetric measurement allows for signal at arbitrary azimuths to be deduced from just four antenna orientations. The results are validated by comparison to ice core data.

The overall presentation is detailed and rigorous. Some improvements may be made by giving a clearer structure to the results and discussion sections (see later specific comments). As a non-expert on glaciological radar measurements, the theory and methods section was challenging but the provided references proofed to be very helpful. For the paper to stand on its own some more context/details may be added (see specific comments). Some more discussion may also be added to section 5.3 (Methods comparisons and limitations). It currently gives a fair comparison between radar and ice core / sonic measurements but is short on the specific limitations and assumptions involved in generating data for arbitrary azimuth angles using the quad-polarimetric data.

AC1.1. Thank you for your detailed review, and you as well as Reinhard Drews have correctly identified that we need to further discuss some method comparisons and limitations. We hope that we've addressed your concerns in the specific comments section below.

**Specific comments**

RC1.2. The readability of the results section would be greatly improved by structuring it in sub-sections, as is also done for the other sections. A possible structure could be:

- 4.1 Experimental results from WAIS (up to line 224)

**- 4.2 Modelling the observed data (lines 224-255)**

- 4.3 Fabric asymmetry estimation (lines 255 ff)

AC1.2. Thank you for these suggestions. We will implement them in the revised manuscript.

**RC1.3. The meaning of the pad factor mentioned in line 193 is unclear.**

AC1.3. The pad factor refers to the amount of zero-padding applied to the time-domain signal. We apply zero-padding to our data with a pad factor of 2, as recommended by Brennan et al. (2014). While zero-padding is a common application in signal processing, we will make it explicit that the pad factor refers to the amount of zero padding equivalent to the total length of the signal.

**RC1.4. The details and reliability of the firn correction as introduced in line 198 are unclear. It is mentioned that this correction amplifies the estimated $E_2 - E_1$ values in the shallow ice and surprisingly large fabric asymmetries are then measured in that depth range. Thus the firn correction merits more attention (and maybe test without the correction) during the discussion (line 265 and 305) may be warranted.**

AC1.4. We implement firn correction as suggested in the Appendix of Jordan et al. (2020a). Because the equation that drives this correction was derived from founding ice-penetrating radar principles (e.g. the ice-air volume fractions in the mixing relations of Looyenga 1965), we believe these corrections to be physically representative of firn anisotropy. Notwithstanding, we agree with your belief that the fabric asymmetries in firn are surprisingly large. Therefore, we will: (i) make a new figure (tentatively Figure 5) that shows near-surface  $E_2 - E_1$  with and without firn correction; (ii) clarify why we apply firn correction in our revised manuscript (i.e. the reasoning given above); and (iii) attempt to address the physical origins and implications of high anisotropy in snow and firn, in particular, considering the effects of firn densification on crystal rotation (e.g. Burr et al. 2017) which, though the end member is a vertical pole, would therefore induce azimuthal anisotropy when considering the non-limiting case.

**RC1.5. In line 196 it may be worth mentioning that the Jordan et al. (2019) prescription to evaluate $d\varphi_{hhvv}/dz$ is not actually based on the phase plot itself but on the real and imaginary components of the coherence as given in equation 7b.**

AC1.5. We will implement this suggestion and clarify our methods in the revised manuscript.

**RC1.6. The reasons for and consequences of switching from an FIR filter to the method described in I.197ff remain unclear.**

AC1.6. We had used our method (a 2-D median filter + 2-D peak convolution) in a previous paper that processes 2-dimensional (x-z and y-z) imagery from a multistatic pRES array (Young et al. 2018). The reason for using this then was to remove high-frequency noise and

speckle. We have also visually checked the filtered image to ensure that there are no remaining "rapid phase excursions due to a modulo  $2\pi$  artefact" (Dall 2010). Therefore, we are confident that the goals of our method are comparable to those of Jordan et al. (2019). We are happy to apply the FIR filter in the revised manuscript if you still believe it to be necessary to do so.

**RC1.7. The anisotropy parameter $\beta$ seems to be missing a unit (dB?) in the caption of Figure 3.**

AC1.7. You are correct. We will correct this omission in the revised manuscript.

RC1.8. For depth greater than 1200 m (and to a lesser extent around 600 m) the derived  $E_2 - E_1$  values become rather unstable. While this is commented on and partially reflected in larger error bars, a population of outliers with small fabric asymmetries as well as small error bars is a bit worrying. It may be beneficial to show a plot of the coherence magnitude. Given vanishing magnitudes, the phase becomes unconstrained leading to erratic  $d\varphi_{hhvv}/dz$  values. In the deepest region the phase in Figure 2.e is more unstable as a function of depth than expected from the model calculation.

AC1.8. We agree with your suggestion to include  $|c_{hhvv}|$ . At the same time, because the antenna polarisation plane is aligned with the ice optic axis along all sites (see AC2.3 for the location of all ApRES measurement sites). Your hypothesis that the outliers in Figure 4 correspond to small  $|c_{hhvv}|$  is correct: beyond 1300 m, values of  $|c_{hhvv}|$  are generally <0.4. We propose to add a plot of  $|c_{hhvo}|$  in either Figure 2 or Figure 4 to constrain the validity of these outliers (the exact form to be determined upon resubmission). We will set an appropriate  $|c_{hhvv}|$  bound to use to filter outliers, which will change the visualisation of Fig. 2e and 4 somewhat.

RC1.9. The sentence "The birefringence of an individual crystal and its COF are related to the bulk ..." in line 97 reads a bit odd as a COF only applies to an ensemble of crystals. Maybe change to something like "the birefringence of individual crystals and their COF".

AC1.9. Your suggestion is good, and we will implement this in the revised manuscript.

**RC1.10. The term "depth step" in line 124 is a bit technical. Something like "depth where a reflection occurs" would be clearer to the reader.**

AC1.10. We will replace "... for each depth step and azimuthal orientation" with "... at each discrete scattering layer and azimuthal orientation", which is in line with the terminology used by Fujita et al. (2006) in their Section 2.2.

RC1.11. The meaning and relevance of the rotation matrix R as introduced in line 133 is unclear. To my understanding, it represents the rotation of the COF principle axis of *each traversed ice layer* with respect to a reference system defined by the antennas.

AC1.11. You are correct: the rotation matrix R is used in Equation (1) to reconstruct the theoretical signal components for each azimuthal shift  $\theta$ , for which the component in question is either T (transmission between the antennas and the scattering layer) or  $\Gamma$  (reflection at the scattering layer). The use of R in essence replicates an azimuthal-rotational experiment (acquisitions made at each rotational increment of the antenna acquisition plane). Your suggestion to clarify the use of R is valid, and we will summarise the above explanation and add this as a statement at the end of the paragraph in question.

**RC1.12. It is mentioned that $s_{hv}$ and $s_{vh}$ should be identical given an ideal measurement. Has the difference between these two orientations been studied and the potential impact quantified?**

AC1.12. In theory,  $s_{hv} = s_{vh}$  which is well known according to the Lorentz reciprocity theorem. In practice, there will be small differences including (but not limited to): (i) differences in the beam pattern between the transmitting and receiving aerials, (ii) random clutter within the transmitted media; and (iii) human error in antenna positioning. This topic is well studied (e.g. there is a 100+ page book (Stumpf 2017) purely dedicated to antenna signal reciprocity!) but in general, the main differences between  $s_{hv}$  and  $s_{vh}$  are due to human error during data collection. In general, our data has minimal difference between  $s_{hv}$  and  $s_{vh}$ , although there is some additional variability seen at the near surface (Figure AC1). This discrepancy may account for the mismatch between measured (Figure 2b,c,d) and modelled co-polarised results (Figure 3a,b,c) in the top ~200 m. We will modify Figure 2a (which already includes  $s_{uv}$  and  $s_{uv}$  to show how the Lorentz Reciprocity Theorem works in practice. We will also state this discrepancy in the results and mention our hypothesis in the discussion.

Additionally, sometimes  $s_{hv} = -s_{vh}$  due to the 180° ambiguity in antenna position. This is true in our case, and we explain why and the ramifications to the output polarisation plots (specifically Figure 2b and 2c) in AC2.3).

**Figure AC1**. Mean (polarisation-averaged) power return for each antenna orientation. Insets show magnification of power returns.

RC1.13. The COF orientation of each depth layer is resolved "by tracing the azimuthal minima in the cross-polarized power anomaly". While this may be a good approximation for this measurement, I wonder if this technique is generally applicable in the presence of strongly varying COF orientations. Assuming, for example, a constant angle of 20° in the top 500 m and a constant angle of 60° below, my understanding is that the minimum in the cross-polarized power anomaly would only slowly migrate towards 60° below 500 m as the bulk propagation is initially still dominated by the conditions above. One essentially measures the average COF orientation up to the scattering depth. But there may be a misunderstanding on my part here. A comment would be appreciated.

AC1.13. In general, this method (as well as other published methods so far, e.g. Brisbourne et al. 2019, Jordan et al. 2020a) is accurate if the COF orientation is depth invariant. If the orientation undergoes an abrupt switch, such as the example that you propose above and below 500 m, the minima in power anomaly would similarly undergo an abrupt switch (e.g. at ~215 m in Figure 4 of Brisbourne et al. 2019). If this happens, the 90° ambiguity in the cross-polarised power anomaly plot would need to be resolved by either the co-polarised power anomaly or the co-polarised phase difference plots (see AC2.19). If the orientation undergoes a slow rotation, the identified ice optic axis from the cross-polarised minima method is lagged from the true fabric orientation (e.g. Column 3, Figure 4c of Jordan et al. 2020c). The inaccuracy in identification of the ice optic axis will then result in corresponding over- or under-estimation of fabric asymmetry (e.g. Column 4, Figure 4c of Jordan et al. 2020c).

We will summarise the above paragraph in the Discussion (see AC2.2 of how exactly we will incorporate this within suggestions to include other relevant limitations).

**RC1.14. As noted in the general comments section 5.3 would benefit from a discussion of the specific limitations and assumptions involved in generating data for arbitrary azimuth angles using the quad-polarimetric data.**

AC1.14. See AC2.4 for a discussion on the comparisons between quad-polarimetric and azimuthal rotational experiments.

RC1.15. Section 5.1 paragraph 2 (lines 281-288) seems better suited in section 4.1 (results, modeling), here a Figure similar to Figure 5 in the Fujita 2006 paper may also be illustrative, showing that birefringence results in nodes in the power anomalies while anisotropic scattering results in a band structure, the spacing of which is a function of the scattering strength.

AC1.15. While a figure similar to Figure 5 of Fujita et al. (2006) may be useful, this specific figure has already been reproduced multiple times in subsequent papers (e.g. Figure 7 of Matsuoka et al. 2012, Figure 2 of Brisbourne et al. 2019) and we feel like there is already enough literature that describes the depth and azimuthal variability of the co-polarised nodes. Furthermore, we believe that Figure 3 already serves the same purpose, in particular, a comparison between Figure 3b and 3d which shows the controls of anisotropic scattering ( $\beta$ ) and birefringence ( $\epsilon(z)$ , through  $E_2$ - $E_1$ ). We will however add a sentence at L246 that

explicitly states that the azimuthal fabric asymmetry (as a representation of birefringence) is proportional to the depth-periodicity of co-polarised nodes. We will also move Section 5.1 Paragraph 2 (L281-288) to Section 4.1 which we agree is a good suggestion.

**Technical comments**

RC1.16. The link for the Mott, H. (2006) reference appears to be dead.

AC1.16. Thank you for pointing this out. We will fix the link.

RC1.17. In line 156, spectra so should be plural as are the amplitudes.

AC1.17. We will implement this correction in the revised manuscript.

RC1.18. In the title of subsection 5.1 it would be more consistent to refer to "anisotropic scattering" instead of the more ambiguous "anisotropy"

AC1.18. We agree with this suggestion and will implement this in the revised manuscript.

**Review by Reinhard Drews (19 November 2020)**

**Summary**

RC2.1. In their paper "Rapid and accurate polarimetric radar measurements of ice crystal fabric orientation at the Western Antarctic Ice Sheet (WAIS) Divide deep ice core site", Young and co-authors present an ApRES radar dataset, which they use to infer the ice-fabric characteristics continuously to a depth of 1500 m. Main results include quantification of the horizontal ice anisotropy with a depth invariant ice-fabric orientation that is aligned with the directions of the principal strain rates. The inferences are validated with data from the WAIS ice core, and some conclusions are drawn about the ice-divide stability throughout the Holocene.

Overall, this paper is nicely written and the authors do a commendable job in guiding the reader through the methods and results. However, in places I find the paper unnecessarily superficial and I don't see novel aspects clearly. I also suspect (but I am not certain) that parts of the azimuthal reconstruction may be erroneous leading to wrong inferences in terms of the ice-fabric orientation. Below, I mention a number of major comments/questions how this can be improved. Applications of radar polarimetry are still rare, and I hope that the points raised below will help to improve the next version of this paper.

AC2.1. Thank you for your detailed review, and we appreciate your honesty in that you have disclosed your potential conflict of interest. We believe your review is unbiased and correctly identifies several areas will need to be addressed, specifically the three major comments (RC2.2 - 2.4). We have attempted to alleviate your concerns in our responses below. Ultimately, we hope that our paper can contribute to the radar polarimetry literature and

**RC2.2. Clarify methodological advance**

It is stated that this study "..extends previous qualitative analyses [...] to obtain quantitative measurements.." (I. 285). Can you highlight more clearly what those extensions were compared to previous studies? From what I can see so far, this study nicely applies previous developments to a single new site, but I struggle to see the extensions. The link between the polarimetric phase gradient and ice- fabric parameters is based on the cited papers Fujita et al., 2006 and Jordan et al., 2019. Arguably matching the angular distance of co-polarization nodes with a 2D optimisation is new (I.233), but at least the dependency of this distance as a function of anisotropic scattering is already approximated in Fujita 2006. Also advantages or pitfalls (e.g., in terms of uniqueness and uncertainties involved) of this approach are not discussed.

I suppose that this paper is the first to explicitly focus on synthesising quad-polarimetric measurements for ApRES, although the related methodology is known from radar polarimetry textbooks (e.g., the cited Mott, 2006). The inferences drawn from this method about the "high angular" resolution are not credible as currently presented (see comment below). Also the lack of rotational dataset at this

**site makes it hard to discuss advantages/disadvantages of both approaches. I suggest a dedicated section were improvements and distinct differences compared to previous studies are highlighted more explicitly.**

AC2.2. In our opinion, this manuscript extends the literature (this being radar polarimetry for glaciological applications) through: (i) validation of co-polarised power anomaly and phase difference plots; (ii) publication of the quad-polarimetric reconstruction method; and (iii) direct validation of radar-derived measurements of ice fabric to that of ice cores at high depth resolution. We are fortunate that our results are simple to comprehend, and we believe that this, combined with the straightforward layout of the quad-polarimetric reconstruction method will help those who wish to apply radar polarimetric methods to glaciology. Due to the growing interest in radar polarimetry applications within the glaciological community as of late, we believe that our manuscript is not only valuable, but also timely for the above three reasons. In detail:

(i) is shown through the direct comparison between measured (Figure 2) and modelled (Figure 3) results, where the birefringence in the latter was induced using fabric eigenvalues from thin-section analysis of ice cores. (The anisotropic reflection ratio is instead estimated through matching the locations of the co-polarised nodes and the four-guadrant patterns with the measured results.) While we concede that the theory behind this comparison has previously been quantified in Fujita et al. (2006), they stop short of presenting modelled results complementary to their observations (in their case, fabric data from the Mizuho and Dome Fuji ice cores). Similarly, while fabric measurements from polarimetric radar measurements have been verified to ice core data by Jordan et al. (2019), they implement only the polarimetric coherence method (i.e. analysis of  $s_{hhw}$ ) and do not present any co- or cross-polarised datasets. On the other hand, Brisbourne et al. (2019) present co-polarised power anomaly and phase difference plots of radar measurements conducted at Korff Ice Rise but focus on the relative orientation of the antenna polarisation plane relative to the ice optic axis, and present only qualitative observations in terms of fabric strength. Our manuscript not only directly compares measured results to modelled plots, but also guantifies azimuthal fabric strength through the polarimetric coherence method and verifies results directly with ice core data. Therefore, our manuscript reconciles the methods of the above three studies.

Regarding (ii), we felt a need to include the complete equations that show how the received signal (both power in Equation 4 and phase in Equation 5) can be reconstructed from quad-polarimetric measurements. You are correct in that these methods are established, and we do not attempt to take credit for them. However, these methods are often presented within dense literature specifically directed towards radar engineering applications and may be daunting for the majority of glaciologists (myself included!). Rather, we attempt to introduce them to the glaciological literature in an approachable format so that they are easily accessible for researchers wishing to conduct quad-polarimetric experiments.

Regarding (iii), our results show that choosing a nominal depth-averaging window of 15 m (which is our nominal bulk-depth resolution) when applying the polarimetric coherence method (Equation 6) produces estimates of azimuthal fabric asymmetry

that closely match corresponding results from ice-core thin section data at similar resolutions (See AC2.7 regarding the specific use of the word 'resolution'). While Jordan et al. (2019) also use the polarimetric coherence method to match fabric asymmetry estimates between radar and ice core measurements, also showing comparative results, they had chosen to use a conservative nominal depth resolution of 100 m. Our study shows that this nominal depth resolution can be reduced down to levels at or exceeding that of the vertical spacing of ice core thin sections, while still producing comparable results.

As you may already know, there are multiple pitfalls in using the polarimetric coherence method, which are detailed in Jordan et al. (2019) and (2020a). We do not wish to regurgitate what has already been said, but we acknowledge that we have provided little caveats, which makes it seem like we are "overselling" our results. Therefore. we will separate Section 5.3 (Method comparisons and limitations) into two sections: (i) Method comparisons: and (ii) Advantages and limitations of the polarimetric coherence method. In the latter, we will summarise the pitfalls mentioned in Jordan et al. (2019) and (2020a), as well as other observations in our dataset. for example, the issue of symmetry at 90° if the wrong signs are used (see RC and AC2.3). We will also summarise the three advantages listed in the previous three paragraphs in this section.

**RC2.3. Coincidental symmetry at $\theta$ = 90°?**

In Figs. 2b-e one principal axis of the ice-fabric appears at the local azimuthal angle  $\theta$  = 90° (i.e., all panels have a reflectional or rotational symmetry around the  $\theta$  = 90° axis). This means that during measurements antennas were coincidentally placed parallel (*hh*) and perpendicular (*vv*) to the (at the time) unknown ice-fabric orientation. It is possible that the operators in the field made a conscious decision here because  $\theta$  = 90° aligns with the strain rate (not the ice-flow) direction. However, given uncertainties involved in determining the direction of maximum strain rate and the antenna orientation, the  $\theta$  = 90° symmetry almost seems too much of a coincidence. Based on our own experience with analysing quad-polarimetric data, we suggest that the authors double-check that indeed  $s_{hv} = s_{vh}$ . We found occasionally that  $s_{hv} = -s_{vh}$  without satisfying explanation as to why this can be the case (e.g., inconsistencies in labelling and naming of antenna orientations in the field?). However, if it is the case, then reconstruction of the ApRES signal using eq. (4) forces a symmetry axis at  $\theta$  = 90° exemplified below for the  $s_{hh}$  component:

$$S_{11} = s_{hh}(\theta) = \underbrace{s_{hh} \cos^2 \theta + s_{vv} \sin^2 \theta}_{\text{symmetric at } \theta = 90^{\circ}} + \underbrace{(s_{vh} + s_{hv}) \sin \theta \cos \theta}_{\text{anti-symmetric at } \theta = 90^{\circ}}$$

The graphic below illustrates how this would be reflected in a full azimuthal reconstruction where the principal axis around  $\theta$  = 35, 125° in the top plot are erroneously mapped to  $\theta$  = 90, 180°. Without a co-polarized, rotational dataset this will occur unnoticed.

---

## Author Comment (AC2) · 21 Dec 2020

**Author response to the review of Young et al. "Rapid and accurate polarimetric radar measurements of ice crystal fabric orientation at the Western Antarctic Ice Sheet (WAIS) Divide deep ice core site" [Manuscript # tc-2020-264]**

We would like to thank the two reviewers, Martin Rongen and Reinhard Drews, for their thorough, thoughtful and constructive reviews. Please find below the two reviewers' comments (RC) in **bold**, each followed by the authors' response (AC). Line, figure, and page numbers mentioned by the reviewers (within RC) refer to the original manuscript. The proposed changes to a future version of the manuscript are underlined. We note that we have not yet implemented these proposed changes and therefore the actual implementation of these Author Comments may vary somewhat from what is stated below.

**Review by Martin Rongen (30 October 2020)**

**General comments**

**RC1.1. The authors present a quad-polarimetric radar measurement at WAIS Divide. The dataset is interesting in its innovative nature as well as in the quality of the derived results. The method follows in direct succession to earlier works developed by Shuji Fujita and Tom Jordan among others. While previous measurements required a manual rotation of the antenna system in small steps in order to measure the azimuthal variation associated with the birefringence signature, the quad-polarimetric measurement allows for signal at arbitrary azimuths to be deduced from just four antenna orientations. The results are validated by comparison to ice core data.**

**The overall presentation is detailed and rigorous. Some improvements may be made by giving a clearer structure to the results and discussion sections (see later specific comments). As a non-expert on glaciological radar measurements, the theory and methods section was challenging but the provided references proofed to be very helpful. For the paper to stand on its own some more context/details may be added (see specific comments). Some more discussion may also be added to section 5.3 (Methods comparisons and limitations). It currently gives a fair comparison between radar and ice core / sonic measurements but is short on the specific limitations and assumptions involved in generating data for arbitrary azimuth angles using the quad-polarimetric data.**

*AC1.1. Thank you for your detailed review, and you as well as Reinhard Drews have correctly identified that we need to further discuss some method comparisons and limitations. We hope that we've addressed your concerns in the specific comments section below.*

**Specific comments**

**RC1.2. The readability of the results section would be greatly improved by structuring it in sub-sections, as is also done for the other sections. A possible structure could be:**
      **– 4.1 Experimental results from WAIS (up to line 224)**
      **– 4.2 Modelling the observed data (lines 224-255)**
      **– 4.3 Fabric asymmetry estimation (lines 255 ff)**

AC1.2. Thank you for these suggestions. We will implement them in the revised manuscript.

**RC1.3. The meaning of the pad factor mentioned in line 193 is unclear.**

AC1.3. The pad factor refers to the amount of zero-padding applied to the time-domain signal. We apply zero-padding to our data with a pad factor of 2, as recommended by Brennan et al. (2014). While zero-padding is a common application in signal processing, we will make it explicit that the pad factor refers to the amount of zero padding equivalent to the total length of the signal.

**RC1.4. The details and reliability of the firn correction as introduced in line 198 are unclear. It is mentioned that this correction amplifies the estimated $E_2 - E_1$ values in the shallow ice and surprisingly large fabric asymmetries are then measured in that depth range. Thus the firn correction merits more attention (and maybe test without the correction) during the discussion (line 265 and 305) may be warranted.**

AC1.4. We implement firn correction as suggested in the Appendix of Jordan et al. (2020a). Because the equation that drives this correction was derived from founding ice-penetrating radar principles (e.g. the ice-air volume fractions in the mixing relations of Looyenga 1965), we believe these corrections to be physically representative of firn anisotropy. Notwithstanding, we agree with your belief that the fabric asymmetries in firn are surprisingly large. Therefore, we will: (i) make a new figure (tentatively Figure 5) that shows near-surface $E_2 - E_1$ with and without firn correction; (ii) clarify why we apply firn correction in our revised manuscript (i.e. the reasoning given above); and (iii) attempt to address the physical origins and implications of high anisotropy in snow and firn, in particular, considering the effects of firn densification on crystal rotation (e.g. Burr et al. 2017) which, though the end member is a vertical pole, would therefore induce azimuthal anisotropy when considering the non-limiting case.

**RC1.5. In line 196 it may be worth mentioning that the Jordan et al. (2019) prescription to evaluate $d\varphi_{hhvv}/dz$ is not actually based on the phase plot itself but on the real and imaginary components of the coherence as given in equation 7b.**

AC1.5. We will implement this suggestion and clarify our methods in the revised manuscript.

**RC1.6. The reasons for and consequences of switching from an FIR filter to the method described in l.197ff remain unclear.**

AC1.6. We had used our method (a 2-D median filter + 2-D peak convolution) in a previous paper that processes 2-dimensional (x-z and y-z) imagery from a multistatic pRES array (Young et al. 2018). The reason for using this then was to remove high-frequency noise and

speckle. We have also visually checked the filtered image to ensure that there are no remaining "rapid phase excursions due to a modulo $2\pi$ artefact" (Dall 2010). Therefore, we are confident that the goals of our method are comparable to those of Jordan et al. (2019). We are happy to apply the FIR filter in the revised manuscript if you still believe it to be necessary to do so.

**RC1.7. The anisotropy parameter $\beta$ seems to be missing a unit (dB?) in the caption of Figure 3.**

AC1.7. You are correct. We will correct this omission in the revised manuscript.

**RC1.8. For depth greater than 1200 m (and to a lesser extent around 600 m) the derived $E_2 - E_1$ values become rather unstable. While this is commented on and partially reflected in larger error bars, a population of outliers with small fabric asymmetries as well as small error bars is a bit worrying. It may be beneficial to show a plot of the coherence magnitude. Given vanishing magnitudes, the phase becomes unconstrained leading to erratic $d\varphi_{hhvv}/dz$ values. In the deepest region the phase in Figure 2.e is more unstable as a function of depth than expected from the model calculation.**

AC1.8. We agree with your suggestion to include $|c_{hhvv}|$. At the same time, because the antenna polarisation plane is aligned with the ice optic axis along all sites (see AC2.3 for the location of all ApRES measurement sites). Your hypothesis that the outliers in Figure 4 correspond to small $|c_{hhvv}|$ is correct: beyond 1300 m, values of $|c_{hhvv}|$ are generally <0.4. We propose to add a plot of $|c_{hhvv}|$ in either Figure 2 or Figure 4 to constrain the validity of these outliers (the exact form to be determined upon resubmission). We will set an appropriate $|c_{hhvv}|$ bound to use to filter outliers, which will change the visualisation of Fig. 2e and 4 somewhat.

**RC1.9. The sentence "The birefringence of an individual crystal and its COF are related to the bulk ..." in line 97 reads a bit odd as a COF only applies to an ensemble of crystals. Maybe change to something like "the birefringence of individual crystals and their COF".**

AC1.9. Your suggestion is good, and we will implement this in the revised manuscript.

**RC1.10. The term "depth step" in line 124 is a bit technical. Something like "depth where a reflection occurs" would be clearer to the reader.**

AC1.10. We will replace "... for each depth step and azimuthal orientation" with "... at each discrete scattering layer and azimuthal orientation", which is in line with the terminology used by Fujita et al. (2006) in their Section 2.2.

**RC1.11. The meaning and relevance of the rotation matrix $R$ as introduced in line 133 is unclear. To my understanding, it represents the rotation of the COF principle axis of each traversed ice layer with respect to a reference system defined by the antennas.**

AC1.11. You are correct: the rotation matrix $R$ is used in Equation (1) to reconstruct the theoretical signal components for each azimuthal shift $\theta$, for which the component in question is either $T$ (transmission between the antennas and the scattering layer) or $\Gamma$ (reflection at the scattering layer). The use of $R$ in essence replicates an azimuthal-rotational experiment (acquisitions made at each rotational increment of the antenna acquisition plane). Your suggestion to clarify the use of $R$ is valid, and we will summarise the above explanation and add this as a statement at the end of the paragraph in question.

**RC1.12. It is mentioned that $s_{hv}$ and $s_{vh}$ should be identical given an ideal measurement. Has the difference between these two orientations been studied and the potential impact quantified?**

AC1.12. In theory, $s_{hv} = s_{vh}$ which is well known according to the Lorentz reciprocity theorem. In practice, there will be small differences including (but not limited to): (i) differences in the beam pattern between the transmitting and receiving aerials, (ii) random clutter within the transmitted media; and (iii) human error in antenna positioning. This topic is well studied (e.g. there is a 100+ page book (Stumpf 2017) purely dedicated to antenna signal reciprocity!) but in general, the main differences between $s_{hv}$ and $s_{vh}$ are due to human error during data collection. In general, our data has minimal difference between $s_{hv}$ and $s_{vh}$, although there is some additional variability seen at the near surface (Figure AC1). This discrepancy may account for the mismatch between measured (Figure 2b,c,d) and modelled co-polarised results (Figure 3a,b,c) in the top ~200 m. We will modify Figure 2a (which already includes $s_{vh}$ and $s_{vh}$ incorrectly labelled as $e_{vh}$, $e_{vh}$, which we will correct in our revised manuscript) to also include $s_{hv}$ and $s_{vv}$ to show how the Lorentz Reciprocity Theorem works in practice. We will also state this discrepancy in the results and mention our hypothesis in the discussion.

Additionally, sometimes $s_{hv} = -s_{vh}$ due to the 180° ambiguity in antenna position. This is true in our case, and we explain why and the ramifications to the output polarisation plots (specifically Figure 2b and 2c) in AC2.3).

[Figure]

**Figure AC1**. Mean (polarisation-averaged) power return for each antenna orientation. Insets show magnification of power returns.

**RC1.13. The COF orientation of each depth layer is resolved "by tracing the azimuthal minima in the cross-polarized power anomaly". While this may be a good approximation for this measurement, I wonder if this technique is generally applicable in the presence of strongly varying COF orientations. Assuming, for example, a constant angle of 20° in the top 500 m and a constant angle of 60° below, my understanding is that the minimum in the cross-polarized power anomaly would only slowly migrate towards 60° below 500 m as the bulk propagation is initially still dominated by the conditions above. One essentially measures the average COF orientation up to the scattering depth. But there may be a misunderstanding on my part here. A comment would be appreciated.**

AC1.13. In general, this method (as well as other published methods so far, e.g. Brisbourne et al. 2019, Jordan et al. 2020a) is accurate if the COF orientation is depth invariant. If the orientation undergoes an abrupt switch, such as the example that you propose above and below 500 m, the minima in power anomaly would similarly undergo an abrupt switch (e.g. at ~215 m in Figure 4 of Brisbourne et al. 2019). If this happens, the 90° ambiguity in the cross-polarised power anomaly plot would need to be resolved by either the co-polarised power anomaly or the co-polarised phase difference plots (see AC2.19). If the orientation undergoes a slow rotation, the identified ice optic axis from the cross-polarised minima method is lagged from the true fabric orientation (e.g. Column 3, Figure 4c of Jordan et al. 2020c). The inaccuracy in identification of the ice optic axis will then result in corresponding over- or under-estimation of fabric asymmetry (e.g. Column 4, Figure 4c of Jordan et al. 2020c).

We will summarise the above paragraph in the Discussion (see AC2.2 of how exactly we will incorporate this within suggestions to include other relevant limitations).

**RC1.14. As noted in the general comments section 5.3 would benefit from a discussion of the specific limitations and assumptions involved in generating data for arbitrary azimuth angles using the quad-polarimetric data.**

AC1.14. See AC2.4 for a discussion on the comparisons between quad-polarimetric and azimuthal rotational experiments.

**RC1.15. Section 5.1 paragraph 2 (lines 281-288) seems better suited in section 4.1 (results, modeling), here a Figure similar to Figure 5 in the Fujita 2006 paper may also be illustrative, showing that birefringence results in nodes in the power anomalies while anisotropic scattering results in a band structure, the spacing of which is a function of the scattering strength.**

AC1.15. While a figure similar to Figure 5 of Fujita et al. (2006) may be useful, this specific figure has already been reproduced multiple times in subsequent papers (e.g. Figure 7 of Matsuoka et al. 2012, Figure 2 of Brisbourne et al. 2019) and we feel like there is already enough literature that describes the depth and azimuthal variability of the co-polarised nodes. Furthermore, we believe that Figure 3 already serves the same purpose, in particular, a comparison between Figure 3b and 3d which shows the controls of anisotropic scattering ($\beta$) and birefringence ($\varepsilon(z)$, through $E_2$-$E_1$). We will however add a sentence at L246 that

explicitly states that the azimuthal fabric asymmetry (as a representation of birefringence) is proportional to the depth-periodicity of co-polarised nodes. We will also move Section 5.1 Paragraph 2 (L281-288) to Section 4.1 which we agree is a good suggestion.

**Technical comments**

**RC1.16. The link for the Mott, H. (2006) reference appears to be dead.**

AC1.16. Thank you for pointing this out. We will fix the link.

**RC1.17. In line 156, spectra so should be plural as are the amplitudes.**

AC1.17. We will implement this correction in the revised manuscript.

**RC1.18. In the title of subsection 5.1 it would be more consistent to refer to "anisotropic scattering" instead of the more ambiguous "anisotropy"**

AC1.18. We agree with this suggestion and will implement this in the revised manuscript.

**Review by Reinhard Drews (19 November 2020)**

**Summary**

RC2.1. In their paper "Rapid and accurate polarimetric radar measurements of ice crystal fabric orientation at the Western Antarctic Ice Sheet (WAIS) Divide deep ice core site", Young and co-authors present an ApRES radar dataset, which they use to infer the ice-fabric characteristics continuously to a depth of 1500 m. Main results include quantification of the horizontal ice anisotropy with a depth invariant ice-fabric orientation that is aligned with the directions of the principal strain rates. The inferences are validated with data from the WAIS ice core, and some conclusions are drawn about the ice-divide stability throughout the Holocene.

Overall, this paper is nicely written and the authors do a commendable job in guiding the reader through the methods and results. However, in places I find the paper unnecessarily superficial and I don't see novel aspects clearly. I also suspect (but I am not certain) that parts of the azimuthal reconstruction may be erroneous leading to wrong inferences in terms of the ice-fabric orientation. Below, I mention a number of major comments/questions how this can be improved. Applications of radar polarimetry are still rare, and I hope that the points raised below will help to improve the next version of this paper.

*AC2.1. Thank you for your detailed review, and we appreciate your honesty in that you have disclosed your potential conflict of interest. We believe your review is unbiased and correctly identifies several areas will need to be addressed, specifically the three major comments (RC2.2 - 2.4). We have attempted to alleviate your concerns in our responses below. Ultimately, we hope that our paper can contribute to the radar polarimetry literature and*

**RC2.2. Clarify methodological advance**

It is stated that this study "..extends previous qualitative analyses [...] to obtain quantitative measurements.." (l. 285). Can you highlight more clearly what those extensions were compared to previous studies? From what I can see so far, this study nicely applies previous developments to a single new site, but I struggle to see the extensions. The link between the polarimetric phase gradient and ice- fabric parameters is based on the cited papers Fujita et al., 2006 and Jordan et al., 2019. Arguably matching the angular distance of co-polarization nodes with a 2D optimisation is new (l.233), but at least the dependency of this distance as a function of anisotropic scattering is already approximated in Fujita 2006. Also advantages or pitfalls (e.g., in terms of uniqueness and uncertainties involved) of this approach are not discussed.

I suppose that this paper is the first to explicitly focus on synthesising quad-polarimetric measurements for ApRES, although the related methodology is known from radar polarimetry textbooks (e.g., the cited Mott, 2006). The inferences drawn from this method about the "high angular" resolution are not credible as currently presented (see comment below). Also the lack of rotational dataset at this

**site makes it hard to discuss advantages/disadvantages of both approaches. I suggest a dedicated section were improvements and distinct differences compared to previous studies are highlighted more explicitly.**

AC2.2. In our opinion, this manuscript extends the literature (this being radar polarimetry for glaciological applications) through: (i) validation of co-polarised power anomaly and phase difference plots; (ii) publication of the quad-polarimetric reconstruction method; and (iii) direct validation of radar-derived measurements of ice fabric to that of ice cores at high depth resolution. We are fortunate that our results are simple to comprehend, and we believe that this, combined with the straightforward layout of the quad-polarimetric reconstruction method will help those who wish to apply radar polarimetric methods to glaciology. Due to the growing interest in radar polarimetry applications within the glaciological community as of late, we believe that our manuscript is not only valuable, but also timely for the above three reasons. In detail:

> (i) is shown through the direct comparison between measured (Figure 2) and modelled (Figure 3) results, where the birefringence in the latter was induced using fabric eigenvalues from thin-section analysis of ice cores. (The anisotropic reflection ratio is instead estimated through matching the locations of the co-polarised nodes and the four-quadrant patterns with the measured results.) While we concede that the theory behind this comparison has previously been quantified in Fujita et al. (2006), they stop short of presenting modelled results complementary to their observations (in their case, fabric data from the Mizuho and Dome Fuji ice cores). Similarly, while fabric measurements from polarimetric radar measurements have been verified to ice core data by Jordan et al. (2019), they implement only the polarimetric coherence method (i.e. analysis of $s_{hhvv}$) and do not present any co- or cross-polarised datasets. On the other hand, Brisbourne et al. (2019) present co-polarised power anomaly and phase difference plots of radar measurements conducted at Korff Ice Rise but focus on the relative orientation of the antenna polarisation plane relative to the ice optic axis, and present only qualitative observations in terms of fabric strength. Our manuscript not only directly compares measured results to modelled plots, but also quantifies azimuthal fabric strength through the polarimetric coherence method and verifies results directly with ice core data. Therefore, our manuscript reconciles the methods of the above three studies.

> Regarding (ii), we felt a need to include the complete equations that show how the received signal (both power in Equation 4 and phase in Equation 5) can be reconstructed from quad-polarimetric measurements. You are correct in that these methods are established, and we do not attempt to take credit for them. However, these methods are often presented within dense literature specifically directed towards radar engineering applications and may be daunting for the majority of glaciologists (myself included!). Rather, we attempt to introduce them to the glaciological literature in an approachable format so that they are easily accessible for researchers wishing to conduct quad-polarimetric experiments.

> Regarding (iii), our results show that choosing a nominal depth-averaging window of 15 m (which is our nominal bulk-depth resolution) when applying the polarimetric coherence method (Equation 6) produces estimates of azimuthal fabric asymmetry

that closely match corresponding results from ice-core thin section data at similar resolutions (See AC2.7 regarding the specific use of the word 'resolution'). While Jordan et al. (2019) also use the polarimetric coherence method to match fabric asymmetry estimates between radar and ice core measurements, also showing comparative results, they had chosen to use a conservative nominal depth resolution of 100 m. Our study shows that this nominal depth resolution can be reduced down to levels at or exceeding that of the vertical spacing of ice core thin sections, while still producing comparable results.

As you may already know, there are multiple pitfalls in using the polarimetric coherence method, which are detailed in Jordan et al. (2019) and (2020a). We do not wish to regurgitate what has already been said, but we acknowledge that we have provided little caveats, which makes it seem like we are "overselling" our results. Therefore, we will separate Section 5.3 (Method comparisons and limitations) into two sections: (i) Method comparisons; and (ii) Advantages and limitations of the polarimetric coherence method. In the latter, we will summarise the pitfalls mentioned in Jordan et al. (2019) and (2020a), as well as other observations in our dataset, for example, the issue of symmetry at 90° if the wrong signs are used (see RC and AC2.3). We will also summarise the three advantages listed in the previous three paragraphs in this section.

**RC2.3. Coincidental symmetry at $\theta$ = 90°?**

**In Figs. 2b-e one principal axis of the ice-fabric appears at the local azimuthal angle $\theta$ = 90° (i.e., all panels have a reflectional or rotational symmetry around the $\theta$ = 90° axis). This means that during measurements antennas were coincidentally placed parallel ($hh$) and perpendicular ($vv$) to the (at the time) unknown ice-fabric orientation. It is possible that the operators in the field made a conscious decision here because $\theta$ = 90° aligns with the strain rate (not the ice-flow) direction. However, given uncertainties involved in determining the direction of maximum strain rate and the antenna orientation, the $\theta$ = 90° symmetry almost seems too much of a coincidence. Based on our own experience with analysing quad-polarimetric data, we suggest that the authors double-check that indeed $s_{hv} = s_{vh}$. We found occasionally that $s_{hv} = -s_{vh}$ without satisfying explanation as to why this can be the case (e.g., inconsistencies in labelling and naming of antenna orientations in the field?). However, if it is the case, then reconstruction of the ApRES signal using eq. (4) forces a symmetry axis at $\theta$ = 90° exemplified below for the $s_{hh}$ component:**

$$S_{11} = s_{hh}(\theta) = \underbrace{s_{hh}\cos^2\theta + s_{vv}\sin^2\theta}_{\substack{\text{symmetric at} \quad \theta=90°}} + \underbrace{(s_{vh} + s_{hv})\sin\theta\cos\theta}_{\substack{\text{anti-symmetric at} \quad \theta=90°}}$$
$$\underbrace{\qquad\qquad\qquad\qquad\qquad\qquad\qquad\qquad\qquad\qquad}_{\text{in general no symmetry axis at} \quad \theta=90° \quad \text{unless} \quad s_{hv}=-s_{vh}}$$

**The graphic below illustrates how this would be reflected in a full azimuthal reconstruction where the principal axis around $\theta$ = 35, 125° in the top plot are erroneously mapped to $\theta$ = 90, 180°. Without a co-polarized, rotational dataset this will occur unnoticed.**

[Figure]

synthetic 2 layer
fantasy froward model

"wrong", but very symmetric
reconstruction

**Maybe it will be helpful to investigate this further. Alternatively, state explicitly how the *hh* and *vv* directions were defined in the field, and why it makes sense that those axis align almost perfectly with the principal directions of the ice-fabric.**

AC2.3. Your suspicions in the exact alignment of the antenna polarimetry plane in Fig. 2 with the strain axes is warranted. In our initial manuscript submission, we show the results from one ApRES site, but we did not mention in our manuscript that we had also obtained ApRES measurements from nine other sites along a ~6 km-long transect. The results displayed in Fig. 2 represents Site I, and is one of 10 total sites in terms of their relative position along the transect. The antenna axis (i.e. the plane that runs between the transmitting and receiving antenna) is orthogonal to the transect line, which is shown in Figure (AC2a).

[Figure]

**Figure AC2**. (a) Map of local surface (grey contours) and basal topography (background colour) in the WAIS Divide area. Map is equivalent to Figure 1 with the addition of the ApRES transect. The red dot shows Site I where the data that produced Figure 2 was collected. (b) Surface and bed topography along the ApRES transect showing the relative locations of the 10 ApRES measurement sites.

Surface topography was obtained from REMA (Howat et al. 2019) and bed topography from
BedMachine Antarctica v1 (Morlighem et al. 2020).

Upon re-inspection of our code, we have noticed a rounding error when importing the
Northing and Easting coordinates that resulted in the antenna polarisation plane to be
aligned exactly with the direction of maximum compression. In reality, the antenna
polarisation plane is actually +7° (where positive numbers represent counterclockwise
directions) from the strain compressive axis (rounded to the nearest degree).

Using data collected at Site I (the data used to produce Figure 2 in the initial TCD
manuscript), we present in Figure (AC3) 2 sets of plots of co- and cross-polarised power
anomaly and phase difference. Set 1 (panels a) assumes that $s_{hv} = s_{vh}$, and set 2 (panels b)
assumes $s_{hv} = -s_{vh}$. In these figures, the mean and standard error of the principal axis for
panels (a) were calculated to be 89.7° ± 0.05°, and for panels (b), 90.3° ± 0.09°, using the
same methods described in the TCD manuscript. Figure (AC3) shows that the principal axis
is indeed oriented at 90° (given appropriate rounding of significant figures).

However, because the two results give approximately the same results, we are not able to
determine whether or not panels (a) or (b) shows the correct polarimetric patterns from
Figure (AC3) alone. We then run the same investigation on the other 9 sites and show the
output plots from one of these sites (Site C) in Figure (AC4). We regard the results shown in
Figure (AC4) to have higher variability between results produced using each of the two
assumptions in turn, but overall representative of all sites. We note three differences in
Figure (AC4) that are not immediately obvious in Figure (AC3). First: for all sites, the first
assumption that $s_{hv} = s_{vh}$ will always produce a symmetric co-polarised power anomaly and
phase difference plot about $\theta = 90°$. Second: we note that the principal axis inferred solely
from the co-polarised plots using this first assumption is misaligned with the principal axis as
determined from the cross-polarised power anomaly. Third: the principal axes in all plots
produced under the second assumption that $s_{hv} = -s_{vh}$ are all consistent with each other. For
the example site (Site C) shown in Figure (AC4), the plots produce a principal axis oriented
at 81.1° ± 0.07° under the first assumption ($s_{hv} = s_{vh}$, Figure AC4a) and 101.8° ± 0.09° under
the second assumption ($s_{hv} = -s_{vh}$, Figure AC4b). We assume that the deviation of the
principal axis from exactly 90° in both assumptions is the result of human errors in antenna
positioning. The reflectional symmetry in the orientation of the principal axis (i.e. ~ -10° from
90° with the first assumption and ~ +10° from 90° with the second assumption at Site C) is
seen at all sites, which reflects the importance of knowing whether an increase in azimuth in
the polarimetry plots corresponds to a clockwise or anticlockwise rotation in antenna
orientation.

Assuming that the four orthogonal combinations of antenna orientations were conducted in
the same way for all 10 sites (which, to the best of our knowledge, is true), the results from
Figure (AC3) and Figure (AC4) show that (i) the positions of [$h$ | $v$] and [$v$ | $h$] that we used in
the field (Figure 1a) correspond to the second assumption that $s_{hv} = -s_{vh}$; (ii) the use of the
cross-polarised power anomaly in determining the principal axis is valid regardless of
whether the first or second assumption is used; and (iii) the principal axis for Site I at 90° and
aligned exactly with the strain axes is valid as well as being fortuitous. At the same time, you
will notice that Figure (AC3a1-3) are the mirror images to Figure (2b-d) in the TCD

manuscript. This is because we had implemented the first assumption ($s_{hv} = s_{vh}$) upon submitting to TCD, where positive differences between the ice optic axis and the antenna polarisation plane results in clockwise rotation. This error was initially not obvious because of the apparent symmetry of the *hh* and *hv* plots about 90°. After discussion with co-authors, we modified the code so that positive differences now reflect counterclockwise rotation, which is in line with the convention used for polar coordinates. We apologise for the confusion caused.

Therefore, given the new knowledge gleaned from this investigation, we will implement the following in our revised manuscript draft: (a) update our estimate of relative orientation of the ice optic axis from 0 to +2° (and a statement that we have rounded to the nearest degree): (b) re-process our datasets using the assumption that $s_{hv} = -s_{vh}$: (c) provide equivalent plots of Figure 2 for the other 9 sites in Supporting Information to show consistency across the entire transect: (d) provide an explanation in Section 3.3 as to why the principal axis is aligned almost exactly with the strain compression axis; and (e) provide a statement in Section 3.4 that details the importance of checking whether $s_{hv} = s_{vh}$ or $s_{hv} = -s_{vh}$. We very much thank you for pointing out to check the assumptions of symmetric reciprocity; otherwise, we would have published erroneous results!

[Figure]

**Figure AC3**. ApRES polarimetric power anomaly and phase difference measured at Site I, WAIS Divide (Figure AC2). The top row of panels (a) were produced assuming $s_{hv} = s_{vh}$, and the bottom row of panels (b) assume $s_{hv} = -s_{vh}$. Columns (1) show co-polarised power anomaly, columns (2) cross-polarised power anomaly, columns (3) co-polarised phase difference, and columns (4) cross-polarised phase difference. Bright green dots represent azimuthal minima at each range bin, and the dark green line is the best estimate of the symmetry axis calculated using a

Gaussian-weighted moving average of the azimuthal minima. Depths with insufficient SNR are greyed out.

[Figure]

**Figure AC4**. ApRES polarimetric power anomaly and phase difference measured at Site C, WAIS Divide. Legends are identical with those for Figure AC3.

**RC2.4. Terminology linked to azimuthal resolution**

**In numerous instances (e.g., l.7, l39, l49.. ) the authors advertise that synthesizing the azimuthal response from quad-polarimetric data (eq. 4) results in improved angular resolution compared to rotational setups. I disagree with that. The chosen azimuthal spacing of 1° (l. 397) is completely arbitrary and any value works with eq. 4. I agree that advantages and disadvantages of quad- polarimetric vs. rotational measurements should be discussed, but choosing an arbitrary gridding for θ is not enough in this regard. Also, no rotational dataset is presented so that the claims about the superiority of quad-polarimetric measurements are not rigorously substantiated (apart from the obvious fact that they are much quicker to obtain).**

AC2.4. Unfortunately we did not conduct azimuthal rotational measurements at WAIS Divide and so we are unable to make a full and direct comparison between quad-polarimetric and rotational measurements specifically with regards to our own datasets at WAIS Divide. However, comparisons between the two field methods were made when analysing ApRES data from Brisbourne et al. (2019) of which we found minimal difference in the results generated from the two methods.

That being said, Matsuoka et al. (2012) conducted an azimuthal rotational survey close to (~5 km from) our study site of which the results are shown in their Figure 3, Panel SW-24. Although they had used radar systems with different frequencies from ours (60 and 179 MHz as opposed to 300 MHz for ApRES), we can see that, at least to 1400 m, our estimate of the fabric orientation matches their results.

We acknowledge that we cannot fully claim the superiority one measurement method over the other without conducting both at the same study site, and so we will mention the caveat in that, because we did not conduct azimuthal rotational measurements, we are unable to make a full and direct comparison between the two methods. Because of this caveat, we will drop the claims of azimuthal resolution superiority from using quad-pol measurements. We will then qualitatively compare our results to that of Matsuoka et al. (2012) to show that, visually, the results are similar. We will also mention that from analysis of experiments conducted at Korff Ice Rise, we observe that the produced results are qualitatively similar (C. Martin, unpublished data).

**Minor remarks**

**RC2.5. Abstract should state limitation that the methodology only works if one of the c-axis is pointing upwards.**

AC2.5. That is a fair point and we will make this limitation explicit in the abstract.

**RC2.6. l 57: Eq. 4 reconstructs the azimuthal response, but this is something different than resolving it. See comments above.**

AC2.6. See AC2.4.

**RC2.7. l 63: Specify what resolution you refer to. ApRES surely has lower potential for vertical resolution than ice-core data.**

AC2.7. ApRES certainly has lower resolution potential than ice-core data and we apologise for not being specific. The issue with defining the vertical resolution of ice cores is due to allocation of core sections to different scientific goals (at least, to the best of my knowledge), so while the depth resolution of ice cores in terms of fabric eigenvalue estimates can potentially be down to the centimeter scale, in practice the thin sections used to estimate eigenvalues are taken at larger intervals (~50-100m for the WAIS Divide ice core). The depth resolution of ApRES refers to the bulk-averaged fabric of the local depth window (15 m in our case). We propose to distinguish the two by using the term "bulk-averaged depth resolution" for the vertical resolution of ApRES measurements of ice fabric, and "depth separation" for the corresponding measurements of ice-core thin sections. Following the terminology of Jordan et al. (2019), we will modify this sentence to state: "... at a nominal bulk-averaged depth resolution of 15 m. We show that, using this setup and method, our estimates of fabric asymmetry are comparable to that from ice cores taken at similar depth intervals. From our results, we determine…".

**RC2.8. l 65: Specify what the angular resolution is. It cannot be the 1° . I would also prefer more modest wording for "unambiguously". Defining the direction of the ApRES antennas alone is already error-inflicted and there is no rigorous statement in this paper on how this was done.**

AC2.8. We will replace "unambiguously identify" with "explicitly determine". Because there is ambiguity in the attribution of angular resolution to either the model azimuthal bin size (1°) or the accuracy of orienting the antennas, we choose to instead remove "...to high angular resolution" from this sentence.

**RC2.9. Overall nice structure of the introduction. This works for me.**

AC2.9. Thanks for the compliment!

**RC2.10. Fig. 1a include orientation of the E-field vector. Statement that antennas are oriented parallel to the divide is conflicting with inference that principal axis is at θ = 90° (which is parallel to flow, which is oblique to the divide according to Fig 1b). See major comment 2.**

AC2.10. We will include the orientation of the E-field vector in Figure 1 in the revised manuscript. The angle stated here (250°) is actually the orientation of the velocity vectors and should instead state 268°. We will correct this in the future manuscript, and calculate uncertainties for this angle from human error (see RC2.18 for how we will calculate this).

**RC2.11. l 114 not only ice-dynamics but also ice properties induced through climate variations imprint on the ice-fabric evolution. I am not sure the principal ice-fabric axis always line up with today's strain rate regime as suggested here.**

AC2.11. We acknowledge the effect of climate variations on climate history, however, although we have some idea of how climate can perturb the magnitude of fabric from Kennedy et al. (2013), its influence in influencing its orientation remains unclear (to the best of my knowledge). Therefore, we respectfully choose to not take your suggestion on including the role of climate perturbation in this statement. We will however reduce the certainty of this sentence, which will now read: "The direction of the greatest horizontal c-axis concentration in general reflects the orientation of horizontal strain extension and corresponds with $E_2$ in our notation.

**RC2.12. l 127 the terminology "anisotropy" for β has confused me. You also need "anisotropy" for birefringence. Why not call it the "anisotropic reflection ratio" or something like that? "boundary reflection" is more appropriate than "boundary scattering" (the latter suggests some diffuse scattering which is not accounted for in this context. However, this may be a matter of taste.)**

AC2.12. We will take your suggestion and rename β as the "anisotropic reflection ratio".

**RC2.13. l 137 remove abbreviation SISO. It is not used later on.**

AC2.13. We will remove abbreviation "SISO" in the revised manuscript.

**RC2.14. l 150 include uncertainties for this angle here and elsewhere.**

AC2.14. See AC2.10.

**RC2.15. l 169 double-check that $s_{vh} = s_{hv}$ (see comment above).**

AC2.15. We address this comment in AC2.3.

**RC2.16. I understand how this azimuthal phase difference is calculated, but I don't understand what the additional value is. What inferences are drawn from the co-polarised phase difference that cannot be drawn from the hhvv phase angle?**

AC2.16. The four-quadrant patterns in the co-polarised (*hh*) phase difference plots are coincident with the location of the nodes in the co-polarised power anomaly plots (Brisbourne et al. 2019). The location of these nodes and four-quadrant patterns then constrains the 90° ambiguity in the cross-polarised (*hv*) power anomaly (See AC2.19). We find it to be more clear to locate the nodes and check the fabric orientation using a combination of investigating the co-polarised nodes and four-quadrant patterns (as in Brisbourne et al. 2019) than in the *hhvv* phase angle plot. Having the co-polarised power anomaly and phase difference plot may be useful in case of an abrupt shift in fabric orientation (again, as evidenced in Brisbourne et al. 2019).

We acknowledge that we may not have made our above method clear in the manuscript. We will summarise the above explanation when describing our method of identifying and resolving the fabric orientation at the end of Section 3.4.

**RC2.17. l 193 This first paragraph is more methods to me than results.**

AC2.17. We will move the sentences "A pad factor of 2… with the same moving matrix dimensions." to Section 3.3.

**RC2.18. As stated above the 90° are suspicious. Also, what are the ±7° based on? I would think that errors in antenna positioning are larger.**

AC2.18. In AC2.3, we state that the 90° is accurate. The ±7° is the 95% confidence interval (mistakenly attributed as standard error in the manuscript, which we will correct). The confidence interval was calculated from the ensemble of estimates of axis orientation at each depth bin (azimuthal minima, bright green dots in Figure 2). These errors are unrelated to antenna positioning. To address your concern, in the future revised manuscript, we will incorporate errors in antenna positioning by calculating the deviation of the symmetry axis from 90° for all 10 measurement sites (under assumption that (i) the flow regime and the resulting fabric at all sites are similar and therefore oriented in the same direction; and (ii) antennas were oriented in the same direction at each site, which we state in AC2.3, with deviations resulting from human error in antenna positioning).

**RC2.19. l 213 Typo? This ambiguity cannot be resolved in the *hh* power anomaly shown in Fig 2b. You need to use the polarity of the phase gradient.**

AC2.19. On the contrary: if anisotropic scattering is present in the imaged ice, the 90° ambiguity in the cross-polarised (*hv*) power anomaly can be resolved by determining either (i) the azimuthal minima of the co-polarised (*hh*) power anomaly plot, or (ii) the azimuth of the centre of the four-quadrant patterns in the co-polarised phase difference plots, as stated in Section 5.1 of Brisbourne et al. (2019). If co-polarised nodes are weak due to strong anisotropic scattering, the sign reversals in the four-quadrant patterns remain diagnostic.

On the other hand, if there is no anisotropic scattering present, then both the co-polarised power anomaly and phase difference plots will present 90° ambiguity as shown in Figure 2a of Brisbourne et al. (2019).

Because we observe at least moderate levels of anisotropic scattering in our observations (i.e. $1 \leq \beta \leq 5$ in Figure 3), we stand by our belief that the use of the *hh* power anomaly and phase difference plots to resolve the 90° ambiguity in the *hv* power anomaly is a valid technique.

**RC2.20. l 221 at least estimate these "human" errors**

AC2.20. See AC2.18 for how we will estimate and incorporate human errors in antenna positioning.

**RC2.21. why is the "anisotropy" an integer value?**

AC2.21. Apologies. $\beta$ (the anisotropic scattering parameter in the manuscript, now termed the anisotropic reflection ratio) should have units of dB. Because anisotropic scattering is poorly constrained, we restricted our model fitting to only using integer values and interpolated through depth between points to avoid overfitting the data.

**RC2.22. I 237 I think I missed something here: Aren't those node pairs simply depths where the phase shift between ordinary and extraordinary wave is odd integer multiple of $\pi$ ? Clearly they will have a correspondence in the azimuthal phase difference (which is directly related to the phase angle). I don't understand the deeper physical implication of this 'four quadrant pattern yet.**

AC2.22. See AC2.16 for further discussion on the relevance of the co-polarised phase difference.

**RC2.23. Fig. 4 I appreciate the error bars on the ApRES derived $E_2 - E_1$. Please state more clearly how those were derived.**

AC2.23. We state our error calculation in L187 (using the Cramer-Rao bound to obtain the associated phase error of $d\phi_{hhvv}/dz$, and following the ensemble method of Jordan et al. 2019). $E_2 - E_1$ is obtained directly through $d\phi_{hhvv}/dz$, and the error is then transferred accordingly. We will modify L187 to explicitly state that we calculate the phase error of $d\phi_{hhvv}/dz$, which then transfers accordingly to uncertainties for each $E_2 - E_1$ depth bin.

**RC2.24. I 289 This is not the "best model" that matches observed results. It is a model that explains some of the features in the observations**

AC2.24. We will implement this correction in the revised manuscript.

**RC2.25. I 330 How fast does the ice-fabric structure adapt to a new strain regime? I think some sort of statement in this regard is required to better justify statements of ice-divide stability.**

AC2.25. Brisbourne et al. (2019) states that, although the time taken to overprint a preexisting fabric is dependent on temperature, strain rate, and stress regime, it is poorly constrained excluding those from laboratory results. Recent results by Lilien et al. (in review) suggest that it takes tens of thousands of years to overprint fabric (assuming that lattice rotation dominates the fabric evolution, as most evidence indicates). Therefore, we can implement the same reasoning as that of Brisbourne et al. (2019) and state that the measured COF distribution aligned with the observed surface strain orientation through depth reflects the current ice flow regime, assuming that the vertical assumption (i.e. $E_2 - E_1$) holds through the range of measured depths. We will add these above statements to the revised version of our manuscript.

**References**

Brennan, P. V., Nicholls, K. W., Lok, L. B., and Corr, H. F. J.: Phase-sensitive FMCW radar system for high-precision Antarctic ice shelf profile monitoring, IET Radar, Sonar & Navigation, 8, 776–786, https://doi.org/10.1049/iet-rsn.2013.0053, 2014.

Brisbourne, A. M., Martín, C., Smith, A. M., Baird, A. F., Kendall, J. M., and Kingslake, J.: Constraining Recent Ice Flow History at Korff Ice Rise, West Antarctica, Using Radar and Seismic Measurements of Ice Fabric, Journal of Geophysical Research: Earth Surface, 124, 175–194, https://doi.org/10.1029/2018JF004776, 2019.

Burr, A., Noël, W., Trecourt, P., Bourcier, M., Gillet-Chaulet, F., Philip, A., and Martin, C. L.: The anisotropic contact response of viscoplastic monocrystalline ice particles, Acta Materialia, 132, 576–585, https://doi.org/10.1016/j.actamat.2017.04.069, 2017.

Dall, J.: Ice sheet anisotropy measured with polarimetric ice sounding radar, in: 30th International Geoscience and Remote Sensing Sympo- sium (IGARSS 2010), pp. 2507–2510, Honolulu, HI, https://doi.org/10.1109/IGARSS.2010.5653528, 2010.

Fujita, S., Maeno, H., and Matsuoka, K.: Radio-wave depolarization and scattering within ice sheets: A matrix-based model to link radar and ice-core measurements and its application, Journal of Glaciology, 52, 407–424, https://doi.org/10.3189/172756506781828548, 2006.

Horgan, H. J., Anandakrishnan, S., Alley, R. B., Burkett, P. G., and Peters, L. E.: Englacial seismic reflectivity: Imaging crystal-orientation fabric in West Antarctica, Journal of Glaciology, 57, 639–650, https://doi.org/10.3189/002214311797409686, 2011.

Howat, I. M., Porter, C., Smith, B. E., Noh, M.-J., and Morin, P.: The Reference Elevation Model of Antarctica, The Cryosphere, 13, 665–674, https://doi.org/10.5194/tc-2018-240, 2019.

Jordan, T. M., Schroeder, D. M., Castelletti, D., Li, J., and Dall, J.: A Polarimetric Coherence Method to Determine Ice Crystal Orientation Fabric From Radar Sounding: Application to the NEEM Ice Core Region, IEEE Transactions on Geoscience and Remote Sensing, pp. 1–17, https://doi.org/10.1109/tgrs.2019.2921980, 2019.

Jordan, T. M., Schroeder, D. M., Elsworth, C.W., and Siegfried, M. R.: Estimation of ice fabric within Whillans Ice Stream using polarimetric phase-sensitive radar sounding, Annals of Glaciology, 61, 74–83, https://doi.org/10.1017/aog.2020.6, 2020a.

Jordan, T. M., Martin, C., Brisbourne, A. C., Schroeder, D. M., and Smith, A. M.: Radar characterization of ice crystal orientation fabric and anisotropic rheology within an Antarctic ice stream, Earth and Space Science Open Archive ESSOAr, pp.1-48, https://doi.org/10.1002/essoar.10504765.1, 2020c.

Kennedy, J. H., Pettit, E. C., and Di Prinzio, C. L.: The evolution of crystal fabric in ice sheets and its link to climate history, Journal of Glaciology, 59, 357–373, https://doi.org/10.3189/2013JoG12J159, 2013.

Looyenga, H.: Dielectric constants of heterogeneous mixtures. Physica, 31, 401–406, https://doi.org/10.1016/0031-8914(65)90045-5, 1965.

Lilien, D. A., Rathmann, N. M., Hvidberg, C. S., and Dahl-Jensen, D.: Modeling ice-crystal fabric as a proxy for ice-stream stability, Journal of Glaciology, pp. 1-37, in review.

Matsuoka, K., Power, D., Fujita, S., and Raymond, C. F.: Rapid development of anisotropic ice-crystal-alignment fabrics in- ferred from englacial radar polarimetry, central West Antarctica, Journal of Geophysical Research: Earth Surface, 117, 1–16, https://doi.org/10.1029/2012JF002440, 2012.

Morlighem, M., Rignot, E., Binder, T., Blankenship, D., Drews, R., Eagles, G., Eisen, O., Ferraccioli, F., Forsberg, R., Fretwell, P., Goel, V., Greenbaum, J. S., Gudmundsson, H., Guo, J., Helm, V., Hofstede, C., Howat, I., Humbert, A., Jokat, W., Karlsson, N. B., Lee, W. S., Matsuoka, K., Millan, R., Mouginot, J., Paden, J., Pattyn, F., Roberts, J., Rosier, S., Ruppel, A., Seroussi, H., Smith, E. C., Steinhage, D., Sun, B., van den Broeke, M. R., van Ommen, T. D., van Wessem, M., and Young, D. A.: Deep glacial troughs and stabilizing ridges unveiled beneath the margins of the Antarctic ice sheet, Nature Geoscience, 13, 132–137, https://doi.org/10.1038/s41561-019-0510-8, 2020.

Young, T. J., Schroeder, D. M., Christoffersen, P., Lok, L. B., Nicholls, K. W., Brennan, P. V., Doyle, S. H., Hubbard, B., and Hubbard, A.: Resolving the internal and basal geometry of ice masses using imaging phase-sensitive radar, Journal of Glaciology, 64, 649–660, https://doi.org/10.1017/jog.2018.54, 2018.

---

## Author Response (AR1)

**Author response to the review of Young et al. "Rapid and accurate polarimetric radar measurements of ice crystal fabric orientation at the Western Antarctic Ice Sheet (WAIS) Divide deep ice core site" [Manuscript # tc-2020-264]**

We would like to thank the two reviewers, Martin Rongen and Reinhard Drews, for their thorough, thoughtful and constructive reviews. Our initial response is available under Interactive Discussion, and we document the actual implementation of these changes below. Please find below the two reviewers' comments (RC) in **bold**, each followed by the authors' response (AC). Line, figure, and page numbers mentioned by the reviewers (within RC) refer to the original manuscript, and those mentioned by the co-authors (within AC) refer to the revised manuscript.

**Review by Martin Rongen (30 October 2020)**

**General comments**

**RC1.1. The authors present a quad-polarimetric radar measurement at WAIS Divide. The dataset is interesting in its innovative nature as well as in the quality of the derived results. The method follows in direct succession to earlier works developed by Shuji Fujita and Tom Jordan among others. While previous measurements required a manual rotation of the antenna system in small steps in order to measure the azimuthal variation associated with the birefringence signature, the quad-polarimetric measurement allows for signal at arbitrary azimuths to be deduced from just four antenna orientations. The results are validated by comparison to ice core data.**

**The overall presentation is detailed and rigorous. Some improvements may be made by giving a clearer structure to the results and discussion sections (see later specific comments). As a non-expert on glaciological radar measurements, the theory and methods section was challenging but the provided references proofed to be very helpful. For the paper to stand on its own some more context/details may be added (see specific comments). Some more discussion may also be added to section 5.3 (Methods comparisons and limitations). It currently gives a fair comparison between radar and ice core / sonic measurements but is short on the specific limitations and assumptions involved in generating data for arbitrary azimuth angles using the quad-polarimetric data.**

AC1.1. Thank you for your detailed review, and you as well as Reinhard Drews have correctly identified that we need to further discuss some method comparisons and limitations. We hope that we've addressed your concerns in the specific comments section below.

**Specific comments**

**RC1.2. The readability of the results section would be greatly improved by structuring it in sub-sections, as is also done for the other sections. A possible structure could be:**
- **4.1 Experimental results from WAIS (up to line 224)**
- **4.2 Modelling the observed data (lines 224-255)**
- **4.3 Fabric asymmetry estimation (lines 255 ff)**

AC1.2. Thank you for these suggestions. We have implemented these subsection titles with slight modifications:

- 4.1 Experimental results from WAIS Divide (L221)
- 4.2 Comparison between observed and modelled polarimetric signals (L259)
- 4.3 Estimation of azimuthal fabric asymmetry (L299)

**RC1.3. The meaning of the pad factor mentioned in line 193 is unclear.**

AC1.3. The pad factor refers to the amount of zero-padding applied to the time-domain signal relative to the total length of the original signal. We apply zero-padding to our data with a pad factor of 2, as recommended by Brennan et al. (2014). We have incorporated this description in L212-213: "Here, the pad factor represents the total length of the signal after zero-padding relative to the total length of the original signal." Note that this description has been moved to the end of Section 3.3 in response to RC2.17.

**RC1.4. The details and reliability of the firn correction as introduced in line 198 are unclear. It is mentioned that this correction amplifies the estimated $E_2 - E_1$ values in the shallow ice and surprisingly large fabric asymmetries are then measured in that depth range. Thus the firn correction merits more attention (and maybe test without the correction) during the discussion (line 265 and 305) may be warranted.**

AC1.4. We implement firn correction as suggested in the Appendix of Jordan et al. (2020a). Because the equation that drives this correction was derived from founding ice-penetrating radar principles (e.g. the ice-air volume fractions in the mixing relations of Looyenga 1965), we believe these corrections to be physically representative of firn anisotropy. Notwithstanding, we agree with your belief that the fabric asymmetries in firn are surprisingly large. We have therefore updated Figure 6 to include an inset that shows the effect of firn correction on resulting $E_2 - E_1$ values. We have added a sentence that justifies why we apply firn correction (L344-346): "As the applied firn correction is based upon established ice-air volume fractions in the mixing relations of Looyenga (1965), we believe these corrections to be physically representative of any fabric anisotropy within the firn layer." We then provide a physical explanation to why anisotropy is enhanced (L352-355): "Although comparative studies addressing the physical origins of fabric anisotropy do not exist for the firn layer, it is likely that the effects of prolonged firn densification on crystal rotation will induce some amount of azimuthal anisotropy within this layer (Burr et al. 2017)."

**RC1.5. In line 196 it may be worth mentioning that the Jordan et al. (2019) prescription to evaluate $d\varphi_{hhvv}/dz$ is not actually based on the phase plot itself but on the real and imaginary components of the coherence as given in equation 7b.**

AC1.5. We have clarified our methods with the inclusion of the following in L214-215: "We evaluated $d\phi_{hhvv}/dz$ using the real and imaginary components of $c_{hhvv}$ (Eq. 7b) and estimated its respective error…"

**RC1.6. The reasons for and consequences of switching from an FIR filter to the method described in l.197ff remain unclear.**

AC1.6. We had used our method (a 2-D median filter + 2-D peak convolution) in a previous paper that processes 2-dimensional (x-z and y-z) imagery from a multistatic pRES array (Young et al. 2018). The reason for using this then was to remove high-frequency noise and speckle. We have also visually checked the filtered image to ensure that there are no remaining "rapid phase excursions due to a modulo $2\pi$ artefact" (Dall 2010). Our results are not sensitive to filtering within a wide range of parameters: we can validate our final results with equivalent ice core measurements, which allowed us to test several parameters as well as show that the choice of filter is not limited. Crucially, we increase our bulk depth resolution to 15 m compared with the 100 m used by Jordan et al. (2019), which shows the benefits of experimenting with different filters. Therefore, we are confident that the results of our method choices are comparable to and potentially exceed those of Jordan et al. (2019).

**RC1.7. The anisotropy parameter $\beta$ seems to be missing a unit (dB?) in the caption of Figure 3.**

AC1.7. We had originally referred to $\beta$ as the ratio between the (E-field) Fresnel reflection coefficient along the y-plane relative to the x-plane, and therefore $\beta$ is supposed to be unitless. However, we think that expressing it in dB would relate better to birefringence. Therefore, we have redefined $\beta$ to be the intensity ratio of anisotropic scattering and have rescaled the values to dB ($20 \log_{10}$). This is now in line with the majority of previous studies (e.g. Fujita et al. 2006, Jordan et al. 2019). This explanation is written explicitly in L142-146.

**RC1.8. For depth greater than 1200 m (and to a lesser extent around 600 m) the derived $E_2 - E_1$ values become rather unstable. While this is commented on and partially reflected in larger error bars, a population of outliers with small fabric asymmetries as well as small error bars is a bit worrying. It may be beneficial to show a plot of the coherence magnitude. Given vanishing magnitudes, the phase becomes unconstrained leading to erratic $d\varphi_{hhvv}/dz$ values. In the deepest region the phase in Figure 2.e is more unstable as a function of depth than expected from the model calculation.**

AC1.8. We agree with your suggestion to include $|c_{hhvv}|$. Your hypothesis that the outliers in Figure 6 (originally Figure 4) correspond to small $|c_{hhvv}|$ is correct: beyond 1300 m, values of $|c_{hhvv}|$ are generally <0.3, with increasing amounts of points removed with higher $|c_{hhvv}|$ thresholds. We have added a plot of $|c_{hhvv}|$ in Figure 4 (as well as Figures S1-10) as panel (d) to constrain the validity of these outliers, and have re-plotted Figure 6 with a $|c_{hhvv}|$ threshold of 0.3. We have included a description of the application of $|c_{hhvv}|$ in L312-319:

> "Even after filtering out $E_2 - E_1$ below a $|c_{hhvv}|$ threshold of 0.3 (green asterisks in Fig. 6), ApRES measurements beyond 1200 m show a marked increase in variability that, although centred around corresponding depth values in the WDC, varied between

0.04 to 0.42. We similarly observe a sevenfold jump increase in the associated standard deviation, ranging from values averaging 0.006 at depths of 200-1200 m to 0.04 within the depth range of 1200-1400 m. There exists a small cluster of four outliers with low values ($E_2$ - $E_1$ < 0.1) at depths between 1250 and 1350 m with anomalously low error bars, even after initial $|c_{hhvv}|$ filtering. Setting increasingly higher $|c_{hhvv}|$ thresholds to 0.4 and 0.5 removes these outliers as well as all calculated $E_2$ - $E_1$ values beyond 1250 and 1100 m respectively."

We have also added two sentences to Section 5.3 to suggest potential ways to overcome this limitation (L435-437): "High SNR does not always equate to high polarisation coherence ($c_{hhvv}$), and vice versa. It is, however, plausible that larger datasets that employ higher amounts of chirp-averaging may increase the SNR needed to extend beyond the current depth limitation of 1400 m."

**RC1.9. The sentence "The birefringence of an individual crystal and its COF are related to the bulk ..." in line 97 reads a bit odd as a COF only applies to an ensemble of crystals. Maybe change to something like "the birefringence of individual crystals and their COF".**

AC1.9. Your suggestion is good, and we have implemented this in the revised manuscript.

**RC1.10. The term "depth step" in line 124 is a bit technical. Something like "depth where a reflection occurs" would be clearer to the reader.**

AC1.10. We have replaced "... for each depth step and azimuthal orientation" with "... at each discrete scattering layer and azimuthal orientation", which is in line with the terminology used by Fujita et al. (2006) in their Section 2.2.

**RC1.11. The meaning and relevance of the rotation matrix *R* as introduced in line 133 is unclear. To my understanding, it represents the rotation of the COF principle axis *of each traversed ice layer* with respect to a reference system defined by the antennas.**

AC1.11. You are correct: the rotation matrix *R* is used in Equation 2 to reconstruct the theoretical signal components for each azimuthal shift *θ*, for which the component in question is either *T* (transmission between the antennas and the scattering layer) or *Γ* (reflection at the scattering layer). The use of *R* in essence replicates an azimuthal-rotational experiment (acquisitions made at each rotational increment of the antenna acquisition plane). The second half of the mentioned paragraph has been rewritten to now read (L137-140): "The rotation matrix *R*, with *R'* its transposition, is used in Equation 2 to reconstruct the theoretical signal components with respect to *θ*, for which the components are either *T* (transmission between the antennas and the scattering layer) or *Γ* (reflection at the scattering layer). *T*, *Γ*, and *R* are all 2x2 matrices and are each detailed respectively..."

Please also note that we have also redefined our rotation matrix to be [cos *θ* , -sin *θ* ; sin *θ* cos *θ*] (Eq. 4) as part of a larger restructuring of the definitions of *h* and *v* (see RC2.10).

**RC1.12. It is mentioned that $s_{hv}$ and $s_{vh}$ should be identical given an ideal measurement. Has the difference between these two orientations been studied and the potential impact quantified?**

AC1.12. In theory, $s_{hv} = s_{vh}$ which is well known according to the Lorentz reciprocity theorem. In practice, there will be small differences including (but not limited to): (i) differences in the beam pattern between the transmitting and receiving aerials, (ii) random clutter within the transmitted media; and (iii) human error in antenna positioning. This topic is well studied (e.g. there is a 100+ page book (Stumpf 2018) purely dedicated to antenna signal reciprocity!) but in general, the main differences between $s_{hv}$ and $s_{vh}$ are due to human error during data collection. In general, our data has minimal difference between $s_{hv}$ and $s_{vh}$, although there is some additional variability seen at the near surface. This discrepancy may account for the mismatch between measured (Figure 4) and modelled co-polarised results (Figure 5) in the top ~200 m.

We have shown the additional variability in the near surface between $s_{hv}$ and $s_{vh}$ (note that this was incorrectly labelled as $e$ in the previous draft) by modifying Figure 2 (originally Figure 2a, with the new figure shown below). We have added the description above regarding deviations from the Lorentz reciprocity theorem in L181-188. We have suggested this mismatch to be the link between the discrepancy between measured and modelled results in L294-296.

Additionally, sometimes $s_{hv} = -s_{vh}$ due to the 180° ambiguity in antenna position. This is true in our case, we state this anomaly in L187-188, and we explain further in AC2.3 and how this changes some of our figures and results.

[Figure]

**Figure**. Mean (polarisation-averaged) power return for each antenna orientation. Insets show magnification of power returns at two different 20 m intervals (one shallow, one deep).

**RC1.13. The COF orientation of each depth layer is resolved "by tracing the azimuthal minima in the cross-polarized power anomaly". While this may be a good**

approximation for this measurement, I wonder if this technique is generally applicable in the presence of strongly varying COF orientations. Assuming, for example, a constant angle of 20° in the top 500 m and a constant angle of 60° below, my understanding is that the minimum in the cross-polarized power anomaly would only slowly migrate towards 60° below 500 m as the bulk propagation is initially still dominated by the conditions above. One essentially measures the average COF orientation up to the scattering depth. But there may be a misunderstanding on my part here. A comment would be appreciated.**

AC1.13. In general, this method (as well as other published methods so far, e.g. Brisbourne et al. 2019, Jordan et al. 2020a) is accurate if the COF orientation is depth invariant. If the orientation undergoes an abrupt switch, such as the example that you propose above and below 500 m, the minima in power anomaly would similarly undergo an abrupt switch (e.g. at ~215 m in Figure 4 of Brisbourne et al. 2019). If this happens, the 90° ambiguity in the cross-polarised power anomaly plot would need to be resolved by either the co-polarised power anomaly or the co-polarised phase difference plots (see AC2.19). Additionally, there may also exist over or under-estimation of fabric asymmetry if the identified ice optic axis deviates from the true fabric orientation (e.g. Figure 4c of Jordan et al. 2020b).

We have summarised the above paragraph in Section 5.3 (L406-416):

> The cross-polarised power anomaly is generally a robust method of identifying the fabric orientation in slow-moving ice (Li et al. 2018). Here, we show that this method is reasonably accurate for depth-invariant eigenvectors (Fig. 4). In the case of a gradual rotation of the fabric orientation through depth, the cross-polarised power anomaly should undergo a similarly gradual rotation (Ershadi et al. 2021). This is also true in the case of an abrupt switch in COF, as evidenced at Korff Ice Rise (Brisbourne et al. 2019), where the cross-polarised power anomaly undergoes a similarly abrupt shift in azimuth. In elementary cases, the 90° ambiguity that exists in the cross-polarised power anomaly (Li et al. 2018) can potentially be resolved from the methods given in the previous paragraph. However, if the fabric orientation were to change rapidly with depth, using only the cross-polarised power anomaly to determine and distinguish between the two eigenvector orientations may produce erroneous results as demonstrated by Ershadi et al. (2021). In all cases, if the radar-derived fabric orientation is offset in azimuth from its true orientation, this mismatch will result in corresponding over- or under-estimation of azimuthal fabric asymmetry (Jordan et al. 2020b).

**RC1.14. As noted in the general comments section 5.3 would benefit from a discussion of the specific limitations and assumptions involved in generating data for arbitrary azimuth angles using the quad-polarimetric data.**

AC1.14. See AC2.4 for a discussion on the comparisons between quad-polarimetric and azimuthal rotational experiments.

**RC1.15. Section 5.1 paragraph 2 (lines 281-288) seems better suited in section 4.1 (results, modeling), here a Figure similar to Figure 5 in the Fujita 2006 paper may also be illustrative, showing that birefringence results in nodes in the power anomalies while**

**anisotropic scattering results in a band structure, the spacing of which is a function of the scattering strength.**

AC1.15. While a figure similar to Figure 5 of Fujita et al. (2006) may be useful, this specific figure has already been reproduced multiple times in subsequent papers (e.g. Figure 7 of Matsuoka et al. 2012, Figure 2 of Brisbourne et al. 2019) and we feel like there is already enough literature that describes the depth and azimuthal variability of the co-polarised nodes. Furthermore, we believe that Figure 5 (originally Figure 3) already serves the same purpose, in particular, a comparison between Figure 5a-c and 5d which shows the controls of anisotropic scattering ($β$) and birefringence ($ε(z)$, through $E_2$-$E_1$). Therefore, we do not reproduce Figure 5 of Fujita et al. (2006).

We have chosen to remove the second paragraph of Section 5.1, as (i) you have rightly pointed out that it would have been better suited in the Results section; and (ii) these statements are reminders of what has already been stated in the Results section as well as elsewhere, such as in the Introduction.

**Technical comments**

**RC1.16. The link for the Mott, H. (2006) reference appears to be dead.**

AC1.16. Thank you for pointing this out. We have fixed the link and ensured that it works.

**RC1.17. In line 156, spectra so should be plural as are the amplitudes.**

AC1.17. Corrected by changing "complex amplitudes" to the plural case (L172).

**RC1.18. In the title of subsection 5.1 it would be more consistent to refer to "anisotropic scattering" instead of the more ambiguous "anisotropy"**

AC1.18. We apologise for the ambiguity and have corrected the subsection title to now state "anisotropic scattering".

**Review by Reinhard Drews (19 November 2020)**

**Summary**

**RC2.1. In their paper "Rapid and accurate polarimetric radar measurements of ice crystal fabric orientation at the Western Antarctic Ice Sheet (WAIS) Divide deep ice core site", Young and co-authors present an ApRES radar dataset, which they use to infer the ice-fabric characteristics continuously to a depth of 1500 m. Main results include quantification of the horizontal ice anisotropy with a depth invariant ice-fabric orientation that is aligned with the directions of the principal strain rates. The inferences are validated with data from the WAIS ice core, and some conclusions are drawn about the ice-divide stability throughout the Holocene.**

**Overall, this paper is nicely written and the authors do a commendable job in guiding the reader through the methods and results. However, in places I find the paper unnecessarily superficial and I don't see novel aspects clearly. I also suspect (but I am not certain) that parts of the azimuthal reconstruction may be erroneous leading to wrong inferences in terms of the ice-fabric orientation. Below, I mention a number of major comments/questions how this can be improved. Applications of radar polarimetry are still rare, and I hope that the points raised below will help to improve the next version of this paper.**

AC2.1. Thank you for your detailed review, and we appreciate your honesty in that you have disclosed your potential conflict of interest. We believe your review is unbiased and correctly identifies several areas will need to be addressed, specifically the three major comments (RC2.2 - 2.4). We have attempted to alleviate your concerns in our responses below. Ultimately, we hope that our paper can contribute to the radar polarimetry literature and advocate for radar as a robust tool to derive fabric strength and orientation estimates.

**RC2.2. Clarify methodological advance**

**It is stated that this study "..extends previous qualitative analyses [...] to obtain quantitative measurements.." (l. 285). Can you highlight more clearly what those extensions were compared to previous studies? From what I can see so far, this study nicely applies previous developments to a single new site, but I struggle to see the extensions. The link between the polarimetric phase gradient and ice- fabric parameters is based on the cited papers Fujita et al., 2006 and Jordan et al., 2019. Arguably matching the angular distance of co-polarization nodes with a 2D optimisation is new (l.233), but at least the dependency of this distance as a function of anisotropic scattering is already approximated in Fujita 2006. Also advantages or pitfalls (e.g., in terms of uniqueness and uncertainties involved) of this approach are not discussed.**

**I suppose that this paper is the first to explicitly focus on synthesising quad-polarimetric measurements for ApRES, although the related methodology is known from radar polarimetry textbooks (e.g., the cited Mott, 2006). The inferences drawn from this method about the "high angular" resolution are not credible as**

**currently presented (see comment below). Also the lack of rotational dataset at this site makes it hard to discuss advantages/disadvantages of both approaches. I suggest a dedicated section were improvements and distinct differences compared to previous studies are highlighted more explicitly.**

AC2.2. In our opinion, this manuscript extends the literature (this being radar polarimetry for glaciological applications) through: (i) validation of co-polarised power anomaly and phase difference plots; (ii) publication of the quad-polarimetric reconstruction method; and (iii) direct validation of radar-derived measurements of ice fabric to that of ice cores at high depth resolution. We are fortunate that our results are simple to comprehend, and we believe that this, combined with the straightforward layout of the quad-polarimetric reconstruction method will help those who wish to apply radar polarimetric methods to glaciology. Due to the growing interest in radar polarimetry applications within the glaciological community as of late, we believe that our manuscript is not only valuable, but also timely for the above three reasons. In detail:

> (i) is shown through the direct comparison between measured (Figure 4) and modelled (Figure 5) results, where the birefringence in the latter was induced using fabric eigenvalues from thin-section analysis of ice cores. (The anisotropic reflection ratio is instead estimated through matching the locations of the co-polarised nodes, the four-quadrant patterns, and the *hhvv* phase with the measured results.) While we concede that the theory behind this comparison has previously been quantified in Fujita et al. (2006), they stop short of presenting modelled results complementary to their observations (in their case, fabric data from the Mizuho and Dome Fuji ice cores). Similarly, while fabric measurements from polarimetric radar measurements have been verified to ice core data by Jordan et al. (2019), they implement only the polarimetric coherence method (i.e. analysis of $s_{hhvv}$) and do not present any co- or cross-polarised datasets. On the other hand, Brisbourne et al. (2019) present co-polarised power anomaly and phase difference plots of radar measurements conducted at Korff Ice Rise but focus on the relative orientation of the antenna polarisation plane relative to the ice optic axis, and present only qualitative observations in terms of fabric strength. Our manuscript not only directly compares measured results to modelled plots, but also quantifies azimuthal fabric strength through the polarimetric coherence method and verifies results directly with ice core data. Therefore, our manuscript reconciles the methods of the above three studies.

> Regarding (ii), we felt a need to include the complete equations that show how the received signal (both power in Equation 4 and phase in Equation 5) can be reconstructed from quad-polarimetric measurements. You are correct in that these methods are established, and we do not attempt to take credit for them. However, these methods are often presented within dense literature specifically directed towards radar engineering applications and may be daunting for the majority of glaciologists (myself included!). Rather, we attempt to introduce them to the glaciological literature in an approachable format so that they are easily accessible for researchers wishing to conduct quad-polarimetric experiments.

> Regarding (iii), our results show that choosing a nominal depth-averaging window of 15 m (which is our nominal bulk-depth resolution) when applying the polarimetric

coherence method (Equation 6) produces estimates of azimuthal fabric asymmetry that closely match corresponding results from ice-core thin section data at similar intervals. While Jordan et al. (2019) also use the polarimetric coherence method to match fabric asymmetry estimates between radar and ice core measurements, also showing comparative results, they had chosen to use a conservative nominal depth interval (the bulk-depth resolution) of 100 m. Our study shows that this nominal depth interval can be reduced down to levels at or exceeding that of the vertical spacing of ice core thin sections, while still producing comparable results to ice core measurements. The ability to validate the processing parameters with ice core measurements gives confidence in both our methods and results.

As you may already know, there are multiple pitfalls in using the polarimetric coherence method, which are detailed in Jordan et al. (2019) and (2020a). We do not wish to regurgitate what has already been said, but we acknowledge that we have provided little caveats, which makes it seem like we are "overselling" our results. We have separated what was Section 5.3 (Method comparisons and limitations) into two sections: Section 5.3 (Radar polarimetric methods to determine fabric strength and orientation) and Section 5.4 (Broader comparisons of geophysical methods to infer ice fabric properties).

There are 5 paragraphs comprising Section 5.3. The first paragraph highlights the strengths of this study with reference to its similarities and differences from previous studies of power anomaly and polarimetric coherence. The second paragraph describes how to distinguish between the two horizontal eigenvector orientations (see AC2.16). The third paragraph critiques the use of the cross-polarised power anomaly to identify the ice optic axis (see AC1.13). The fourth paragraph presents a discussion between azimuthal rotation methods and quad-polarimetric measurements (see AC2.4). The fifth paragraph discusses the signal-to-noise ratio, the hhvv coherence, and how this may be overcome (see AC1.12).

**RC2.3. Coincidental symmetry at $\theta$ = 90°?**

**In Figs. 2b-e one principal axis of the ice-fabric appears at the local azimuthal angle $\theta$ = 90° (i.e., all panels have a reflectional or rotational symmetry around the $\theta$ = 90° axis). This means that during measurements antennas were coincidentally placed parallel (*hh*) and perpendicular (*vv*) to the (at the time) unknown ice-fabric orientation. It is possible that the operators in the field made a conscious decision here because $\theta$ = 90° aligns with the strain rate (not the ice-flow) direction. However, given uncertainties involved in determining the direction of maximum strain rate and the antenna orientation, the $\theta$ = 90° symmetry almost seems too much of a coincidence. Based on our own experience with analysing quad-polarimetric data, we suggest that the authors double-check that indeed $s_{hv} = s_{vh}$. We found occasionally that $s_{hv} = -s_{vh}$ without satisfying explanation as to why this can be the case (e.g., inconsistencies in labelling and naming of antenna orientations in the field?). However, if it is the case, then reconstruction of the ApRES signal using eq. (4) forces a symmetry axis at $\theta$ = 90° exemplified below for the $s_{hh}$ component:**

$$S_{11} = s_{hh}(\theta) = \underbrace{s_{hh}\cos^2\theta + s_{vv}\sin^2\theta}_{\substack{\text{symmetric at} \quad \theta=90°}} + \underbrace{(s_{vh} + s_{hv})\sin\theta\cos\theta}_{\substack{\text{anti-symmetric at} \quad \theta=90°}}$$

$$\underbrace{\qquad\qquad\qquad\qquad\qquad\qquad\qquad\qquad\qquad\qquad\qquad\qquad}_{\text{in general no symmetry axis at} \quad \theta=90° \quad \text{unless} \quad s_{hv}=-s_{vh}}$$

**The graphic below illustrates how this would be reflected in a full azimuthal reconstruction where the principal axis around θ = 35, 125° in the top plot are erroneously mapped to θ = 90, 180°. Without a co-polarized, rotational dataset this will occur unnoticed.**

[Figure]

synthetic 2 layer fantasy froward model

"wrong", but very symmetric reconstruction

**Maybe it will be helpful to investigate this further. Alternatively, state explicitly how the *hh* and *vv* directions were defined in the field, and why it makes sense that those axis align almost perfectly with the principal directions of the ice-fabric.**

AC2.3. Your suspicions in the exact alignment of the antenna polarimetry plane in Fig. 2 with the strain axes is warranted. In our initial manuscript submission, we show the results from one ApRES site, but we did not mention in our manuscript that we had also obtained ApRES measurements from nine other sites along a ~6 km-long transect. We have included the transect in the updated Figure 2. We now include results from all 10 sites in Figures S1-S10 and results from all sites are consistent with each other. The results displayed in Fig. 4 (originally Fig. 2) represents Site I, and is the 9th of 10 total sites in terms of their relative position along the transect. The antenna axis (i.e. the plane that runs between the transmitting and receiving antenna) is orthogonal to the transect line, and we have clarified this in the text as well as in Figure 1 (see AC2.10). This is opposite to the assignments we had used in the original TCD manuscript draft (i.e. *h* is now *v* and *v* is now *h*), and therefore all plots had shifted by 90°.

Upon re-inspection of our code, we noticed a rounding error when importing the Northing and Easting coordinates that resulted in the antenna polarisation plane to be aligned exactly with the direction of maximum compression. In reality, the antenna polarisation plane is actually +7° (where positive numbers represent counterclockwise directions) from the strain compressive axis (rounded to the nearest degree). The orientations of velocities, strain axes, antenna axes, and eigenvectors have all been double-checked and should now all be consistent with each other.

Using data collected at Site I (the data used to produce Figure 2 in the initial TCD manuscript), we present in Figure AC1 2 sets of plots of co- and cross-polarised power anomaly and phase difference. Set 1 (panels a) assumes that $s_{hv} = s_{vh}$, and set 2 (panels b) assumes $s_{hv} = -s_{vh}$. In these figures, the mean and standard deviation of the $E_1$ (green) and $E_2$ (yellow) eigenvectors for panels (a) were calculated to be 90.3° ± 3.98° and 1.8° ± 3.81° respectively, and for panels (b), -90.6° ± 6.06° and -2.7° ± 5.99° respectively, using the same methods described in the TCD manuscript. Figure (AC1) shows that the eigenvectors were oriented at or very close to 0 and 90° irregardless of signage. Note that the x-axes are shifted 90° due to our re-assignment of $h$ and $v$ with respect to the E-field (see AC2.10).

However, because the two results give approximately the same results, we are not able to determine whether or not panels (a) or (b) shows the correct polarimetric patterns from Figure (AC1) alone. We then run the same investigation on the other 9 sites and show the output plots from one of these sites (Site D) in Figure (AC2). We regard the results shown in Figure (AC2) to have higher variability between results produced using each of the two assumptions in turn, but overall representative of all sites. With the additional consideration of Figure (AC2), we can say that all sites are centred about 0° regardless of the signage of $s_{hv}$ and $s_{vh}$, but the assumption that $s_{hv} = -s_{vh}$ produces realistic variations between the plots whereas $s_{hv} = s_{vh}$ essentially produces replicates of the co-polarised power anomaly regardless of the site, which may or may not match its cross-polarised power anomaly. For the example site (Site D) shown in Figure (AC2), the plots produce an $E_1$ and $E_2$ eigenvector orientation 98.9° ± 4.71° and -7.5° ± 4.68° despite having a co-polarised power anomaly plot centred around 0° under the first assumption ($s_{hv} = s_{vh}$, Figure AC2a), while the co-polarised and cross-polarised power anomaly plots are consistent with each other with the eigenvectors at 101.6° ± 5.96° and 7.4° ± 4.43° under the second assumption ($s_{hv} = -s_{vh}$, Figure AC2b). We assume that the deviation of the principal axis from exactly 90° in both assumptions are primarily the result of human errors in antenna positioning, as well as lateral migration of the ice optic axis across distance.

Assuming that the four orthogonal combinations of antenna orientations were conducted in the same way for all 10 sites (which, to the best of our knowledge, is true), the results from Figure (AC1) and Figure (AC2) show that (i) the positions of [$h$ | $v$] and [$v$ | $h$] that we used in the field (Figure 1a) correspond to the second assumption that $s_{hv} = -s_{vh}$; (ii) the use of the cross-polarised power anomaly in determining the principal axis is valid regardless of whether the first or second assumption is used; and (iii) the lateral eigenvectors for Site I about 90° and 0°, and aligned with the strain axes, is valid as well as being fortuitous.

Given the new knowledge gleaned from this investigation, we have:
   a. Re-processed our datasets using the assumption that $s_{hv} = -s_{vh}$, with an explicit mention in the methods (L187-188);

b. Updated our estimate of relative orientation of the $E_1$ and $E_2$ eigenvectors to 91° and -3° from the results of the re-processing;

c. Provided equivalent plots of Figure 4 for all 10 sites along the transect in the Supplementary Information as Figures S1-S10, with collated results in Table S1. The 10 sites show consistency across the entire transect;

d. Updated the modelled anisotropic scattering parameter and therefore updated Figure 5 to match the re-processed Figure 4.

[Figure]

**Figure AC1**. ApRES polarimetric power anomaly and phase difference measured at Site I, WAIS Divide. (Note that this is labelled as W08 in these plots as these are our internal numbering system) The top row of panels (a) were produced assuming $s_{hv} = s_{vh}$, and the bottom row of panels (b) assume $s_{hv} = -s_{vh}$. Columns (1) show co-polarised power anomaly, columns (2) cross-polarised power anomaly, columns (3) co-polarised phase difference, and columns (4) cross-polarised phase difference. Bright green dots represent azimuthal minima at each range bin, and the dark green line is the best estimate of the symmetry axis calculated using a Gaussian-weighted moving average of the azimuthal minima. Depths with insufficient SNR are greyed out.

[Figure]

**Figure AC2**. ApRES polarimetric power anomaly and phase difference measured at Site D, WAIS Divide. Legends are identical with those for Figure AC1. (Note that this is labelled as W03 in these plots as these are our internal numbering system)

**RC2.4. Terminology linked to azimuthal resolution**

**In numerous instances (e.g., l.7, l39, l49.. ) the authors advertise that synthesizing the azimuthal response from quad-polarimetric data (eq. 4) results in improved angular resolution compared to rotational setups. I disagree with that. The chosen azimuthal spacing of 1° (l. 397) is completely arbitrary and any value works with eq. 4. I agree that advantages and disadvantages of quad- polarimetric vs. rotational measurements should be discussed, but choosing an arbitrary gridding for *θ* is not enough in this regard. Also, no rotational dataset is presented so that the claims about the superiority of quad-polarimetric measurements are not rigorously substantiated (apart from the obvious fact that they are much quicker to obtain).**

AC2.4. Unfortunately we did not conduct azimuthal rotational measurements at WAIS Divide and so we are unable to make a full and direct comparison between quad-polarimetric and rotational measurements specifically with regards to our own datasets at WAIS Divide. However, comparisons between the two field methods were made when analysing ApRES data from Brisbourne et al. (2019) of which we found minimal difference in the results generated from the two methods.

That being said, Matsuoka et al. (2012) conducted an azimuthal rotational survey close to (~5 km from) our study site of which the results are shown in their Figure 3, Panel SW-24. Although they had used radar systems with different frequencies from ours (60 and 179 MHz as opposed to 300 MHz for ApRES), we can see that, at least to 1400 m, our estimate of the fabric orientation matches their results.

We acknowledge that we cannot fully claim the superiority of one measurement method over the other without conducting both at the same study site. We therefore add a paragraph in Section 5.3 that compares the two methods (azimuthal rotational measurements v. quad-polarimetric measurements) and cites recent papers (notably that of Ershadi et al., 2021) in addition to comparing our results with that of Matsuoka et al. (2012) to state that there should be no structural differences between the outputs of both methods (L417-429):

> Because we did not conduct azimuthal rotational measurements at our study sites, we are unable to make a full and direct comparison between quad-polarimetric and rotational measurements in terms of their output results, and therefore are unable to advocate for one method over the other. However, a visual comparison between our results and those obtained at Site S-W24 of Matsuoka et al. (2012) show similar polarimetric power anomalies in the upper 1400 m of ice, which give us confidence in our results. Separately, comparative analyses of results obtained using both types of measurements at Korff Ice Rise (C. Martin, *unpublished data*) as well as at EPICA Dronning Maud Land (Ershadi et al., 2021) reveal no structural differences between datasets. This comparative similarity may not hold in areas with more dynamic and/or complex flow, where the $E_3$ eigenvector is not vertically-aligned, and requires further investigation. While our estimation of the $E_1$ and $E_2$ eigenvector orientations in our measurements is to the nearest 1°, this precision reflects the angular bin size used to azimuthally reconstruct the received signal from quad-polarised data in this study, and is not synonymous with angular resolution, which instead is largely dependent on human errors in positioning the antennas for each acquisition (here assumed to be

±8°). However, under the assumption that the two acquisition methods do produce physically equivalent datasets, then a quad-polarimetric reconstruction allows for a comparatively higher precision in the identification of the two eigenvector orientations.

**Minor remarks**

**RC2.5. Abstract should state limitation that the methodology only works if one of the c-axis is pointing upwards.**

AC2.5. You make a fair point. We have explicitly mentioned this limitation in L3-4: "... at radar frequencies, with the assumption that one of the crystallographic axes is aligned in the vertical direction."

**RC2.6. l 57: Eq. 4 reconstructs the azimuthal response, but this is something different than resolving it. See comments above.**

AC2.6. See AC2.4 for a detailed response. For this specific reference, we have removed this claim in the sentence which now reads (L58-59): "This quadrature- (quad-) polarised setup significantly reduces the field time required to obtain each set of acquisitions."

**RC2.7. l 63: Specify what resolution you refer to. ApRES surely has lower potential for vertical resolution than ice-core data.**

AC2.7. ApRES certainly has lower resolution potential than ice-core data and we apologise for not being specific. The issue with defining the vertical resolution of ice cores is due to allocation of core sections to different scientific goals (at least, to the best of my knowledge), so while the depth resolution of ice cores in terms of fabric eigenvalue estimates can potentially be down to the centimeter scale, in practice the thin sections used to estimate eigenvalues are taken at larger intervals (~50-100m for the WAIS Divide ice core). The depth resolution of ApRES refers to the bulk-averaged fabric of the local depth window (15 m in our case). We remove our use of "resolution" and instead use terms like "bulk-averaged depth resolution" and "depth interval" for the vertical resolution of ApRES measurements of ice fabric. "Bulk-averaged depth resolution" follows the terminology of Jordan et al. (2019). For example, the lines referenced in this comment now state (L62-64): "...at a nominal bulk-averaged resolution of 15 m. We show that, using this setup and method, our estimates of fabric asymmetry are comparable to that from ice core thin sections taken at similar depth intervals. From our results, we explicitly determine…"

**RC2.8. l 65: Specify what the angular resolution is. It cannot be the 1° . I would also prefer more modest wording for "unambiguously". Defining the direction of the ApRES antennas alone is already error-inflicted and there is no rigorous statement in this paper on how this was done.**

AC2.8. We have replaced "unambiguously identify" with "explicitly determine". Because there is ambiguity in the attribution of angular resolution to either the model azimuthal bin size (1°) or the accuracy of orienting the antennas, we have removed "...to high angular resolution"

from this sentence. We state how we define the antenna orientation in L162-163 (see AC2.10 for a full description).

**RC2.9. Overall nice structure of the introduction. This works for me.**

AC2.9. Thanks for the compliment!

**RC2.10. Fig. 1a include orientation of the E-field vector. Statement that antennas are oriented parallel to the divide is conflicting with inference that principal axis is at θ = 90° (which is parallel to flow, which is oblique to the divide according to Fig 1b). See major comment 2.**

AC2.10. There is probably a confusion that originates from the assignment of *v* and *h*. In the original TCD submission, we followed the conventions of Brisbourne et al. (2019) and defined *v* to be parallel to the antenna E-plane (which *h* is perpendicular to the E-plane). In this revised draft, we opt for consistency among the recent wave of radar polarimetry manuscripts-in-submission, and switch to using the convention presented in Figure 2 of Ershadi et al. (2021), where alignment nomenclature corresponds to the E-field vector. Therefore, in our revised draft, *hh* now becomes *vv*, *hv* becomes *vh*, etc. This has been updated in Figure 1 (in which panels (a) and (b) are now in separate figures) which also includes the E-field vector. All plots are therefore shifted so that the x-axis is now [-90° +90°], and *θ* therefore now represents the angle from the antenna plane, which is orthogonal to our transect direction and ~parallel to WAIS Divide. The angle stated here (250°) is correct but the orientations of the velocity vectors and Divide orientation are not, and this has been updated in Section 2 and 3 to have all orientations be consistent.

We also explicitly write that the nomenclature of *h* and *v* vary between manuscripts (L164-167): "The nomenclature attached to the *h* and *v* alignments are indicative of the electric field (Fig. 1) and are consistent with those used in polarimetric ApRES studies of ice fabric (Jordan et al., 2019, 2020b, Ershadi et al., 2021) with the exception of Brisbourne et al. (2019), which reverses the two assignments (i.e. *h* in our study corresponds to *v* in their study, and *v* in our study study corresponds to *h* in their study)." This discrepancy took us some time and frustration to realise, and we hope that this statement can aid in preventing future confusion.

Because we depart from the *h* and *v* nomenclature of Brisbourne et al. (2019), we also forego the use of "symmetry axis". We instead define the axes identified by the cross-polarised power anomaly as the $E_1$ and $E_2$ eigenvector orientations, and distinguish between the two using the sign of the *hhvv* phase derivative following Jordan et al. (2019).

**RC2.11. l 114 not only ice-dynamics but also ice properties induced through climate variations imprint on the ice-fabric evolution. I am not sure the principal ice-fabric axis always line up with today's strain rate regime as suggested here.**

AC2.11. We have included mention of climate effects with the addition of the following sentence (L114-115): "In addition, fabric strength and orientation are also influenced to some extent by perturbations in climate (Kennedy et al. 2013)." We have reduced the certainty of the sentence regarding how the fabric axis lines up with the strain rate regime, and moved

this to be specific regarding the vertical girdle (L120-122): "In the case of a vertical girdle fabric, the *c*-axes are oriented in a girdle that is planar to the $E_2$ and $E_3$ eigenvectors, with the $E_1$ eigenvector indicative of the orientation of lateral flow extension at its corresponding age-depth (Brisbourne et al. 2019; Matsuoka et al. 2012)."

**RC2.12. l 127 the terminology "anisotropy" for *β* has confused me. You also need "anisotropy" for birefringence. Why not call it the "anisotropic reflection ratio" or something like that? "boundary reflection" is more appropriate than "boundary scattering" (the latter suggests some diffuse scattering which is not accounted for in this context. However, this may be a matter of taste.)**

AC2.12. We have redefined *β* to be the *intensity ratio parameter for anisotropic scattering*, in line with the majority of the previous radar polarimetric studies (e.g. Fujita et al. 2006, Drews et al. 2012). See AC1.7 for a full explanation.

**RC2.13. l 137 remove abbreviation SISO. It is not used later on.**

AC2.13. Removed. We have similarly removed the abbreviation "MIMO" in the same paragraph for the same reason.

**RC2.14. l 150 include uncertainties for this angle here and elsewhere.**

AC2.14. See AC2.18.

**RC2.15. l 169 double-check that $s_{vh} = s_{hv}$ (see comment above).**

AC2.15. We address this comment in AC2.3.

**RC2.16. I understand how this azimuthal phase difference is calculated, but I don't understand what the additional value is. What inferences are drawn from the co-polarised phase difference that cannot be drawn from the hhvv phase angle?**

AC2.16. The four-quadrant patterns in the co-polarised (*hh*) phase difference plots are coincident with the location of the nodes in the co-polarised power anomaly plots (Brisbourne et al. 2019). The location of these nodes and four-quadrant patterns then constrains the 90° ambiguity in the cross-polarised (*hv*) power anomaly (see AC2.19). We find it to be more clear to locate the nodes and check the fabric orientation using a combination of investigating the co-polarised nodes and four-quadrant patterns (as in Brisbourne et al. 2019) than in the *hhvv* phase angle plot. Having the co-polarised power anomaly and phase difference plot may be useful in case of an abrupt shift in fabric orientation (again, as evidenced in Brisbourne et al. 2019).

Because we choose to define *h* and *v* differently to Brisbourne et al. (2019), the four-quadrant patterns are now centered on the $E_2$ axis (they were centered on the $E_1$ axis in Brisbourne et al. 2019). However, we choose to still include this in the manuscript as it allows readers to see the relationship between power-based, phase-based, and (*hhvv*) coherence-based methods and connects past studies with recent, quantitative studies. We

also provide this explanation as an additional way to distinguish between the two eigenvectors (L401-405):

> However, if anisotropic scattering is present, the azimuthal location of the four-quadrant patterns in the co-polarised phase difference is also an effective way to discriminate between the two eigenvectors (Brisbourne et al. 2019). Here, the four-quadrant patterns are centred around the $E_2$ eigenvector (Fig. 4c, Fig. 5c). Although the results of Brisbourne et al. (2019) observe the patterns to instead be centred around the $E_1$ eigenvector, we can reconcile this discrepancy due to opposite assignments of $h$ and $v$ antenna alignments used between the two studies.

**RC2.17. l 193 This first paragraph is more methods to me than results.**

We have moved the sentences "A pad factor of 2… with the same moving matrix dimensions." to the end of Section 3.3, with slight alterations to the phrasing to account for its new placement. We have also added an explanation of the "pad factor" term within these sentences to address RC1.3.

**RC2.18. As stated above the 90° are suspicious. Also, what are the ±7° based on? I would think that errors in antenna positioning are larger.**

AC2.18. (Note that our definitions of $h$ and $v$ are switched and all measurements are shifted by 90°, see AC2.10) In AC2.3, we note that the $E_1$ eigenvector is more or less symmetric about 0°. The ±6° (originally 7°) is the standard deviation of the calculated eigenvector orientations from the cross-polarised power anomaly at each depth bin (azimuthal minima, bright dots for the $E_1$ (green) and $E_2$ (yellow) eigenvectors in Figure 4). These errors are unrelated to antenna positioning. We have adopted Brisbourne et al. (2019)'s nominal assignment of ±8°as the human error associated with antenna positioning (L163-164). We have stated the independence of the two errors in L250-251.

**RC2.19. l 213 Typo? This ambiguity cannot be resolved in the *hh* power anomaly shown in Fig 2b. You need to use the polarity of the phase gradient.**

AC2.19. On the contrary: if anisotropic scattering is present in the imaged ice, the 90° ambiguity in the cross-polarised ($hv$) power anomaly can be resolved by determining either (i) the azimuthal minima of the co-polarised ($hh$) power anomaly plot, or (ii) the azimuth of the centre of the four-quadrant patterns in the co-polarised phase difference plots, as stated in Section 5.1 of Brisbourne et al. (2019). If co-polarised nodes are weak due to strong anisotropic scattering, the sign reversals in the four-quadrant patterns remain diagnostic. On the other hand, if there is no anisotropic scattering present, then both the co-polarised power anomaly and phase difference plots will present 90° ambiguity as shown in Figure 2a of Brisbourne et al. (2019).

This being said, we rely on the polarity of the phase gradient in resolving the $E_1$ and $E_2$ eigenvectors in the revised draft (see AC2.16). We have modified this section to now reflect our updated analysis (L241-247):

By tracing the azimuthal minima in the cross-polarised power anomaly profiles through depth (Fig. 4c), we can identify the orientations of the $E_1$ and $E_2$ eigenvectors (Li et al., 2018). However, because there exists a 90° ambiguity in the cross-polarised power anomaly profiles, we rely on the sign of the gradient of the *hhvv* phase angle (Fig. 4e) to distinguish between the two eigenvectors. Because the the $E_1$ and $E_2$ eigenvectors align with the directions of the smallest and largest dielectric permittivities respectively, the location of the azimuthal minima resulting in a negative $\phi_{hhvv}$ gradient through depth indicates the direction of the $E_1$ eigenvector, and the azimuthal minima resulting in a positive $\phi_{hhvv}$ gradient indicates the direction of the $E_2$ eigenvector (Jordan et al., 2019).

**RC2.20. l 221 at least estimate these "human" errors**

AC2.20. See AC2.18.

**RC2.21. why is the "anisotropy" an integer value?**

AC2.21. This comment is a duplicate of RC1.7, please see our response there.

**RC2.22. l 237 I think I missed something here: Aren't those node pairs simply depths where the phase shift between ordinary and extraordinary wave is odd integer multiple of $\pi$ ? Clearly they will have a correspondence in the azimuthal phase difference (which is directly related to the phase angle). I don't understand the deeper physical implication of this 'four quadrant pattern yet.**

AC2.22. See AC2.16 for further discussion on the relevance of the co-polarised phase difference.

**RC2.23. Fig. 4 I appreciate the error bars on the ApRES derived $E_2 − E_1$. Please state more clearly how those were derived.**

AC2.23. We state our error calculation in L206-207: "...with the associated phase error (standard deviation) estimated through the Cramer-Rao bound, following the methods of Jordan et al. (2019)". $E_2$ - $E_1$ is obtained directly through $d\phi_{hhvv}/dz$, and the error is then transferred accordingly. We have made this clear with the addition of the following sentence (L218-219): "From here, estimates of $d\phi_{hhvv}/dz$ and their respective errors for each depth bin were both scaled using Equation 7a to then produce estimates and uncertainties for $E_2$ - $E_1$."

**RC2.24. l 289 This is not the "best model" that matches observed results. It is a model that explains some of the features in the observations**

AC2.24. We have reworded this sentence to (L331-332): "The power anomaly model that we used to emulate measured results (Fig. 5a) incorporated a variable anisotropic scattering ratio…"

**RC2.25. l 330 How fast does the ice-fabric structure adapt to a new strain regime? I think some sort of statement in this regard is required to better justify statements of ice-divide stability.**

AC2.25. Brisbourne et al. (2019) states that, although the time taken to overprint a preexisting fabric is dependent on temperature, strain rate, and stress regime, it is poorly constrained excluding those from laboratory results. Therefore, we can implement the same reasoning as that of Brisbourne et al. (2019) and state that the measured COF distribution aligned with the observed surface strain orientation through depth reflects the current ice flow regime, assuming that the vertical assumption (i.e. $E_2$ - $E_1$) holds through the range of measured depths.

We have summarised the above explanation into Section 5.2 (L376-382):

> "Although the time taken to overprint a pre-existing fabric is poorly constrained, excluding those from laboratory results (Brisbourne et al. 2019), the removal of previous fabric evidence is thought to take significant time and may require anomalously strong deformation regimes (Alley 1988). At all sites, the alignment of our identified $E_1$ eigenvector orientation with the observed present-day strain regime is consistent with theory relating ice flow and crystal anisotropy (Azuma 1994). We are confident that the observed surface strain orientation likely reflects the current deformation regime, given this alignment, the temporal permanence of fabric signatures, and the comparatively short depth-age of our record."

**References**

Brennan, P. V., Nicholls, K. W., Lok, L. B., and Corr, H. F. J.: Phase-sensitive FMCW radar system for high-precision Antarctic ice shelf profile monitoring, IET Radar, Sonar & Navigation, 8, 776–786, https://doi.org/10.1049/iet-rsn.2013.0053, 2014.

Brisbourne, A. M., Martín, C., Smith, A. M., Baird, A. F., Kendall, J. M., and Kingslake, J.: Constraining Recent Ice Flow History at Korff Ice Rise, West Antarctica, Using Radar and Seismic Measurements of Ice Fabric, Journal of Geophysical Research: Earth Surface, 124, 175–194, https://doi.org/10.1029/2018JF004776, 2019.

Burr, A., Noël, W., Trecourt, P., Bourcier, M., Gillet-Chaulet, F., Philip, A., and Martin, C. L.: The anisotropic contact response of viscoplastic monocrystalline ice particles, Acta Materialia, 132, 576–585, https://doi.org/10.1016/j.actamat.2017.04.069, 2017.

Dall, J.: Ice sheet anisotropy measured with polarimetric ice sounding radar, in: 30th International Geoscience and Remote Sensing Sympo- sium (IGARSS 2010), pp. 2507–2510, Honolulu, HI, https://doi.org/10.1109/IGARSS.2010.5653528, 2010.

Fujita, S., Maeno, H., and Matsuoka, K.: Radio-wave depolarization and scattering within ice sheets: A matrix-based model to link radar and ice-core measurements and its application, Journal of Glaciology, 52, 407–424, https://doi.org/10.3189/172756506781828548, 2006.

Horgan, H. J., Anandakrishnan, S., Alley, R. B., Burkett, P. G., and Peters, L. E.: Englacial seismic reflectivity: Imaging crystal-orientation fabric in West Antarctica, Journal of Glaciology, 57, 639–650, https://doi.org/10.3189/002214311797409686, 2011.

Howat, I. M., Porter, C., Smith, B. E., Noh, M.-J., and Morin, P.: The Reference Elevation Model of Antarctica, The Cryosphere, 13, 665–674, https://doi.org/10.5194/tc-2018-240, 2019.

Jordan, T. M., Schroeder, D. M., Castelletti, D., Li, J., and Dall, J.: A Polarimetric Coherence Method to Determine Ice Crystal Orientation Fabric From Radar Sounding: Application to the NEEM Ice Core Region, IEEE Transactions on Geoscience and Remote Sensing, pp. 1–17, https://doi.org/10.1109/tgrs.2019.2921980, 2019.

Jordan, T. M., Schroeder, D. M., Elsworth, C.W., and Siegfried, M. R.: Estimation of ice fabric within Whillans Ice Stream using polarimetric phase-sensitive radar sounding, Annals of Glaciology, 61, 74–83, https://doi.org/10.1017/aog.2020.6, 2020a.

Jordan, T. M., Martin, C., Brisbourne, A. C., Schroeder, D. M., and Smith, A. M.: Radar characterization of ice crystal orientation fabric and anisotropic rheology within an Antarctic ice stream, Earth and Space Science Open Archive ESSOAr, pp.1-48, https://doi.org/10.1002/essoar.10504765.1, 2020b.

Kennedy, J. H., Pettit, E. C., and Di Prinzio, C. L.: The evolution of crystal fabric in ice sheets and its link to climate history, Journal of Glaciology, 59, 357–373, https://doi.org/10.3189/2013JoG12J159, 2013.

Looyenga, H.: Dielectric constants of heterogeneous mixtures. Physica, 31, 401–406, https://doi.org/10.1016/0031-8914(65)90045-5, 1965.

Matsuoka, K., Power, D., Fujita, S., and Raymond, C. F.: Rapid development of anisotropic ice-crystal-alignment fabrics in- ferred from englacial radar polarimetry, central West Antarctica, Journal of Geophysical Research: Earth Surface, 117, 1–16, https://doi.org/10.1029/2012JF002440, 2012.

Morlighem, M., Rignot, E., Binder, T., Blankenship, D., Drews, R., Eagles, G., Eisen, O., Ferraccioli, F., Forsberg, R., Fretwell, P., Goel, V., Greenbaum, J. S., Gudmundsson, H., Guo, J., Helm, V., Hofstede, C., Howat, I., Humbert, A., Jokat, W., Karlsson, N. B., Lee, W. S., Matsuoka, K., Millan, R., Mouginot, J., Paden, J., Pattyn, F., Roberts, J., Rosier, S., Ruppel, A., Seroussi, H., Smith, E. C., Steinhage, D., Sun, B., van den Broeke, M. R., van Ommen, T. D., van Wessem, M., and Young, D. A.: Deep glacial troughs and stabilizing ridges unveiled beneath the margins of the Antarctic ice sheet, Nature Geoscience, 13, 132–137, https://doi.org/10.1038/s41561-019-0510-8, 2020.

Young, T. J., Schroeder, D. M., Christoffersen, P., Lok, L. B., Nicholls, K. W., Brennan, P. V., Doyle, S. H., Hubbard, B., and Hubbard, A.: Resolving the internal and basal geometry of ice masses using imaging phase-sensitive radar, Journal of Glaciology, 64, 649–660, https://doi.org/10.1017/jog.2018.54, 2018.

---

## Referee Report (RR1)

**General comments**

The authors have addressed all issues raised in the initial review and incorporated most suggested changes. In particular the re-analysis (identifying $s_{hv} = -s_{vh}$ and a rounding error), triggered by comments from Reinhard Drews, has improved the manuscript.

The discussion sections now offer a more measured comparison between the presented quad-polarimetric measurement and traditional rotational approach. It is now also acknowledged that a direct quantitative comparison is not possible, as no rotational data-set was obtained in conjunction with the quad-polarimetric data.

While I generally believe the paper to be in good shape, it would be appreciated if the authors could consider some outstanding issues as detailed below.

**Specific comments**

- The newly provided Figure 3 (mean power return) and in particular the insets are greatly appreciated and address concerns about the observed reciprocity of $s_{hv}$ and $s_{vh}$ as raised in the initial review (RC1.12). The reciprocity seems to hold to a very high degree in the deep ice ~1100 m. Yet the same can not be said for the second inset ~100 m. The explanations offered by the authors (AC1.12) do not explicitly address this depth dependence.

  Regardless of the underlying cause, the fabric derivation is likely not applicable in regions where reciprocity is not experimentally observed and it may be warranted to restrict the analysis to depths where it is (in addition to the newly introduced coherence criterion). This may also resolve the surprisingly large anisotropy derived for the firn.

- The addition of data from 9 additional close-by locations, not mentioned in the original manuscript, is a great addition to the paper. While I agree with the chosen presentation (relegating most plots to supplementary material), I would encourage the authors to also perform the fabric calculation for these locations.

  The distance between locations appears to be small enough for the fabric not to be expected to change significantly (and is partially even within the estimated 1000 m beam cone at 1500 m depth) and similar to the distance to WAIS which is already being compared to. The spread between sites as well as the average in comparison to WAIS, might also yield more quantitative insight into the reliability of the method.

**Technical comments**

- Page 12, line 244 doubled "the".

---

## Author Response (AR2)

**Second author response to the review of Young et al. "Rapid and accurate polarimetric radar measurements of ice crystal fabric orientation at the Western Antarctic Ice Sheet (WAIS) Divide deep ice core site" [Manuscript # tc-2020-264]**

This response is in regards to the responses following our first revision (11 March 2021) to the original manuscript submission (28 September 2020). We would like to thank the managing editor (Kenichi Matsuoka) and the two reviewers, Martin Rongen and Reinhard Drews, for their reviews . Please find below the editor's (EC) and the two reviewers' comments (RC) in **bold**, each followed by the authors' response (AC). Line, figure, and page numbers mentioned by the reviewers (within RC) refer to the second version of the manuscript (the revision dated to 11 March 2021) and those mentioned by the authors (within AC) refer to the current version of the manuscript.

TJ Young
Scott Polar Research Institute
13 May 2021

**Review by Editor (Kenichi Matsuoka)**

**General comments**

**EC0.1. Thanks for submitting a thoughtfully revised manuscript. Both referees provided highly constructive comments and I am very glad to see that the authors made satisfactory responses. I recommend the authors to follow all new suggestions made by the referees and calculate depth profiles of E2-E1 at all sites and plot them in supplemental figure, possibly together with the data from site I or from ice core to facilitate reader's comparison between this new figure and Fig. 6. Also, please consider my own comments listed below; all are relatively minor.**

**The authors declared that the full dataset and relevant codes will be released through the UK polar data center, but the link given here is generic. As the review process is nearly finalized now, please upload all relevant files, receive DOI for these resources, and include specific URL/DOI to the manuscript.**

**I request a minor revision. Please submit the revised manuscript, the manuscript with track changes, and the brief response letter.**

AC0.1.  Thank you for your constructive comments and we have responded to each of them in turn below. We apologise for not providing a reference to the dataset and relevant codes in the last revision, as they were in the process of a lengthy quality control. The dataset is now published and the DOI is 10.5285/BA1CAF7A-D4E0-4671-972A-E567A25CCD2C. This as well as the citation (Young and Dawson 2021) are included in the Data Acknowledgements section.

We will attach the code to process the datasets to the DOI once the review process closes, in case there are any further corrections that may involve potential changes to the code.

**Specific comments**

**EC0.2. Figure 1 caption: add geographical orientation (ESE, 110o) of the horizontal-polarization plane. Illustrations representing antennas are unnecessarily complicated; just show the antenna element orientation. The WAIS Divide orients 20 degrees, and the horizontal plane orients 110o, but these two orientations are shown the same in the illustration.**

AC0.2. We choose to keep the illustrations of the antennas, because we suspect (but are not sure) that the specific $h$ and $v$ orientations may play a role in whether $s_{hv} = s_{vh}$ or $s_{hv} = -s_{vh}$ in the radar processing steps. In other words, whether the cable exits upwards or downwards when the antenna is oriented in the $v$ direction may potentially affect the processing output. This figure shows the exact antenna orientations that we use for each of the four acquisitions.

The horizontal plane orients 110°, which is approximately parallel to the WAIS Divide (~110°).

To address this comment, we have explicitly stated the horizontal and vertical polarisation orientation in the figure as well as in the caption.

**EC0.3. Figure 2: adjust color for strain components. Dark blue and red are very hard t read. Clarify the unit of elevations (m a.s.l.?, or mention the geoid height in the caption). Include intervals of the surface elevation contours (10 m), and the ice thickness at site I in the caption.**

AC0.3. I have tried adjusting the colours but making the strain components a lighter shade unfortunately is even less distinct. I have adjusted the scale of the background (bed elevation) so that the strain components stand out more. The elevation units (surface as well as bed) now both read m a.s.l., and contour intervals are clarified in the caption.

**EC0.4. P7L134: The equation of \epsilon (z) is, I think, incorrect. The authors intend to define \delta\epsilon(z), permittivity anisotropy in the horizontal plane. Figure 5 caption gives the range of \epsilon(z) [0 1.5e-2], but I think it is the range of \delta\epsilon(z), not \epsilon(z).**

AC0.4. You are correct on both parts of your comment: the equation should read $\Delta\varepsilon(z) = \Delta\varepsilon(z)' (E_2 - E_1)$, and Figure 5 should give $\Delta\varepsilon(z)$. This has been corrected in the text and in Figure 5 (both in the figure and in the caption).

**EC0.5. Figure 3: add the level where SNR is considered too low (e.g. horizontal bar showing the minimum signal level that makes adequate SNR, or depth ranges where SNR is too low).**

AC0.5. This has been added as an additional panel to the top of Figure 3.

**EC0.6. P11L240: spell out FIR (finite impulse response?)**

AC0.6. Done.

**EC0.7. P11L269: change "the directions of the smallest…" to "the orientations of…."**

AC0.7. Done.

**EC0.8. P13L300: E1 and E2 orientations are "mostly" depth invariant. So, add "mostly" or indicate the range of depth variability like "depth invariant (within +/- 6 degrees throughout the depth range, Fig. 4a and Table S1)."**

AC0.8. See AC0.9.

**EC0.9. Figure 5: remove WDC from the beginning of the caption. I think \epsilon(z) is indeed \delta \epsilon (z). At the end of the cation it is said that E1 and E2 plane orientations are set to 0o and 90o following measured observations, but it is not accurate enough. See my comment on P13L300.**

AC0.9. WDC has been removed from the caption. Following AC0.4, $\varepsilon(z)$ has been changed to $\Delta\varepsilon(z)$. Because the model requires $E_1$ and $E_2$ to be separated exactly by 90° (as eigenvectors are supposed to be theoretically orthogonal to each other), we chose to offset the measured $E_1$ and $E_2$ orientations (91° and -3°, as stated on L256) by -2 and +2 respectively. As a result, Figure 3 was generated using prescribed $E_1$ and $E_2$ orientations of 89° and -1° respectively. These sentences now read (L277-281):

> For the model, we fixed the $E_1$ and $E_2$ eigenvector orientations at 89° and -1° respectively. These values are -2° and +2° from their measured orientations of 91° and -3° respectively, and this adjustment was made to satisfy the orthogonality of eigenvectors of a symmetric matrix. For simplicity, we prescribed both eigenvectors as depth-invariant, given that both their measured standard deviations were only ±6° across the measured depth range of 1500 m (Fig. 4).

**EC0.10. P14L315: Now Discussion Section is largely expanded so revise this sentence to narrowly indicate the discussion about this point.**

AC0.10. We are not sure which point you are referring to--we have tentatively assumed that it is the sentence about anisotropic scattering (originally L284-285). We have deleted "...the reasons behind elaborated in the Discussion (Section 5)." We have added the following sentence to Section 5.1 in the Discussion (L350-353): "Although the observed anisotropic scattering can be exploited to infer the strength of the third eigenvalue ($E_3$) under assumptions of fabric isotropy at the ice surface (Ershadi et al., 2021), we do not attempt this method given our observations of significant fabric anisotropy in the firn layer (inset of Fig. 6)."

**EC0.11. P15L304: Is beta linearly interpolated and shown in dB scale? It seems that beta is linearly interpolated over the dB scale. Clarify. Also, is it linearly interpolated**

**between 10-m-separated data points or is it linearly fit using the all data together? Fig. 5 colorbar shows that beta varies very smoothly.**

AC0.11. $\beta$ is linearly interpolated between 100 m depth intervals, where at each 100 m depth interval, $\beta$ was estimated to the nearest integer on the linear (as opposed to the dB) scale. The range of $\beta$ on the linear scale is [1 6] which is equivalent to [0 15.6] on the dB scale. We have added the decimal digit in the caption. We have also clarified this in the surrounding sentences, which now reads (L283-286): "At each 100 m depth interval (i.e.100 m, 200 m, 300 m, etc.), $\beta$ was estimated to the nearest integer before converting to the dB scale and linearly interpolated to match the model depth step of 1 m."

**EC0.12. Figure 6 caption (and relevant locations): is the firn correction made only for permittivity anisotropy in the horizontal plane, or both this and mean propagation speed (giving slightly different depths of these data points)? Please clarify.**

AC0.12. Thank you for spotting this detail. We have made firn correction only for fabric strength (horizontal permittivity anisotropy) and not for mean propagation speed. We have clarified this in the caption of Figure 6 as well as in the Methods (L213-215): "Firn correction was implemented only for the permittivity anisotropy in the horizontal plane and not for the mean propagation speed (i.e. no depth correction was made to account for the effect firn density on wave propagation speeds)."

**EC0.13. Figure 6: consider adding beta to this plot.**

AC0.13. We do not implement this suggestion as $\beta$ is modelled, which contrasts with the content of Figure 6 which are both measured results.

**EC0.14. P21L497: Include diameter of the first Fresnel zone. Sampled ice volume is largely constrained by the size of the Fresnel zone, not by the antenna aperture.**

AC0.14. We have removed mention of antennas in the section and have added the size of the first Fresnel zone at two nominal depths (L468-472): "In contrast, waveform-based methods average out fabric properties in bulk where, for radar systems, the planar footprint of which the COF is averaged from is dependent on the radius of the first Fresnel zone (Haynes, 2018). This footprint would be approximately 6 m in radius at a depth of 100 m, and expands to approximately 23 m in radius at 1400 m. Therefore, the bulk COF estimates obtained from ApRES is averaged from a much larger area at depth than near the surface."

**EC0.15. Table S1: Add two columns to show geographical orientations of E1 and E2. Add \pm before all values showing standard deviations (as it is done throughout the text). I assume that these orientations are measured clockwise, which is not mentioned explicitly; please clarify.**

AC0.15. We have added two columns that give the geographical orientations of $E_1$ and $E_2$.

The orientations are actually measured counterclockwise, which means that the numbers in the text are slightly off (L258). The cardinal directions for $E_1$ would then be 110° - 91° = 19°, and for $E_2$, 110° - (-3°) = 113°. The offset from the flow direction is therefore mod(230,180)° -

19° = 31°. Figure 4f is correct and remains unchanged, and the relative principal axis to the nearest strain configuration of -7° also remains unchanged. The reason for this error is that our code automatically converted all directions to polar stereographic (south) coordinates for plotting purposes, and calculates the offsets in this reference frame. The conversion from polar stereographic orientation to true cardinal orientation (relative to True North) was done by hand at the very end, which therefore resulted in this oversight. We apologise for this confusion and have updated the numbers accordingly.

In light of this, we have made the explicit mention that positive angular shifts in the polarimetric reconstruction results in *counterclockwise* rotation (L256-257), and have updated the numbers accordingly.

**EC0.16. COF, ice fabric, etc are used interchangeable ways. Please make the manuscript more consistent.**

AC0.16. Done. We have attempted to refer to "COF" when describing the fabric at the crystal scale and "fabric asymmetry" and "orientation" at the bulk (depth-averaged) scale. Please let us know if you agree (or disagree!) with our nomenclature.

**EC0.17. WDC can be replaced with "WAIS Divide Ice Core", "core site" or such. So, please consider avoiding the acronym.**

AC0.17. Done.

**Review by Martin Rongen**

**General comments**

**RC1.1. The authors have addressed all issues raised in the initial review and incorporated most suggested changes. In particular the re-analysis (identifying shv = −svh and a rounding error), triggered by comments from Reinhard Drews, has improved the manuscript.**

**The discussion sections now offer a more measured comparison between the presented quad-polarimetric measurement and traditional rotational approach. It is now also acknowledged that a direct quantitative comparison is not possible, as no rotational dataset was obtained in conjunction with the quad-polarimetric data. While I generally believe the paper to be in good shape, it would be appreciated if the authors could consider some outstanding issues as detailed below.**

RC1.1. Thank you for your critical eye and we hopefully have addressed your remaining concerns in the responses below.

**Specific comments**

**AC1.2. The newly provided Figure 3 (mean power return) and in particular the insets are greatly appreciated and address concerns about the observed reciprocity of shv and svh as raised in the initial review (RC1.12). The reciprocity seems to hold to a very high degree in the deep ice ~1100m. Yet the same can not be said for the second inset ~100m. The explanations offered by the authors (AC1.12) do not explicitly address this depth dependence.**
**Regardless of the underlying cause, the fabric derivation is likely not applicable in regions where reciprocity is not experimentally observed and it may be warranted to restrict the analysis to depths where it is (in addition to the newly introduced co-herence criterion). This may also resolve the surprisingly large anisotropy derived for the firn.**

RC1.2. We agree with the reviewer that the reciprocity between *hv* and *vh* in the deeper signals provide greater confidence in the estimates and interpretation of fabric at those depths. However, fabric strength ($E_2 - E_1$) is calculated using the polarimetric coherence methods (Eqs. 6 and 7) which rely only on the *hh* and *vv* components, and not the *hv* nor *vh* components. Nothing else seems to suggest that the calculated fabric in the near surface (firn, <200 m) is unreal. Given this, we believe our results in the firn layer to be still valid, and we would like to include these results in case some readers are particularly interested in those depth. However, in response to your point, we now explicitly note the lack of reciprocity in *hv* and *vh* at those depths (while also noting that they are not used in the fabric strength calculation) (L359-361) as both an interpretive caution and as a potential avenue for further inquiry for interested readers.

**AC1.3. The addition of data from 9 additional close-by locations, not mentioned in the original manuscript, is a great addition to the paper. While I agree with the chosen presentation (relegating most plots to supplementary material), I would encourage the authors to also perform the fabric calculation for these locations.**
**The distance between locations appears to be small enough for the fabric not to be expected to change significantly (and is partially even within the estimated 1000m beam cone at 1500m depth) and similar to the distance to WAIS which is already being compared to. The spread between sites as well as the average in comparison to WAIS, might also yield more quantitative insight into the reliability of the method.**

RC1.3. We have included a 10-panel figure in the Supplementary Information (Figure S11). Although the output $E_2 - E_1$ values for some panels are more robust than others, the fabric does not change significantly between sites. Although we are able to individually tune the parameters to ensure more robust results for any one site, we choose to use the same parameters as those chosen for Site I for consistency and transparency.

We have added a short paragraph at the end of Section 4.3 to summarise the areal spread of the fabric strength calculations (L330-333):

> Estimates of azimuthal fabric asymmetry at the other 9 sites reveal similar trends, with those situated closer to the WAIS Divide ice core site in general showing a higher correlation with the core-derived fabric estimates (Fig. S11). The match between the ApRES and ice core fabric asymmetry estimates were generally higher

at depths below 1000 m. Larger errors were observed where ApRES estimates
deviated from the depth-coincident ice core measurement.

**Technical comments**

**AC1.4. Page 12, line 244 doubled "the".**

RC1.4. Done. There was also a similar error at L233 (the line numbers referring to the
previous draft) that we also fixed.

**Review by Reinhard Drews**

**General comments**

**RC2.1. The authors have responded in detail and convincingly to my review. I am glad
to see that comments were perceived as helpful and triggered a number of changes.
The inclusion of the 10 additional sites is helpful as is the study relating to the
mysterious minus sign in the quad-pole synthesizing. The problem is not solved, but
it will help others (and myself) to see this written out explicitly. As detailed below, I
have only very minor comments on the current version and don't insist on the
implementation of any of them. This is a nice paper and I congratulate the authors for
a job well done.**

AC2.1. Thank you for your review and we are glad that you approve of the direction of the
manuscript! We have hopefully addressed your remaining concerns below.

**Specific comments**

**RC2.2: This study resolves depth-variability of COF on a 15 spacing which is correctly
advertised as a step forward compared to previous studies. Maybe it is worthwhile to
elaborate a little bit more about the limits in depth-resolution. I suppose that it is
linked to calculating the polarimetric coherence phase which contains some vertical
averaging, but this is not explicitly stated.**

AC2.2. I share the same suspicions on the resolution of the *hhvv* phase being the primary
limiting factor of the bulk-averaged depth resolution of fabric asymmetry. Your suggestion of
elaborating on this fact is a good one. We have added the following paragraph to Section 5.3
(L409-420):

> In this study, we were able to achieve coherent estimates of fabric strength at depth
> intervals (the bulk-depth resolution) down to 15 m. In combination with a
> convolutional derivative, the use of depth averaging improves fabric estimates by
> reducing noise, removing anomalous "phase excursions", and isolates the effects of
> propagation-related phase behaviour with that from scattering (Dall 2010, Jordan et
> al. 2019). A limitation of this method is that, due to the depth-averaging when
> calculating the *hhvv* phase, it is not suited to detect and calculate fabric strength at

and crossing fabric boundaries (Jordan et al. 2019). Additionally, the size of the window and filter used is important, especially when applied over sections with high fabric asymmetry. As the depth periodicity of asymptotes present in the *hhvv* phase angle (which manifests in phase wrapping) are proportional with azimuthal fabric strength (Fig. 3), a large window has a greater risk of smoothing over these areas. This caveat may possibly be the reason behind the cluster of anomalously low measured $E_2$ - $E_1$ values at depth (Fig. 6). Conversely, a smaller window may naturally produce results with higher variability as a result of lower number of samples used to calculate the bulk-average. Therefore, with respeco the methods used in this study, there is like a delicate balance between the bulk-average resolution and precision.

**RC2.3. Another take-away (at least for me) is that COF patterns do not change significantly over this 10 km profile. Also in Ershadi et al. (TCD) differences along a similar transect are not very significant. Knowing this, if I were to collected new data somewhere in the ice-sheet center I would aim for larger distances between sites (but obviously this is site-specific).**

AC2.3. We have added the following sentence to the Conclusion (L523-525): "These observations of fabric orientation and strength were consistent for all 10 measured sites along a 6 km-long transect extending away from the core site."

**RC2.4. Fig 1a: It seems that some of the compression axis at approximately -460 km polar stereographic northing are¬ blue but should be red.**

AC2.4. We have checked the dataset that these measurements come from and they show extension at both orthogonal axes (which is odd). We have removed the smaller extensional axis and made a note in the caption that says that measured strain rates below 2.5 x $10^{-5}$ $a^{-1}$ are not shown.

**RC2.5. Fig 1a: The velocity arrows (black?) are not arrows but start with a dot. Maybe double check that this is consistent with what is shown in the legend.¬¬**

AC2.5. Thanks for noticing the inconsistencies. We have modified the legend so that it reflects the dot and line. We have also removed the arrowheads for the strain rate tensors as they were very small.

**RC2.6. Fig 1a: Consider putting white line and red dot in foreground. I was looking for it for a while.**

AC2.6. The white line and red dot are already in the foreground but you are correct that it is difficult to spot among the other information within the figure. We have made the white star (WAIS Divide Ice Core site) larger and have specified in the caption that the white line and red dot are ~5 km NE of the core site.

**RC2.7. L 96: "…birefringence reflects the bulk COF as [WORD MISSING].."**

AC2.7. Sorry! "as" has been removed.

**RC2.8. L 125 "… eq. 1 relates to microscopic ice fabric anisotropy.." not sure what microscopic refers to here. It could be misunderstood as single-crystal.**

AC2.8. We have clarified the various components of this sentence which now reads (L126-128): "This equation directly relates the bulk-averaged ($Δε$) and crystal ($Δε'$) birefringence anisotropy to dielectric anisotropy, which serves as the basis for the radar processing methods that follow."

**RC2.9. L 130f: It is possible that I missed it, but it should be clearly defined that the angle \theta is in the horizontal. Also I would prefer to call R' the "inverse" of R which is the same as the transpose in this case. However, possibly this makes it clearer to the readers indicating that R' undoes what R did.**

AC2.9. Corrected (for both points).

**RC2.10. L 224 state how SNR threshold is estimated**

AC2.10. We have moved this sentence up to the last paragraph of the Methods section, and added two additional sentences to explain how we calculate and implement the SNR threshold (L218-222): "We restrict our observations of the co- and cross-polarised measurements only to measurements with sufficiently high signal-to-noise ratios (SNR). For each of the four acquisitions, the SNR was found by calculating the 95th percentile of the noise floor. Observations were excluded from the output if the magnitude of the complex amplitude of any one acquisition falls below the calculated SNR for any one acquisition at a given depth (Fig. 3). "

**RC2.11. Figure 4: Nice Figure. Consider adjusting flow arrows in 4f as in comment to Fig 1a. It is unclear if ice flow is from top-left to bottom-right or the other way around.**

AC2.11. We have chosen to leave the flow markers and tensor as is (this is a direct copy of Conway & Rasmussen 2009). We have written the flow direction (230°, SW) in the caption.

**RC2.12. L 490 add reference point to the 21 \pm 8 degrees (e.g., relative to magnetic North..)**

AC2.12. Added the reference point (to true North).

**RC2.13. L 440 For information: COF in the NEEM ice core is (and will be) measured at much much higher vertical resolution. Those results will be interesting.**

AC2.13. This is interesting, thank you for the information!

**References**

Conway, H. and Rasmussen, L. A. (2009) Recent thinning and migration of the Western Divide, central West Antarctica, Geophys. Res. Lett., 36(12), 1–5. doi:10.1029/2009GL038072.

Young, T. J., & Dawson, E. J. (2021). Quad-polarimetric ApRES measurements along a 6 km-long transect at the WAIS Divide, December 2019 (Version 1.0) [Data set]. NERC EDS UK Polar Data Centre. doi:10.5285/BA1CAF7A-D4E0-4671-972A-E567A25CCD2C